# Entanglement entropy in quantum spin chains with broken parity number symmetry

A. Jafarizadeh[1*] and M. A. Rajabpour[1]

**1** Universidade Federal Fluminense, Niterói, Brazil
* arashjafarizadeh@id.uff.br

January 13, 2022

## Abstract

Consider a generic quantum spin chain that can be mapped to free quadratic fermions via Jordan-Wigner (JW) transformation. In the presence of arbitrary boundary magnetic fields, this Hamiltonian is no longer a quadratic Hamiltonian after JW transformation. Using ancillary sites and enlarging the Hamiltonian we first introduce a bigger quadratic Hamiltonian. Then we diagonalize this enlarged Hamiltonian in its most generic form and show that all the states are degenerate because of the presence of a zero mode. The eigenstates of the original spin chain with boundary magnetic fields can be derived after appropriate projection. We study in depth the properties of the eigenstates of the enlarged Hamiltonian. In particular we find: 1) the eigenstates in configuration bases, 2) calculate all the correlation functions, 3) find the reduced density matrices, 4) calculate the entanglement entropy. We show that the generic eigenstate of the enlarged Hamiltonian (including the eigenstates of the original spin chain) breaks the parity number symmetry and consequently one needs to take care of some technicalities regarding the calculation of the reduced density matrix and entanglement entropy. Interestingly we show that the entanglement structure of these eigenstates is quite universal and independent of the Hamiltonian. We support our results by applying them to a couple of examples.

# 1 Introduction

There exist many quantum spin chains which can be transformed into quadratic fermion models using Jordan Wigner (JW) transformation, the spin-1/2 $XY$ chain is just one example of such spin

chains. These fermionic models have been studied thoroughly in the past, and it was shown that they are exactly solvable [1]. In the context of the non-interacting fermions, the calculation of some quantities such as reduced density matrix (RDM) [2,3], entanglement [4–7] and formation probabilities [8–10] can be written in terms of correlation functions, which reduces the adversity of such calculations. The connection to the free fermion models also makes it possible to study the Rényi entanglement entropy for excited eigenstates of free fermions and related spin chains [11–19].

Most of the above mentioned studies were based on the bulk properties, however, there are also studies regarding the entanglement entropy in systems with boundaries and impurities. In the presence of boundaries analytical and numerical calculations of the entanglement entropy can be more challenging due to the lack of the translational symmetry. The entanglement entropy of a few quantum chains in the presence of the boundaries has been studied with analytical and numerical techniques, see for instance [20–27]. An interesting consequence of presence of the boundary is the breach of connection between spin chains and quadratic fermion models, specially in subsystem entanglement [27,28]. An spin chain (containing $L$ spins) with arbitrary boundary magnetic fields can be modeled as: bulk Hamiltonian plus boundary terms,

$$H_{\text{SpinChain}} = H_{\text{bulk}} + H_{\text{boundary}}. \tag{1}$$

The $H_{\text{boundary}}$ resembles the effect of boundary on far end sites or spins. For instance, a general boundary condition produced by external magnetic fields reads as $H_{\text{boundary}} = \vec{B}_1.\vec{S}_1 + \vec{B}_L.\vec{S}_L$, where $\vec{S}_1$, $\vec{S}_L$ are spin operators at the beginning and end of the chain, and $\vec{B}_L$, $\vec{B}_1$ denote the preferred direction of alignment of boundary magnetic fields. While such a non uniform boundary condition can be physically valid for a spin chain, the fermionization of such a spin chain would end up in a non-physical fermion model. A fermion model which violates the parity symmetry will break the locality. Locality forbids a Hamiltonian that does not commute with fermionic parity symmetry [29–31]. However, as far as one is concerned with spin model this violation is not a problem.

Some spin models, such as the XXZ chain, with arbitrary direction of boundary magnetic fields (ADBMF) have been already exactly solved by the thermodynamic Bethe ansatz method [32–44]. However, calculation of some quantities such as the entanglement entropy seems out of reach at this moment. In this work we take advantage of a method proposed in [45,46], see also [47–49] to transform a generic quantum spin chain; with $H_{\text{bulk}}$ that can be mapped to the Hamiltonian of free fermions; with non uniform magnetic field at the boundaries into a quadratic fermion Hamiltonian. It is done by adding auxiliary spins to the system with coupling to the boundary spins and enlarging the Hilbert space of the original model. Afterward, we would be able to fermionize the spin system via JW transformation and make use of postulation of quadratic fermion Hamiltonians to study the quantities of interest. The quadratic (or bi-linear) form of Hamiltonian in fermionic operators is crucial since the Hamiltonian can be diagonalized exactly and the correlation functions can be reduced to the expectation values of pairs of fermionic operators (Wick theorem [50]). To retrieve the eigenstates of original model, we can use particular projection of new model's eigenstates. In [45], the same method was used to study a fermionic model with linear operator terms, which breaks the parity. In this case, the author couples the auxiliary fermions to every other in the system, and gets a quadratic Hamiltonian.

Starting from the enlarged bi-linear fermionic representation of the Hamiltonian, similar to [45, 47, 49], the degeneracy of (at least) two is expected. This degeneracy results in degen-

erate ground states with opposing parities, which allows us to find a superposition for the ground state with broken parity number. However, for such an eigenstate, we would not be able to use conventional methods to find the RDM, entanglement and so forth. We study the aspects of this state and how it is related to the ground states of the original boundary magnetic field (BMF) Hamiltonian. Also, the comparison of the correlation of this state with those of the parity symmetry states is investigated. Such an investigation enables us to make an ansatz for RDM based on the correlation matrices. This ansatz was first proposed in [49] for a special type of parity broken state and special type of subsystem. Here, we generalize the previous results and provide a proof for consistency of this ansatz. In addition, the RDM has been calculated in different methods. We also show that all of these results can be extended to arbitrary enigenstates of the Hamiltonian (1).

Having the RDM in terms of correlation matrices facilitates the calculation of entanglement. For the parity broken state the behavior of entanglement entropy with respect to parity number is intriguing. In this paper, we mostly deal with a connected subsystem starting from one end of the chain to some point in the middle of the system. An interesting observation is that based on the way one breaks the parity of the ground state entanglement can be minimized or unaffected. Besides, we show that with an adjustment to the Peschel method [2,3,5,51,52], it is possible to get the entanglement of parity broken state in terms of entanglement of parity protected ground states. With these results, we will be able to study the effect of boundary conditions on the entanglement of subsystem, in particular, change in the direction of magnetic field at boundaries on the value of entanglement entropy.

It is worth mentioning that one can treat the enlarged Hamiltonian as a model of interest and study its different properties independent of the quantum spin chain with boundaries. This might be an interesting approach to study a phenomena like spontaneous breaking of parity number symmetry. Having this in mind most of our studies and results go beyond just the quantum spin chains with ADBMF. We find the correlations, reduced density matrix and entanglement for generic eigenstates of the enlarged Hamiltonian. Quite surprisingly, the general features of the results are quite independent of the bulk Hamiltonian.

The remainder of this paper is structured as follows: In section 2 we first summarize the main results of the paper. In section 3, we start by general quadratic Hamiltonian that one can derive after the addition of extra sites. We, provide a general discussion on diagonalization of such a system, and the connection to the eigenstates of our original model, the BMF Hamiltonian. In section 4, we use the result of diagonalization of quadratic Hamiltonian to find the correlation matrices for the generic eigenstates including the most interesting ones, i.e. states with ±1 parity and the one with no parity. Section 5 contains study of the reduced density matrix (RDM). We first present a general formulation of RDM using Berezin integration of Grassmann variables, followed by the RDM in terms of correlation matrices. We address the RDM not only for typical eigenstates of Hamiltonian, but also eigenstates which break the parity. With the results of former sections, we study the entanglement in section 6. Notably, we dig to the behavior of entanglement for a state which defies the parity symmetry. In section 7 we give an interesting physical interpretation of the parity-broken state based on three-part system. This gives a simple way to reproduce some of our results in especial cases. Finally, in section 8, we look at some interesting examples of systems with no parity symmetry. The first being free fermion model with no bulk term, and the second being the $XY$-spin chain with arbitrary boundary magnetic fields.

## 2 Summary of Main Results

Consider the spin chain Hamiltonian

$$H_{\text{SpinChain}} = H_{\text{bulk}} + \vec{B}_1.\vec{S}_1 + \vec{B}_L.\vec{S}_L, \tag{2}$$

where $\vec{S}_1$ and $\vec{S}_L$, are spin operators at the beginning and end of the chain, and $\vec{B}_L, \vec{B}_1$ are arbitrary boundary magnetic fields. We consider here bulk Hamiltonians, i.e. $H_{\text{bulk}}$, that can be mapped to quadratic free fermions via Jordan-Wigner (JW) transformation. The above Hamiltonian does not have quadratic form after JW transformation and a priory is not clear that can be solved exactly. However, using the ancillary spins, $S_0$ and $S_{L+1}$ one can transform the above Hamiltonian to

$$H = H_{\text{bulk}} + B_1^x S_0^x S_1^x + B_1^y S_0^x S_1^y + B_1^z S_1^z + B_L^x S_L^x S_{L+1}^x + B_L^y S_L^y S_{L+1}^x + B_L^z S_L^z \tag{3}$$

The above Hamiltonian after JW transformation has a quadratic form and can be solved exactly. The Hamiltonian (5) is considered with this goal in mind. The eigenstates and eigenvalues of the original Hamiltonian can be found with proper projections. The section 3 of this paper shows how this procedure can be done in its most general form. The idea of using ancillary sites to solve boundary spin chains is already explored before in [45–49], however, in this work we solve the problem in its most general form without restricting to a particular Hamiltonian. The eigenstates, eigenvalues and correlation functions are found in the most general cases in sections 3 and 4.

Interestingly all the eigenstates of the Hamiltonian (3) have at least the degeneracy of two due to the presence of a zero mode. In the fermionic language the vacuum, i.e. $|0\rangle_\eta$, and the state with the zero mode excited, i.e. $\eta_0^\dagger |0\rangle_\eta$, are degenerate. This means one can define

$$|\beta\rangle = \frac{1}{\sqrt{1+|\beta|^2}} \big( |0\rangle_\eta + \beta \eta_0^\dagger |0\rangle_\eta \big); \tag{4}$$

as the most generic ground state of this Hamiltonian[1]. In particular, the ground state of the boundary quantum spin chain can be derived out of the especial cases of $\beta = \pm 1$. Due to its generality in sections 3 and 4 we study some basic properties of this state such as parity and correlation functions. This state has also an interesting interpretation as three part system which we will explore in section 7.

In section 5 we give two different forms for the reduced density matrix of the generic $|\beta\rangle$ state. We first find the reduced density matrix in fermionic coherent basis. The presented form can be used to calculate the Rényi entanglement entropy as it was shown in section 6. We also think that this form is useful to calculate the formation probabilities in the configuration basis which are although interesting for their own sake, we are not going to explore them in this paper. The second method is a generalization of the results of [49] for generic eigenstates of the generic Hamiltonians with arbitrary $\beta$. It is an extension of a method which is based on making an ansatz for the reduced density matrix and then fix the exact form by matching the correlation functions [53]. Our result for $\beta = \pm 1$ gives also the reduced density matrix of the quantum spin chain with arbitrary boundary magnetic fields. Using the results of this section in section 6 we write an exact formula for the entanglement entropy of the $|\beta\rangle$ state which its complexity grows polynomially

---

[1]Note that the same can be done with all the eigenstates. Our results can be extended for the $|\beta\rangle$ states made of each eigenstates.

with the size of the subsystem. As before, this also solves the problem of the calculation of the entanglement entropy of the eigenstates of the Hamiltonian (2).

In section 7 we show that the $|\beta\rangle$ state has a very interesting property. It can be written like a three-part system which helps to understand the entanglement of the ancillary sites with the rest of the system using this simple interpretation. This property makes studying the Hamiltonian (3) and its entanglement content interesting for its own sake independent of the original motivation of solving the quantum spin chain with arbitrary boundary magnetic fields. This has been the main motivation to start with the fermionic version of the enlarged Hamiltonian in section 3.

All of the discussions up to section 8 are independent of the Hamiltonian and also valid for arbitrary eigenstates of the Hamiltonians (2) and (3). In section 8 we give two explicit examples. The first one is a slight generalization of the Hamiltonian (3) without a bulk term. In the first sight this seems oversimplification but interestingly a lot of entanglement properties of the generic Hamiltonian have similar features which was the main motivation for its presentation. To give a non-trivial example we also study the entanglement entropy in the XY chain. The boundary entropy of this model is already studied in [49], however, here we fill a few holes such as the presentation of the exact extra zero modes present in the Ising chain which their form is essential to calculate the entanglement entropy exactly.

## 3 The Hamiltonian

In this section, we introduce a general bi-linear fermionic Hamiltonian which is related to spin chain Hamiltonians with boundary magnetic field (1). We first study the general properties of this model and point out the evident zero mode of the model. In subsection 3.3, we tend to present the diagonalization procedure using the standard methods. Later, in subsection 3.5, we look into the particular eigenstates of the Hamiltonian which does not respect the parity number symmetry. Finally, subsection 3.6 presents a set of selection rules to retrieve the eigenvalues of the boundary magnetic field (BMF) model. Interestingly the structure that unfold is very similar to the ones in [49], see also [47] with a slightly different notation.

### 3.1 General properties

We are going to study the quadratic fermionic Hamiltonian of the form:

$$H = H_0 + H_b \tag{5}$$

where

$$H_0 = \sum_{i,j=1}^{L} \left[ c_i^\dagger A_{ij} c_j + \tfrac{1}{2} c_i^\dagger B_{ij} c_j^\dagger + \frac{1}{2} c_i B_{ji}^* c_j \right] - \frac{1}{2} \text{tr}(\mathbf{A}^*), \tag{6}$$

and

$$H_b = \sum_{j=1}^{L} \left[ \alpha_j^0 (c_0 c_j - c_0^\dagger c_j) + \alpha_j^{L+1} (c_j c_{L+1}^\dagger + c_j c_{L+1}) + \text{H. C.} \right]. \tag{7}$$

As we mentioned in the introduction, $H_b$ part can be related to the boundary terms. Note that the above Hamiltonian is a bit more general than just the extension of a quantum chain with

boundaries because the added extra sites $0$ and $L+1$ are coupled to all the sites. Note also that we are not bounded here to any particular dimension. That means all of the upcoming results, as far as we do not talk about spin chains, are valid in arbitrary dimensions. Here, **A** matrix should be Hermitian and **B** matrix should be anti-symmetric. The Hamiltonian can be written as:

$$H = \frac{1}{2} \begin{pmatrix} \mathbf{c}^\dagger & \mathbf{c} \end{pmatrix} \mathbf{M} \begin{pmatrix} \mathbf{c} \\ \mathbf{c}^\dagger \end{pmatrix}, \tag{8}$$

where $\begin{pmatrix} \mathbf{c}^\dagger & \mathbf{c} \end{pmatrix}$ stands for $\begin{pmatrix} c_0^\dagger & \cdots & c_{L+1}^\dagger & c_0 & \cdots & c_{L+1} \end{pmatrix}$. The matrix **M** in (8) should look like below.

$$\mathbf{M} = \left( \begin{array}{ccccc|ccccc} 0 & -\alpha_1^0 & \cdots & -\alpha_L^0 & 0 & 0 & -\alpha_1^{0*} & \cdots & -\alpha_L^{0*} & 0 \\ -\alpha_1^{0*} & & & & -\alpha_1^{L+1*} & \alpha_1^{0*} & & & & -\alpha_1^{L+1*} \\ \vdots & & \mathbf{A} & & \vdots & \vdots & & \mathbf{B} & & \vdots \\ -\alpha_L^{0*} & & & & -\alpha_L^{L+1*} & \alpha_L^{0*} & & & & -\alpha_L^{L+1*} \\ 0 & -\alpha_1^{L+1} & \cdots & -\alpha_L^{L+1} & 0 & 0 & \alpha_1^{L+1*} & \cdots & \alpha_L^{L+1*} & 0 \\ \hline 0 & \alpha_1^0 & \cdots & \alpha_L^0 & 0 & 0 & \alpha_1^{0*} & \cdots & \alpha_L^{0*} & 0 \\ -\alpha_1^0 & & & & \alpha_1^{L+1} & \alpha_1^0 & & & & \alpha_1^{L+1} \\ \vdots & & -\mathbf{B}^* & & \vdots & \vdots & & -\mathbf{A}^* & & \vdots \\ -\alpha_L^0 & & & & \alpha_L^{L+1} & \alpha_L^0 & & & & \alpha_L^{L+1} \\ 0 & -\alpha_1^{L+1} & \cdots & -\alpha_L^{L+1} & 0 & 0 & \alpha_1^{L+1*} & \cdots & \alpha_L^{L+1*} & 0 \end{array} \right). \tag{9}$$

For later use, we define the following spin operators at positions $0$ and $L+1$,

$$\sigma_0^x = c_0 + c_0^\dagger \quad \text{and} \quad \sigma_{L+1}^x = (1 - 2c_0 c_0^\dagger) \prod_{l=1}^{L} (1 - 2c_l^\dagger c_l)(c_{L+1} + c_{L+1}^\dagger). \tag{10}$$

These operators commute with each other and the Hamiltonian (5); i.e.

$$\left[ H, \sigma_{L+1}^x \right] = \left[ H, \sigma_0^x \right] = \left[ \sigma_0^x, \sigma_{L+1}^x \right] = 0. \tag{11}$$

The above relations are important in upcoming sections.

### 3.2 Zero mode eigenstates

The **M** matrix in (8) has at least two eigenvectors with zero eigenvalues. These eigenvectors correspond to the modes with zero energy. Later on in this paper, we present a formulation (section 3.3) which simplifies the effort to find the correlations (section 4), reduced density matrix (section 6) and other properties of the system such as entanglement. To use such a formulation, one needs to identify and write the zero eigenvectors in a correct form (see the following subsection).

Due to the form of the **M** matrix (9), we expect to have two zero modes. These zero modes do

not depend on the parameters and interactions of the system. They can be written as:

$$|u_0^1\rangle = \begin{pmatrix} \sqrt{a}e^{i\theta_1} \\ 0 \\ \vdots \\ 0 \\ \sqrt{\frac{1}{2}-a}e^{i\phi_1} \\ \sqrt{a}e^{i\theta_1} \\ 0 \\ \vdots \\ 0 \\ -\sqrt{\frac{1}{2}-a}e^{i\phi_1} \end{pmatrix}, \qquad |u_0^2\rangle = \begin{pmatrix} \sqrt{\frac{1}{2}-a}e^{i\theta_2} \\ 0 \\ \vdots \\ 0 \\ -\sqrt{a}e^{i\phi_2} \\ \sqrt{\frac{1}{2}-a}e^{i\theta_2} \\ 0 \\ \vdots \\ 0 \\ \sqrt{a}e^{i\phi_2} \end{pmatrix}. \tag{12}$$

The orthogonality condition of these states requires the following equality to hold for the parameters

$$\theta_1 - \theta_2 = \phi_1 - \phi_2. \tag{13}$$

The above zero modes are independent of the parameters of the Hamiltonian. In the rest, for the sake of simplicity, we take all the angles $(\theta_1, \theta_2, \phi_1, \phi_2)$ to be zero and put $a = \frac{1}{4}$. Taking other values does not change the upcoming results. Depending on the values of coupling parameters there could arise more zero modes in spectrum of $\mathbf{M}$ matrix[2].

## 3.3 Diagonalization

In this subsection, we want to briefly review the diagonalization and find the eigenstates of the Hamiltonian. First, we introduce a new operator $\mathbf{J}$ which acts on the eigenstates of the matrix $\mathbf{M}$ as

$$\mathbf{J}\begin{pmatrix} u \\ v \end{pmatrix} = \begin{pmatrix} v^* \\ u^* \end{pmatrix}. \tag{14}$$

It is easy to show that the operator $\mathbf{J}$ anticommutes with the matrix $\mathbf{M}$, i.e. $\{\mathbf{M}, \mathbf{J}\} = 0$. As a consequence, if $|V\rangle$ is an eigenvector of matrix $\mathbf{M}$ with eigenvalue $\lambda$ then the vector $\mathbf{J}|V\rangle$ is also an eigenvector with eigenvalue $-\lambda$. It means that finding the eigenstates corresponding to positive eigenvalues is enough; we can get the eigenstates for negative eigenvalues by acting on the eigenstates of the positive eigenvalues with operator $\mathbf{J}$ [54].

As a result of operator $\mathbf{J}$, the Hamiltonian (8) can be diagonalized in the following form:

$$\mathbf{H} = \frac{1}{2}\begin{pmatrix} \mathbf{c}^\dagger & \mathbf{c} \end{pmatrix} \mathbf{U}^\dagger \mathbf{U} \mathbf{M} \mathbf{U}^\dagger \mathbf{U} \begin{pmatrix} \mathbf{c} \\ \mathbf{c}^\dagger \end{pmatrix} = \frac{1}{2}\begin{pmatrix} \eta^\dagger & \eta \end{pmatrix} \begin{pmatrix} \Lambda & 0 \\ 0 & -\Lambda \end{pmatrix} \begin{pmatrix} \eta \\ \eta^\dagger \end{pmatrix}; \tag{15}$$

The matrix $\Lambda$ is a diagonal matrix with non-negative entries and matrix $\mathbf{U}$ has the block from:

$$\mathbf{U} = \begin{pmatrix} g & h \\ h^* & g^* \end{pmatrix}. \tag{16}$$

---

[2]See section 8.2.

The Hamiltonian can also be written with respect to new fermionic operators (Bogoliubov fermions) as

$$\mathbf{H} = \sum_k \lambda_k \eta_k^\dagger \eta_k - \frac{1}{2}\mathrm{tr}(\mathbf{\Lambda}), \tag{17}$$

where

$$\begin{pmatrix} \boldsymbol{\eta} \\ \boldsymbol{\eta}^\dagger \end{pmatrix} = \mathbf{U} \begin{pmatrix} \mathbf{c} \\ \mathbf{c}^\dagger \end{pmatrix}. \tag{18}$$

From (14) and special form of $U$ matrix, we get the following constrains on the elements of $\mathbf{g}$ and $\mathbf{h}$ matrices:

$$g_{k,L+1} = h_{k,L+1} \quad \text{and} \quad g_{k,0} = -h_{k,0} \quad \text{for} \quad k \neq 0 \tag{19}$$

To conclude the diagonalization part, we write the explicit expression of $\eta$ operators:

$$\eta_k = \sum_{j=0}^{L+1} g_{kj}c_j + h_{kj}c_j^\dagger = g_{k0}(c_0 - c_0^\dagger) + \sum_{j=1}^{L}(g_{kj}c_j + h_{kj}c_j^\dagger) + g_{kL+1}(c_{L+1} + c_{L+1}^\dagger); \quad k \neq 0, \tag{20}$$

$$\eta_0 = \frac{1}{2}\left(c_0 + c_{L+1} + c_0^\dagger - c_{L+1}^\dagger\right). \tag{21}$$

From above, we can also write the $c$-fermions in terms of $\eta$-operators as:

$$c_k = \sum_{j=1}^{L+1}(g_{j,k}^*\eta_j + h_{j,k}\eta_j^\dagger); \quad k \neq 0, L+1, \tag{22}$$

$$c_0 = \frac{1}{2}\eta_0 + \frac{1}{2}\eta_0^\dagger + \sum_{j=1}^{L+1}(g_{j,0}^*\eta_j - g_{j,0}\eta_j^\dagger), \tag{23}$$

$$c_{L+1} = \frac{1}{2}\eta_0 - \frac{1}{2}\eta_0^\dagger + \sum_{j=1}^{L+1}(g_{j,L+1}^*\eta_j + g_{j,L+1}\eta_j^\dagger). \tag{24}$$

This final result means that the following commutation (anticommutation) relations hold (for $k \neq 0$)

$$\left[\sigma_{L+1}^x, \eta_k\right] = \left[\sigma_{L+1}^x, \eta_k^\dagger\right] = 0, \tag{25}$$

$$\left\{\sigma_0^x, \eta_k\right\} = \left\{\sigma_0^x, \eta_k^\dagger\right\} = 0. \tag{26}$$

Anticommutations (26) will be useful in subsection 3.6 for sectorization of eigenstates of the Hamiltonian (5).

## 3.4  Eigenstates in configuration basis

The vacuum state $|0\rangle_\eta$ is the state which is annihilated by the action of all $\eta_k$ operators,

$$\eta_k|0\rangle_\eta = 0, \quad \forall\, k. \tag{27}$$

One can write the $|0\rangle_\eta$ as a superposition of configurations of the $c$-fermions [2]. It is called the configurational basis of such a state. For even size, if the parity of $|0\rangle_\eta$ is $+1$ then we have:

$$|0\rangle_\eta = \left(\det[\mathbf{I} + \mathbf{R}^\dagger\mathbf{R}]\right)^{-\frac{1}{4}} e^{\frac{1}{2}\sum_{i,j}R_{ij}c_i^\dagger c_j^\dagger}|0\rangle_c \tag{28}$$

where $c_j |0\rangle_c = 0$ for all $j$, and the $\mathbf{R}$ matrix is an antisymmetric matrix defined as $\mathbf{g}.\mathbf{R} + \mathbf{h} = 0$. The (28) works only for the case where parity of the $|0\rangle_\eta$ is $+1$, and/or the matrix $\mathbf{g}$ is invertible. Otherwise, to use the above equation, one needs to do a canonical transformation to change the parity of the vacuum for $\eta$-fermions, and/or to make the $\mathbf{g}$ invertible [55].

Such a canonical transformation could be the following change in the creation and annihilation operators:

$$\begin{cases} c_j \rightarrow \tilde{c}_j^\dagger, \\ c_j^\dagger \rightarrow \tilde{c}_j. \end{cases} \tag{29}$$

We call this transformation *Tilda transformation*. If we do such a change for even number of the indices, the parity would not change (in the case where $\mathbf{g}$ matrix is not invertible). For odd number of sites, it will change the parity of the state $|0\rangle_\eta$. As an example, when the vacuum state has parity $-1$, we do the tilda transformation

$$c_j \rightarrow \tilde{c}_j^\dagger \qquad \text{and} \qquad c_j^\dagger \rightarrow \tilde{c}_j \tag{30}$$

for only one specific index $j$. Regardless of selected $j$, the entanglement properties of the vacuum would not change. If we choose $j = 0$ then this transformation can be written in operator form as $\widetilde{\mathcal{P}} = e^{i\frac{\pi}{2}\sigma_0^x}$.

As a matter of fact, any eigenstate of Hamiltonian (5) (with parity $+1$) can be written in configurational basis in an exponential form. Excited states are created by exciting different modes on the vacuum (27) as:

$$|\psi\rangle = |k_1, k_2, \cdots, k_N\rangle = \prod_{k_j \in \mathbb{E}} \eta_{k_j}^\dagger |0\rangle; \qquad E_\psi = \sum_{k_j \in \mathbb{E}} \lambda_{k_j} - \frac{1}{2}\text{tr}(\mathbf{\Lambda}), \tag{31}$$

where the set $\mathbb{E}$ can be any subset of indices from 0 to $L+1$. We denote the set of indices of excited modes as $\mathbb{E}$ and the set of indices which are not excited as $\bar{\mathbb{E}}$ ($\bar{\mathbb{E}} \cup \mathbb{E} = \{0, 1, \cdots, L+1\}$). Assume that we can write the following excited state in the configurational form as:

$$|\psi\rangle = C^\psi e^{\frac{1}{2}\sum_{i,j} R_{ij}^\psi c_i^\dagger c_j^\dagger} |0\rangle_c, \tag{32}$$

where $C^\psi = (\det[\mathbf{I} + \mathbf{R}^{\psi\dagger}\mathbf{R}^\psi])^{-\frac{1}{4}}$. For this excited state we have:

$$\begin{aligned} \eta_{k_j} |\psi\rangle &= 0; & k_j &\in \bar{\mathbb{E}}, \\ \eta_{k_n}^\dagger |\psi\rangle &= 0; & k_n &\in \mathbb{E}. \end{aligned} \tag{33}$$

Therefore, from the above equation, we get:

$$g_{k_j l} R_{lm}^\psi + h_{k_j m} = 0; \qquad k_j \in \bar{\mathbb{E}}, \tag{34}$$

$$h_{k_n l}^* R_{lm}^\psi + g_{k_n m}^* = 0; \qquad k_n \in \mathbb{E}. \tag{35}$$

The generalized formula for the $\mathbf{R}^\psi$ would be:

$$\mathfrak{g}R^\psi + \mathfrak{h} = 0, \tag{36}$$

where $\mathfrak{g}$ and $\mathfrak{h}$ are generalized versions of $\mathbf{g}$ and $\mathbf{h}$ given by

$$\mathfrak{g}_{nm} = \begin{cases} g_{nm} & if\ n \in \bar{\mathbb{E}} \\ h_{nm}^* & if\ n \in \mathbb{E} \end{cases} \qquad \mathfrak{h}_{nm} = \begin{cases} h_{nm} & if\ n \in \bar{\mathbb{E}} \\ g_{nm}^* & if\ n \in \mathbb{E} \end{cases} \tag{37}$$

Note that same as the case of vacuum state, one should make sure that the $\mathfrak{g}$ has an inverse, and state $|\psi\rangle$ has the right parity. If not one should use the canonical (Tilda) transformation to be able to write a typical eigenstate in configuration basis. The (32) gives an excited eigenstate in the configuration basis which is advantageous in the study of entanglement in later sections.

## 3.5 Parity broken state

The quadratic Hamiltonian (5) commutes with the parity operator defined as $\mathbf{P} = (-1)^{\hat{N}}$ where $\hat{N} = \sum_{l=0}^{L+1} c_l^\dagger c_l$ is the fermion number operator. This means that the eigenstates of the Hamiltonian have fixed parity $P = \pm 1$. One can have eigenstates that do not respect the parity and these types of states are very interesting to study. For instance, these types of states are related to the ground state of spin systems with boundary magnetic fields which one example is given in section 8.2.

We define a parity broken state ($\beta$-defected parity state) as:

$$|\beta\rangle = \frac{1}{\sqrt{1+|\beta|^2}}\left(|0\rangle_\eta + \beta \eta_0^\dagger |0\rangle_\eta\right); \tag{38}$$

where $\beta$ can be a complex number. The expectation value of parity for such a state is given by

$$P(\beta) = \langle\beta|\mathbf{P}|\beta\rangle = \begin{cases} \frac{|\beta|^2-1}{|\beta|^2+1} & P_0 = -1, \\[2mm] \frac{1-|\beta|^2}{1+|\beta|^2} & P_0 = +1, \end{cases} \tag{39}$$

where $P_0$ is the parity of vacuum state. The first excited state after the vacuum is created by $\eta_0^\dagger |0\rangle$ which inevitably has the same energy as the vacuum, while this excited state has parity $-P_0$. In fact $\beta$ can be considered as a parameter which can be tuned to break the parity.

We define the state $|G_\pm\rangle$ by taking $\beta = \pm 1$ as:

$$|G_\pm\rangle = \frac{1}{\sqrt{2}}\left(|0\rangle_\eta \pm \eta_0^\dagger |0\rangle_\eta\right). \tag{40}$$

These states are especial cases of $\beta$-broken parity states and have interesting properties which cares for special attention. As an example, these states are related to the ground state of Hamiltonian with boundary magnetic field (see 1). Other useful properties of these two states are

$$\sigma_0^x |G_\pm\rangle = \pm |G_\pm\rangle \quad \text{and} \quad \sigma_{L+1}^x |G_\pm\rangle = \delta_\pm |G_\pm\rangle. \tag{41}$$

The $\delta_\pm = \pm$ can be calculated with respect to the expectation values of Majorana fermions [49]. For instance, one can write:

$$\delta_+ = (i)^{L+1}\text{Pf}[\mathcal{D}], \tag{42}$$

where

$$
\mathcal{D} = \begin{pmatrix}
0 & \langle \bar{\gamma}_0 \gamma_1 \rangle & \langle \bar{\gamma}_0 \bar{\gamma}_1 \rangle & \cdots & \langle \bar{\gamma}_0 \gamma_{L+1} \rangle \\
\langle \gamma_1 \bar{\gamma}_0 \rangle & 0 & \langle \gamma_1 \bar{\gamma}_1 \rangle & \cdots & \langle \gamma_1 \gamma_{L+1} \rangle \\
\langle \bar{\gamma}_1 \bar{\gamma}_0 \rangle & \langle \bar{\gamma}_1 \gamma_1 \rangle & 0 & \cdots & \langle \bar{\gamma}_1 \gamma_{L+1} \rangle \\
\vdots & \vdots & \vdots & & \vdots \\
\langle \gamma_{L+1} \bar{\gamma}_0 \rangle & \langle \gamma_{L+1} \gamma_1 \rangle & \langle \gamma_{L+1} \bar{\gamma}_1 \rangle & \cdots & 0
\end{pmatrix}.
\tag{43}
$$

in above, Pf$[\mathcal{D}]$ is the Pfaffian of the matrix $\mathcal{D}$, and Majorana fermions are defined as $\gamma_j = c_j^\dagger + c_j$ and $\bar{\gamma}_j = i(c_j^\dagger - c_j)$. In (43), the $\langle \cdots \rangle$ stands for the expectation value with respect to the vacuum state of the $\eta$-operators ($|0\rangle$).

## 3.6 Eigenstates of boundary magnetic field model

Since the eigenstates of boundary magnetic field model is related to those of Hamiltonian (5), we are going to present selection rules to get the desired eigenstates. In fact, Hilbert space of Hamiltonian (5) is 4 times bigger than BMF Hamiltonian.

The Hilbert space of Hamiltonian (5) can be divided into 4 sub-spaces. Each sub-space can be identified using eigenvalues of operators $\sigma_0^x$ and $\sigma_{L+1}^x$ acting on states $|G_\pm\rangle$. We can make the following argument: consider $\delta_+ = 1$, which means $|G_+\rangle$ belongs to the sector marked by the pair $(\langle G_\pm | \sigma_0^x | G_\pm \rangle, \langle G_\pm | \sigma_{L+1}^x | G_\pm \rangle)$. Then due to commutation (anticommutation) relations (25) and (26) all the states

$$
\prod_{j=1}^{n} \eta_{k_j}^\dagger |G_+\rangle, \qquad n \text{ is even.}
\tag{44}
$$

also belong to $(+, +)$. Note that in the above expression $0 < k_j < k_{j+1} < L + 1$ which means that the dimension of $(+, +)$ sub-space is $2^L$. Next, in this case ($\delta_+ = +1$), the state $|G_-\rangle$ and the following states belong to $(-, -)$ sector,

$$
\prod_{j=1}^{n} \eta_{k_j}^\dagger |G_-\rangle, \qquad n \text{ is even.}
\tag{45}
$$

For the two other sectors we have:

$$
\begin{aligned}
\prod_{j=1}^{n} \eta_{k_j}^\dagger |G_+\rangle, & \qquad n \text{ is odd.} & (-, +), \\
\prod_{j=1}^{n} \eta_{k_j}^\dagger |G_-\rangle, & \qquad n \text{ is odd.} & (+, -),
\end{aligned}
\tag{46}
$$

In the case of $\delta_+ = -1$, we have the following eigenstates for each sector,

$$\prod_{j=0}^{n} \eta_{k_j}^\dagger |G_+\rangle, \qquad n \text{ is even}, \qquad (+,-), \tag{47}$$

$$\prod_{j=0}^{n} \eta_{k_j}^\dagger |G_-\rangle, \qquad n \text{ is even}, \qquad (-,+), \tag{48}$$

$$\prod_{j=1}^{n} \eta_{k_j}^\dagger |G_+\rangle, \qquad n \text{ is odd}, \qquad (-,-), \tag{49}$$

$$\prod_{j=1}^{n} \eta_{k_j}^\dagger |G_-\rangle, \qquad n \text{ is odd}, \qquad (+,+). \tag{50}$$

As an example in the $(+,+)$ sector, we can write:

$$|\phi_k\rangle = |+\rangle_0 \otimes |\varphi_k\rangle \otimes |+\rangle_{L+1}, \tag{51}$$

where $|\phi_k\rangle$ is an eigenstate of Hamiltonian (5), $|+\rangle_{0,L+1}$ are eigenstates of $\sigma^x$ at the position 0 and $L+1$. The $|\varphi_k\rangle$ is an eigenstate of the BMF model. Knowing $|\phi_k\rangle$, we can obtain the $|\varphi_k\rangle$.

The above argument means that to know the sector $(+,+)$ we need to figure out the value of $\delta_+$. The ground state of the Hamiltonian with boundary magnetic field is going to be one of the following two states:

$$|G_+\rangle \qquad \delta_+ = +1, \tag{52}$$

$$\eta_{min}^\dagger |G_-\rangle \qquad \delta_+ = -1. \tag{53}$$

It is interesting to mention that exist transformations which change the sign of boundary couplings in the Hamiltonian (5). Such transformations could be $T_{b_L} = \sigma_{L+1}^z$ and $T_{b_1} = \sigma_0^z$ where they change the sign of $x$ and $y$ component of boundary couplings ($\vec{b}_1$, $\vec{b}_L$ or equivalently $\alpha_1^0$, $\alpha_L^{L+1}$), without affecting the energy spectrum of the Hamiltonian:

$$T_{b_1}^\dagger T_{b_L}^\dagger H(b_{1,x}, b_{1,y}, b_{1,z}, b_{L,x}, b_{L,y}, b_{L,z}) T_{b_L} T_{b_1} = H(-b_{1,x}, -b_{1,y}, b_{1,z}, -b_{L,x}, -b_{L,y}, b_{L,z}). \tag{54}$$

As a consequence, if the eigenstates of the BMF Hamiltonian (1) is found in one of the sectors of the Hilbert space of the Hamiltonian (5), then other sectors are related to the eigenstates of BMF Hamiltonians with different signs of boundary couplings. For example, if a typical eigenstate of the BMF Hamiltonian like $|\varphi_k\rangle$, is in the $(+,+)$ sector, then from (51) we have $|+\rangle_0 \otimes |\varphi_k\rangle \otimes |+\rangle_{L+1}$ for the eigenstates of Hamiltonian (5). The action of $T_{b_1} T_{b_L}$ on a state like $|\phi_k\rangle$ would be

$$T_{b_1} T_{b_L} |\phi_k\rangle = |-\rangle_0 \otimes |\varphi_k\rangle \otimes |-\rangle_{L+1}, \tag{55}$$

which is equal to changing the sign of boundary couplings. Therefore, $|\varphi_k\rangle$ would be an eigenstate of BMF Hamiltonian with boundary couplings: $-b_{1,x}, -b_{1,y}, -b_{L,x}$ and $-b_{L,y}$. Equivalently, spectrum of of BMFH with negative couplings at boundary can be found in the $(-,-)$ sector of Hamiltonian (5).

# 4 Correlation functions

In this section, we would like to calculate the correlation matrix for different eigenstates of the system. It is more convenient to calculate the correlation matrices for Majorana fermions. For instance, one can use Majorana fermion correlations to calculate the entanglement in the system (for particular eigenstates). We introduce Majorana fermions as $\gamma_i = c_i + c_i^\dagger$ and $\bar{\gamma}_i = i(c_i^\dagger - c_i)$. We symbolize the correlation matrices as:

$$\langle \bar{\gamma}_j \gamma_k \rangle = i G_{jk}, \qquad \langle \gamma_j \gamma_k \rangle = K_{jk}, \qquad \langle \bar{\gamma}_j \bar{\gamma}_k \rangle = \bar{K}_{jk}. \tag{56}$$

It is useful to write the later two point correlation in a block matrix form denoted by $\boldsymbol{\Gamma}$ as:

$$\boldsymbol{\Gamma} = \begin{pmatrix} \boldsymbol{K} - \mathbf{I} & -i\boldsymbol{G}^T \\ i\boldsymbol{G} & \bar{\boldsymbol{K}} - \mathbf{I} \end{pmatrix}. \tag{57}$$

One can easily find all the different elements of the $\boldsymbol{\Gamma}$ matrix. It is possible to write $\mathbf{G}$, $\mathbf{K}$ and $\bar{\mathbf{K}}$ in terms of correlation matrices of c-fermions

$$\begin{aligned} \mathbf{K} &= \mathbf{F}^\dagger + \mathbf{F} + \mathbf{C} - \mathbf{C}^T + \mathbb{I}, \\ \bar{\mathbf{K}} &= -\mathbf{F}^\dagger - \mathbf{F} + \mathbf{C} - \mathbf{C}^T + \mathbb{I}, \\ \mathbf{G} &= -\mathbf{F}^\dagger + \mathbf{F} + \mathbf{C} + \mathbf{C}^T - \mathbb{I}, \end{aligned} \tag{58}$$

where $C_{ij} = \langle c_i^\dagger c_j \rangle$ and $F_{ij} = \langle c_i^\dagger c_j^\dagger \rangle$. The $\mathbf{C}$ is a Hermitian matrix and $\mathbf{F}$ is antisymmetric. Therefore, $\mathbf{K}$ and $\bar{\mathbf{K}}$ are Hermitian, and we can conclude that $\mathbf{G}$ is real. Knowing these properties, we can prove that the $\boldsymbol{\Gamma}$ correlation matrix is Hermitian too. All the analysis so far are valid for arbitrary eigenstates of the Hamiltonian (5). In the rest, the correlations for vacuum and zero mode excited eigenstate (ZME state or $\eta_0^\dagger |0\rangle_\eta$) will be presented in details. The calculation of correlations of excited quasi-particle eigenstates is presented in appendix A.

## 4.1 Correlations for vacuum state

Using the notation introduced in (15), we calculate the correlations for the vacuum of $\eta$'s. In this case, we can write $\mathbf{C}^0 = \mathbf{h}^\dagger.\mathbf{h}$ and $\mathbf{F}^0 = \mathbf{h}^\dagger.\mathbf{g}$, where superscript zero stands for the expectation values calculated in the vacuum state. Putting these relations in (58), we get

$$\begin{aligned} \mathbf{K}^0 &= (\mathbf{h}^\dagger + \mathbf{g}^\dagger).(\mathbf{h} + \mathbf{g}), \\ \bar{\mathbf{K}}^0 &= (\mathbf{h}^\dagger - \mathbf{g}^\dagger).(\mathbf{h} - \mathbf{g}), \\ \mathbf{G}^0 &= (\mathbf{h}^\dagger - \mathbf{g}^\dagger).(\mathbf{h} + \mathbf{g}). \end{aligned} \tag{59}$$

Therefore, for the vacuum of $\eta$-operators, we can find the correlations in terms of the elements of $\mathbf{U}$ matrix, which means correlation matrix calculations are straightforward. The above-mentioned

correlation matrices have the form ($i, j \neq 0, L+1$):

$$
\mathbf{C}^0 = \left(
\begin{array}{c|c|c}
\frac{1}{2} & -\sum_{k=1}^{k=L+1} g_{k,0}^* h_{k,j} & \frac{-1}{4} - \sum_{k=1}^{k=L+1} g_{k,0}^* g_{k,L+1} \\
\hline
-\sum_{k=1}^{k=L+1} h_{k,j}^* g_{k,0} & (\mathbf{h}^\dagger . \mathbf{h})_{i,j} & \sum_{k=1}^{k=L+1} h_{k,j}^* g_{k,L+1} \\
\hline
\frac{-1}{4} - \sum_{k=1}^{k=L+1} g_{k,L+1}^* g_{k,0} & \sum_{k=1}^{k=L+1} g_{k,L+1}^* h_{k,j} & \frac{1}{2}
\end{array}
\right) \tag{60}
$$

$$
\mathbf{F}^0 = \left(
\begin{array}{c|c|c}
0 & -\sum_{k=1}^{k=L+1} g_{k,0}^* g_{k,j} & \frac{1}{4} - \sum_{k=1}^{k=L+1} g_{k,0}^* g_{k,L+1} \\
\hline
\sum_{k=1}^{k=L+1} g_{k,j} g_{k,0}^* & (\mathbf{h}^\dagger . \mathbf{g})_{i,j} & -\sum_{k=1}^{k=L+1} g_{k,j} g_{k,L+1}^* \\
\hline
-\frac{1}{4} + \sum_{k=1}^{k=L+1} g_{k,L+1}^* g_{k,0} & \sum_{k=1}^{k=L+1} g_{k,L+1}^* g_{k,j} & 0
\end{array}
\right). \tag{61}
$$

Using the relation (58), for the correlation of Majorana fermions, we can write:

$$
\mathbf{K}^0 = \left(
\begin{array}{c|c|c}
1 & 0 & 0 \\
\hline
0 & (\mathbf{h}^\dagger + \mathbf{g}^\dagger).(\mathbf{h}+\mathbf{g})_{i,j} & 2\sum_{k=1}^{k=L+1}(h_{k,j}^* + g_{k,j}^*) g_{k,L+1} \\
\hline
0 & 2\sum_{k=1}^{k=L+1} g_{k,L+1}^*(h_{k,j} + g_{k,j}) & 1
\end{array}
\right), \tag{62}
$$

$$
\bar{\mathbf{K}}^0 = \left(
\begin{array}{c|c|c}
1 & -2\sum_{k=1}^{k=L+1} g_{k,0}^*(h_{k,j} - g_{k,j}) & 0 \\
\hline
-2\sum_{k=1}^{k=L+1}(h_{k,j}^* - g_{k,j}^*) g_{k,0} & (\mathbf{h}^\dagger - \mathbf{g}^\dagger).(\mathbf{h}-\mathbf{g})_{i,j} & 0 \\
\hline
0 & 0 & 1
\end{array}
\right), \tag{63}
$$

$$
\mathbf{G}^0 = \left(
\begin{array}{c|c|c}
0 & -2\sum_{k=1}^{k=L+1} g_{k,0}^*(h_{k,j} + g_{k,j}) & -4\sum_{k=1}^{k=L+1} g_{k,0}^* g_{k,L+1} \\
\hline
0 & (\mathbf{h}^\dagger - \mathbf{g}^\dagger).(\mathbf{h}+\mathbf{g})_{i,j} & 2\sum_{k=1}^{k=L+1}(h_{k,j}^* - g_{k,j}^*) g_{k,L+1} \\
\hline
-1 & 0 & 0
\end{array}
\right) \tag{64}
$$

To calculate the higher point correlation functions, one can use the Wick theorem, which is computationally favorable.

## 4.2 Correlations for ZME state

From now on (for the sake of simplicity), we are going to indicate the ZME state by $|\emptyset\rangle = \eta_0^\dagger |0\rangle$. This state is degenerate with the vacuum, which is crucial for later studies. For this eigenstate, we

get ($i, j \neq 0, L+1$):

$$\mathbf{C}^{\emptyset} = \begin{pmatrix} \frac{1}{2} & -\sum_{k=1}^{k=L+1} g_{k,0}^* h_{k,j} & \frac{1}{4} - \sum_{k=1}^{k=L+1} g_{k,0}^* g_{k,L+1} \\ -\sum_{k=1}^{k=L+1} h_{k,j}^* g_{k,0} & (\mathbf{h}^\dagger.\mathbf{h})_{i,j} & \sum_{k=1}^{k=L+1} h_{k,j}^* g_{k,L+1} \\ \frac{1}{4} - \sum_{k=1}^{k=L+1} g_{k,L+1}^* g_{k,0} & \sum_{k=1}^{k=L+1} g_{k,L+1}^* h_{k,j} & \frac{1}{2} \end{pmatrix} \tag{65}$$

$$\mathbf{F}^{\emptyset} = \begin{pmatrix} 0 & -\sum_{k=1}^{k=L+1} g_{k,0}^* g_{k,j} & \frac{-1}{4} - \sum_{k=1}^{k=L+1} g_{k,0}^* g_{k,L+1} \\ \sum_{k=1}^{k=L+1} g_{k,j} g_{k,0}^* & (\mathbf{h}^\dagger.\mathbf{g})_{i,j} & -\sum_{k=1}^{k=L+1} g_{k,j} g_{k,L+1}^* \\ \frac{1}{4} + \sum_{k=1}^{k=L+1} g_{k,L+1}^* g_{k,0} & \sum_{k=1}^{k=L+1} g_{k,L+1}^* g_{k,j} & 0 \end{pmatrix}. \tag{66}$$

As a result, the correlation matrices $\mathbf{C}$ and $\mathbf{F}$ are only different in only two elements from state $|0\rangle$ to the state $|\emptyset\rangle$. It can be observed that the correlation matrices $\mathbf{K}$, $\bar{\mathbf{K}}$ does not change from state $|0\rangle$ to $|\emptyset\rangle$. Form of the Majorana correlation matrices are presented below as:

$$\mathbf{K}^{\emptyset} = \mathbf{K}^0, \qquad \bar{\mathbf{K}}^{\emptyset} = \bar{\mathbf{K}}^0, \tag{67}$$

$$\mathbf{G}^{\emptyset} = \begin{pmatrix} 0 & -2\sum_{k=1}^{k=L+1} g_{k,0}^*(h_{k,j} + g_{k,j}) & -4\sum_{k=1}^{k=L+1} g_{k,0}^* g_{k,L+1} \\ 0 & (\mathbf{h}^\dagger - \mathbf{g}^\dagger).(\mathbf{h} + \mathbf{g})_{i,j} & 2\sum_{k=1}^{k=L+1}(h_{k,j}^* - g_{k,j}^*)g_{k,L+1} \\ 1 & 0 & 0 \end{pmatrix}. \tag{68}$$

Similar to the result of previous subsection, for higher point functions, one make use of the Wick theorem to calculate the quantity of interest.

## 4.3 Correlations for the general parity broken state

In this subsection we study the correlation function of general states that break the parity, such as $|\beta\rangle$ defined in (38). Calculating correlations (or any expectation value) with respect to the state $|\beta\rangle$, is not as trivial as the calculations for eigenstates, since we are not able to use the Wick theorem. However, there could be many subtleties when we come across quantities which are evaluated with respect to $|\beta\rangle$.

For instance, such a subtlety could be calculation of one point function with respect to the $|\beta\rangle$:

$$\langle\beta|c_j|\beta\rangle = \frac{\text{Re}[\beta]}{1+|\beta|^2}\delta_{j,0} + \frac{i\,\text{Im}[\beta]}{1+|\beta|^2}\delta_{j,L+1} \tag{69}$$

The above means that one point correlation functions are not necessarily zero for the state $|\beta\rangle$. In general, we can say that $\langle\beta|\hat{\mathcal{O}}|\beta\rangle$ is not necessarily zero, if operator $\hat{\mathcal{O}}$ has odd number of fermionic operators. Nonetheless, if $\hat{\mathcal{O}}$ does not depend on $c_0$, $c_0^\dagger$, $c_{L+1}$ and $c_{L+1}^\dagger$ then we can

write $\langle\beta|\hat{\mathcal{O}}|\beta\rangle = \langle 0|\hat{\mathcal{O}}|0\rangle$. With this condition, we can assume that the state $|\beta\rangle$ obeys the Wick theorem. In the following, we denote the correlation matrices by superscript $\beta$ for the state $|\beta\rangle$. These correlations have the form:

$$\mathbf{C}^{\beta} = \left(\begin{array}{c|c|c} \frac{1}{2} & -\sum\limits_{k=1}^{k=L+1} g_{k,0}^{*} h_{k,j} & \frac{-1}{4}\frac{1-|\beta|^2}{1+|\beta|^2} - \sum\limits_{k=1}^{k=L+1} g_{k,0}^{*} g_{k,L+1} \\ \hline -\sum\limits_{k=1}^{k=L+1} h_{k,j}^{*} g_{k,0} & (\mathbf{h}^{\dagger}.\mathbf{h})_{i,j} & \sum\limits_{k=1}^{k=L+1} h_{k,j}^{*} g_{k,L+1} \\ \hline \frac{-1}{4}\frac{1-|\beta|^2}{1+|\beta|^2} - \sum\limits_{k=1}^{k=L+1} g_{k,L+1}^{*} g_{k,0} & \sum\limits_{k=1}^{k=L+1} g_{k,L+1}^{*} h_{k,j} & \frac{1}{2} \end{array}\right), \quad (70)$$

$$\mathbf{F}^{\beta} = \left(\begin{array}{c|c|c} 0 & -\sum\limits_{k=1}^{k=L+1} g_{k,0}^{*} g_{k,j} & \frac{1}{4}\frac{1-|\beta|^2}{1+|\beta|^2} - \sum\limits_{k=1}^{k=L+1} g_{k,0}^{*} g_{k,L+1} \\ \hline \sum\limits_{k=1}^{k=L+1} g_{k,j} g_{k,0}^{*} & (\mathbf{h}^{\dagger}.\mathbf{g})_{i,j} & -\sum\limits_{k=1}^{k=L+1} g_{k,j} g_{k,L+1}^{*} \\ \hline -\frac{1}{4}\frac{1-|\beta|^2}{1+|\beta|^2} + \sum\limits_{k=1}^{k=L+1} g_{k,L+1}^{*} g_{k,0} & \sum\limits_{k=1}^{k=L+1} g_{k,L+1}^{*} g_{k,j} & 0 \end{array}\right). \quad (71)$$

Based on the above calculations, the correlations for $|\beta\rangle$ can be written in terms of correlations of vacuum and ZME state. For rest of the correlation matrices, we have:

$$\mathbf{K}^{\beta} = \mathbf{K}^{0}, \qquad \bar{\mathbf{K}}^{\beta} = \bar{\mathbf{K}}^{0}, \quad (72)$$

$$\mathbf{G}^{\beta} = \left(\begin{array}{c|c|c} 0 & -2\sum\limits_{k=1}^{k=L+1} g_{k,0}^{*}(h_{k,j} + g_{k,j}) & -4\sum\limits_{k=1}^{k=L+1} g_{k,0}^{*} g_{k,L+1} \\ \hline 0 & (\mathbf{h}^{\dagger} - \mathbf{g}^{\dagger}).(\mathbf{h} + \mathbf{g})_{i,j} & 2\sum\limits_{k=1}^{k=L+1} (h_{k,j}^{*} - g_{k,j}^{*}) g_{k,L+1} \\ \hline \frac{1-|\beta|^2}{1+|\beta|^2} & 0 & 0 \end{array}\right). \quad (73)$$

For a higher point correlation like $\hat{\mathcal{O}}$, when operator $\hat{\mathcal{O}}$ contains fermionic creation and annihilation operators at position $0$ or $L+1$, then the relation between $\langle\hat{\mathcal{O}}\rangle_0$ and $\langle\hat{\mathcal{O}}\rangle_{\beta}$ would not be trivial. For example, in some cases, one can have Wick theorem for the $|\beta\rangle$ too. Some of the interesting cases are listed in table 1.

Results of this part can be extended to the zero parity state. Form of the correlation matrices for the state $|G_{\pm}\rangle$ (40), can simply be obtained by putting $\beta \to \pm 1$.

Table 1: Terms "even" and "odd" mean that the operator $\hat{\mathcal{O}}$ contains even or odd products of $c$-operators. Also, $k,l \neq 0, L+1$ and the Einstein summation rule is assumed. The notation $\langle \cdots \rangle_0$ stands for the expectation value calculated in the vacuum state.

| | $\hat{\mathcal{O}}$ has no $c_0^{(\dagger)}$ and $c_{L+1}^{(\dagger)}$ | |
| --- | --- | --- |
| | even | odd |
| $\langle \beta| c_0 \hat{\mathcal{O}} |\beta\rangle$ | $\frac{\mathrm{Re}[\beta]}{1+|\beta|^2} \langle \hat{\mathcal{O}} \rangle_0$ | $g_{k,0}^* \langle 0| \eta_k \hat{\mathcal{O}} |0\rangle$ |
| $\langle \beta| c_0^\dagger \hat{\mathcal{O}} |\beta\rangle$ | $\frac{\mathrm{Re}[\beta]}{1+|\beta|^2} \langle \hat{\mathcal{O}} \rangle_0$ | $-g_{l,0}^* \langle 0| \eta_l \hat{\mathcal{O}} |0\rangle$ |
| $\langle \beta| c_{L+1} \hat{\mathcal{O}} |\beta\rangle$ | $\frac{i\mathrm{Im}[\beta]}{1+|\beta|^2} \langle \hat{\mathcal{O}} \rangle_0$ | $g_{k,L+1}^* \langle 0| \eta_k \hat{\mathcal{O}} |0\rangle$ |
| $\langle \beta| c_{L+1}^\dagger \hat{\mathcal{O}} |\beta\rangle$ | $\frac{-i\mathrm{Im}[\beta]}{1+|\beta|^2} \langle \hat{\mathcal{O}} \rangle_0$ | $g_{l,L+1}^* \langle 0| \eta_l \hat{\mathcal{O}} |0\rangle$ |
| $\langle \beta| c_0^\dagger c_0 \hat{\mathcal{O}} |\beta\rangle$ | $\frac{1}{2} \langle \hat{\mathcal{O}} \rangle_0$ | $\frac{2\mathrm{Re}[\beta]}{1+|\beta|^2} \langle (g_{l,0}^* \eta_l - g_{l,0} \eta_l^\dagger)\hat{\mathcal{O}} \rangle_0$ |
| $\langle \beta| c_{L+1}^\dagger c_{L+1} \hat{\mathcal{O}} |\beta\rangle$ | $\frac{1}{2} \langle 0| \hat{\mathcal{O}} |0\rangle$ | $\frac{-2i\mathrm{Im}[\beta]}{1+|\beta|^2} \langle (g_{l,L+1}^* \eta_l + g_{l,L+1} \eta_l^\dagger)\hat{\mathcal{O}} \rangle_0$ |
| $\langle \beta| c_0^\dagger c_{L+1} \hat{\mathcal{O}} |\beta\rangle$ | $\frac{1}{4} \frac{|\beta|^2-1}{|\beta|^2+1} \langle \hat{\mathcal{O}} \rangle_0 - g_{l,0}^* g_{k,L+1} \langle \eta_l \eta_k^\dagger \hat{\mathcal{O}} \rangle_0$ | $\frac{\mathrm{Re}[\beta]}{1+|\beta|^2} \langle g_{l,L+1}^* \eta_l \hat{\mathcal{O}} \rangle_0 + \frac{i\mathrm{Im}[\beta]}{1+|\beta|^2} \langle g_{l,0}^* \eta_l \hat{\mathcal{O}} \rangle_0$ |
| $\langle \beta| c_0^\dagger c_{L+1}^\dagger \hat{\mathcal{O}} |\beta\rangle$ | $\frac{1}{4} \frac{1-|\beta|^2}{1+|\beta|^2} \langle \hat{\mathcal{O}} \rangle_0 - g_{l,0}^* g_{k,L+1} \langle \eta_l \eta_k^\dagger \hat{\mathcal{O}} \rangle_0$ | $\frac{\mathrm{Re}[\beta]}{1+|\beta|^2} \langle g_{l,L+1}^* \eta_l \hat{\mathcal{O}} \rangle_0 - \frac{i\mathrm{Im}[\beta]}{1+|\beta|^2} \langle g_{l,0}^* \eta_l \hat{\mathcal{O}} \rangle_0$ |
| | $\hat{\mathcal{O}}$ has no $\bar{\gamma}_0$ and $\gamma_{L+1}$ | |
| $\langle \beta| \gamma_0 \hat{\mathcal{O}} |\beta\rangle$ | $\frac{2\mathrm{Re}[\beta]}{1+|\beta|^2} \langle 0| \hat{\mathcal{O}} |0\rangle$ | $0$ |
| $\langle \beta| \bar{\gamma}_0 \hat{\mathcal{O}} |\beta\rangle$ | $0$ | $-2i g_{k,0}^* \langle 0| \eta_k \hat{\mathcal{O}} |0\rangle$ |

# 5 Reduced Density matrix

In this section, we calculate the density matrix and reduced density matrix (RDM) of the particular states introduced previously. We are going to use the configurational basis result of 3.4 and coherent basis formulation to calculate the density matrix and RDM. The RDM will be presented in both coherent basis and operator form. The operator form of RDM is useful to calculate the entanglement content of the states, while the coherent basis form of the RDM can be used to study the formation probabilities. In subsection 5.1, we start with the $\beta$-broken parity state (38). We calculate the total density matrix and then the RDM in coherent basis and operator format. In subsections 5.2, 5.3 and 5.4 we present the same calculations for the states $|0\rangle_\eta$, $|\emptyset\rangle$ and $|G_\pm\rangle$, respectively.

## 5.1 $\beta$ parity broken state

We start by calculating the reduced density matrix for the state $|\beta\rangle$ defined in (38). The total density matrix of a particular state $|\psi\rangle$ is defined as $\rho = |\psi\rangle\langle\psi|$. We prefer to calculate the density matrix for the state $|\beta\rangle$ explicitly in an exponential form using the definition (28). With the $+1$ parity for the state $|0\rangle_\eta$, the density matrix has the form

$$\rho^\beta = |\beta\rangle\langle\beta| = |C^\beta|^2 e^{\frac{1}{2}R_{ij}c_i^\dagger c_j^\dagger}(1+\beta\mathfrak{M}_{0k}c_k^\dagger)|0\rangle_c {}_c\langle0|(1+\beta^*\mathfrak{M}_{0l}^*c_l)e^{-\frac{1}{2}R_{ij}^*c_ic_j}, \tag{74}$$

where $|C^\beta|^2 = \left((1+|\beta|^2)\sqrt{\det[\mathbf{I}+\mathbf{R}^\dagger\mathbf{R}]}\right)^{-1}$ and $\mathfrak{M} = \mathbf{h}^*.\mathbf{R}+\mathbf{g}^*$. To proceed, we use the Fermionic coherent state defined as

$$\left|\xi\right\rangle = |\xi_1\xi_2\cdots\xi_N\rangle = e^{-\sum_{k=1}^N \xi_k c_k^\dagger}|0\rangle_c, \tag{75}$$

where $\xi_k$ are Grassmann variables. Therefore, we can write (following similar procedure as [2])

$$\left\langle\xi\right|\rho^\beta\left|\xi'\right\rangle = \rho^\beta(\xi,\xi') = |C^\beta|^2 e^{\frac{1}{2}R_{ij}\bar\xi_i\bar\xi_j}(1+\beta\mathfrak{M}_{0k}\bar\xi_k)(\beta^*\mathfrak{M}_{0l}^*\xi_l'+1)e^{-\frac{1}{2}R_{nm}^*\xi_n'\xi_m'}. \tag{76}$$

To obtain the reduced density matrix (RDM), we divide the system into two parts (subsystem) **1** and **2**. Here, we denote parts of any matrix that correspond to the subsystem **1** (**2**) with the subscript 1 (2)[3] . We trace out the subsystem **2** to find the RDM for subsystem **1**, $\rho_1 = \mathrm{tr}_2\rho$. In order to do so, we use the trace formula for operators in the coherent basis. Therefore, we have:

$$\rho_1^\beta(\xi,\xi') = \int \prod_{l\in\mathbf{2}}\mathrm{d}\bar\xi_l\mathrm{d}\xi_l \, e^{-\sum_{n\in2}\bar\xi_n\xi_n}\langle\xi_1,\cdots,\xi_k,-\xi_{k+1},\cdots,-\xi_L|\rho^\beta\left|\xi_1',\cdots,\xi_k',\xi_{k+1},\cdots,\xi_L\right\rangle, \tag{77}$$

where $\xi_1,\cdots,\xi_k$ belong to the subsystem **1** and $\xi_{k+1},\cdots,\xi_L$ belong to the subsystem 2. For the details of calculation, see appendix A. The final result after partial tracing the (76) is:

$$\rho_1^\beta(\xi,\xi') = C^\beta\Big[\Big(\frac{1}{\beta}+\mathcal{L}_1.\bar\xi+\mathcal{L}_2.\xi'\Big)\Big(\frac{1}{\beta^*}+\mathcal{L}_3.\bar\xi+\mathcal{L}_4.\xi'\Big)-\mathrm{Pf}[\mathcal{W}]\Big]e^{\frac{1}{2}\begin{pmatrix}\bar\xi & \xi'\end{pmatrix}\mathbf{\Omega}\begin{pmatrix}\bar\xi\\\xi'\end{pmatrix}}, \tag{78}$$

---

[3]For instance, the $\mathbf{A}_{12}$ stands for the sub-matrix of $\mathbf{A}$ that rows and columns belong to subsystem 1 and 2 respectively

where the $\mathcal{C}^\beta$ and the introduced matrices are given by:

$$\mathcal{C}^\beta = \frac{|\beta|^2 \sqrt{\det\left[\mathbf{I} + \mathbf{R}_{22}{}^\dagger \mathbf{R}_{22}\right]}}{(1 + |\beta|^2)\sqrt{\det\left[\mathbf{I} + \mathbf{R}^\dagger \mathbf{R}\right]}}, \tag{79a}$$

$$\mathbf{\Omega} = \begin{pmatrix} \mathbf{R}_{11} & 0 \\ 0 & -\mathbf{R}_{11}{}^* \end{pmatrix} + \begin{pmatrix} \mathbf{R}_{12} & 0 \\ 0 & \mathbf{R}_{12}{}^* \end{pmatrix} \mathcal{A}^{-1} \begin{pmatrix} \mathbf{R}_{12}{}^T & 0 \\ 0 & \mathbf{R}_{12}{}^\dagger \end{pmatrix}, \tag{79b}$$

$$\begin{pmatrix} \mathcal{L}_1 & \mathcal{L}_2 \\ \mathcal{L}_3 & \mathcal{L}_4 \end{pmatrix} = \begin{pmatrix} \mathfrak{M}_1 & 0 \\ 0 & \mathfrak{M}_1^* \end{pmatrix} + \begin{pmatrix} \mathfrak{M}_2 & 0 \\ 0 & -\mathfrak{M}_2^* \end{pmatrix} \mathcal{A}^{-1} \begin{pmatrix} \mathbf{R}_{12}^T & 0 \\ 0 & \mathbf{R}_{12}^\dagger \end{pmatrix}, \tag{79c}$$

$$\mathcal{W} = \begin{pmatrix} \mathfrak{M}_2 & 0 \\ 0 & -\mathfrak{M}_2^* \end{pmatrix} \mathcal{A}^{-T} \begin{pmatrix} \mathfrak{M}_2^T & 0 \\ 0 & -\mathfrak{M}_2^\dagger \end{pmatrix}, \tag{79d}$$

$$\mathcal{A} = \begin{pmatrix} \mathbf{R}_{22} & -\mathbf{I} \\ \mathbf{I} & -\mathbf{R}_{22}^* \end{pmatrix}. \tag{79e}$$

It is useful also to have the RDM in the operator format (for example to calculate the Rényi entanglement entropy). To derive the operator form for $\rho_1$ from equation (78), we rewrite the exponential term as

$$\text{Exp}\left[\tfrac{1}{2}\begin{pmatrix} \bar{\xi} & \xi' \end{pmatrix} \mathbf{\Omega} \begin{pmatrix} \bar{\xi} \\ \xi' \end{pmatrix}\right] = \text{Exp}\left[\tfrac{1}{2}\mathcal{X}_{ij}\bar{\xi}_i\bar{\xi}_j\right] \text{Exp}\left[\mathcal{Y}_{ij}\bar{\xi}_i\xi'_j\right] \text{Exp}\left[\tfrac{1}{2}\mathcal{Z}_{ij}\xi'_i\xi'_j\right] \tag{80}$$

with properly defined matrices $\mathcal{X}$, $\mathcal{Y}$ and $\mathcal{Z}$. Using the relations $c_i c_j |\xi\rangle = \xi_i \xi_j |\xi\rangle$ and $\langle\xi| c_i^\dagger c_j^\dagger = \langle\xi| \bar{\xi}_i \bar{\xi}_j$, one can replace $\bar{\xi}_i \bar{\xi}_j$ with $c_i^\dagger c_j^\dagger$ and $\xi'_i \xi'_j$ with $c_i c_j$ in the left and right exponentials. The cross term can be rewritten $\mathcal{Y}_{ij}\bar{\xi}_i\xi'_j \rightarrow \ln(\mathcal{Y})_{ij}c_i^\dagger c_j$. The final operator form of RDM (78) is given as:

$$\rho_1^\beta(c, c^\dagger) = \mathcal{C}^\beta \, e^{\tfrac{1}{2}\begin{pmatrix} \mathbf{c}^\dagger & \mathbf{c} \end{pmatrix} \mathcal{M} \begin{pmatrix} \mathbf{c} \\ \mathbf{c}^\dagger \end{pmatrix}} e^{\tfrac{1}{2}\text{tr}\ln\left(\tfrac{1}{2}\Omega_{12} - \tfrac{1}{2}\Omega_{21}^T\right)} \Big[\text{Pf}[\mathcal{W}] - \mathcal{L}_3 \mathcal{T}_{22} \mathcal{L}_2$$
$$+ \left(\mathcal{L}_1 \mathcal{T}_{22} \mathbf{c}^\dagger + (\mathcal{L}_1 \mathcal{T}_{21} + \mathcal{L}_2)\mathbf{c} + \frac{1}{\beta}\right)\left(\mathcal{L}_3 \mathcal{T}_{22}\mathbf{c}^\dagger + (\mathcal{L}_4 + \mathcal{L}_3 \mathcal{T}_{21})\mathbf{c} + \frac{1}{\beta^*}\right)\Big] \tag{81}$$

where

$$\mathcal{M} = \ln\mathcal{T}; \quad \mathcal{T} = \begin{pmatrix} \tfrac{1}{2}\Omega_{12} - \tfrac{1}{2}\Omega_{21}{}^T + 2\Omega_{11}(\Omega_{12}^T - \Omega_{21})^{-T}\Omega_{22} & 2\Omega_{11}(\Omega_{12}^T - \Omega_{21})^{-T} \\ 2(\Omega_{12}^T - \Omega_{21})^{-T}\Omega_{22} & 2(\Omega_{12}^T - \Omega_{21})^{-T} \end{pmatrix}, \tag{82a}$$

$$\boldsymbol{\Omega} = \begin{pmatrix} \mathbf{R}_{11} & 0 \\ 0 & -\mathbf{R}_{11}{}^* \end{pmatrix} + \begin{pmatrix} \mathbf{R}_{12} & 0 \\ 0 & \mathbf{R}_{12}{}^* \end{pmatrix} \mathcal{A}^{-1} \begin{pmatrix} \mathbf{R}_{12}{}^T & 0 \\ 0 & \mathbf{R}_{12}{}^\dagger \end{pmatrix}; \qquad \mathcal{A} = \begin{pmatrix} \mathbf{R}_{22} & -\mathbf{I} \\ \mathbf{I} & -\mathbf{R}_{22}^* \end{pmatrix} \tag{82b}$$

$$\mathcal{W} = \begin{pmatrix} \mathfrak{M}_2 & 0 \\ 0 & -\mathfrak{M}_2^* \end{pmatrix} \mathcal{A}^{-T} \begin{pmatrix} \mathfrak{M}_2^T & 0 \\ 0 & -\mathfrak{M}_2^\dagger \end{pmatrix} \tag{82c}$$

$$\begin{pmatrix} \mathcal{L}_1 & \mathcal{L}_2 \\ \mathcal{L}_3 & \mathcal{L}_4 \end{pmatrix} = \begin{pmatrix} \mathfrak{M}_1 & 0 \\ 0 & \mathfrak{M}_1^* \end{pmatrix} + \begin{pmatrix} \mathfrak{M}_2 & 0 \\ 0 & -\mathfrak{M}_2^* \end{pmatrix} \mathcal{A}^{-1} \begin{pmatrix} \mathbf{R}_{12}^T & 0 \\ 0 & \mathbf{R}_{12}^\dagger \end{pmatrix}. \tag{82d}$$

Also, $\mathcal{T}_{11}$, $\mathcal{T}_{12}$, $\mathcal{T}_{21}$ and $\mathcal{T}_{22}$ stand for the sub-matrices (blocks) of matrix $\mathcal{T}$. In the above expression, to move the exponential with fermionic operators, we have used the following relation, coming from Baker-Hausdorff formula,

$$F^{-1} \begin{pmatrix} \mathbf{c} \\ \mathbf{c}^\dagger \end{pmatrix} F = \mathcal{T} \begin{pmatrix} \mathbf{c} \\ \mathbf{c}^\dagger \end{pmatrix} \Rightarrow \begin{pmatrix} \mathbf{c} \\ \mathbf{c}^\dagger \end{pmatrix} F = F\mathcal{T} \begin{pmatrix} \mathbf{c} \\ \mathbf{c}^\dagger \end{pmatrix}; \qquad F = e^{\frac{1}{2}\begin{pmatrix} \mathbf{c}^\dagger & \mathbf{c} \end{pmatrix} \mathcal{M} \begin{pmatrix} \mathbf{c} \\ \mathbf{c}^\dagger \end{pmatrix}}. \tag{83}$$

In the expression of RDM, having all the creation and annihilation operators in the argument of exponential is preferred (some of calculations would be simplified). For this reason, we present another calculation of $\rho_1^\beta$ in the appendix B, where using a trick, we managed to get the RDM with two exponentials. The above equations are valid for the RDM of $|\beta\rangle$-state, which also means, for any given bi partition, one can use (81). However, it should be noted that the spin and fermion representations for solvable quantum chains lead to different RDMs. For more details see section 5.5.

We can propose an ansatz to write the RDM for $|\beta\rangle$ in term of correlation matrix. Although computationally favorable, the down side of such an ansatz is that we could only use it for a particular type of bipartition. while we can not apply Wick theorem to $|\beta\rangle$, one can use the $\boldsymbol{\Gamma}^\beta$ matrix to calculate the RDM of a subsystem which starts from one boundary. One has to make an adjustment to the Peschel method. Such a procedure for $\beta = \pm 1$ has been shown in [49]; here we are extending that result.

Since the $\boldsymbol{\Gamma}^\beta$ matrix is a skew symmetric matrix it can be written in a block form using an orthogonal matrix $\mathbf{V}$ as:

$$\mathbf{V}\boldsymbol{\Gamma}^\beta \mathbf{V}^T = \begin{pmatrix} 0 & i\boldsymbol{\nu} \\ -i\boldsymbol{\nu} & 0 \end{pmatrix}, \tag{84}$$

where $\boldsymbol{\nu}$ is a diagonal matrix. Then, we can define the following fermionic operators:

$$\begin{pmatrix} \boldsymbol{d}^\dagger \\ \boldsymbol{d} \end{pmatrix} = \frac{1}{\sqrt{2}} \mathbf{W} \begin{pmatrix} \boldsymbol{\gamma} \\ \bar{\boldsymbol{\gamma}} \end{pmatrix} = \frac{1}{2} \begin{pmatrix} \mathbf{I} & i\mathbf{I} \\ \mathbf{I} & -i\mathbf{I} \end{pmatrix} \mathbf{V} \begin{pmatrix} \boldsymbol{\gamma} \\ \bar{\boldsymbol{\gamma}} \end{pmatrix}. \tag{85}$$

Similar to the results in [49], one can make an ansatz for the RDM of the subsystem **1**. To be precise, we are assuming that the subsystem **1** is a connected bipartite of the system starting from site 0 to $\ell$. The ansatz should have a form like below with respect to the operators that diagonalize the $\mathbf{\Gamma}^\beta$ matrix.

$$\rho_1^\beta(d, d^\dagger) = g(d_0, d_0^\dagger, d_0^\dagger d_0) \times \prod_{k=1}^\ell \left( \frac{1 + \nu_k}{2} d_k^\dagger d_k + \frac{1 - \nu_k}{2} d_k d_k^\dagger \right),$$

where g is an arbitrary function to be determined. From correlation matrices (section 4), we can realize that

$$g(c_0, c_0^\dagger, c_0^\dagger c_0) = \frac{\text{Re}[\beta]}{1 + |\beta|^2}(c_0^\dagger + c_0) + \frac{1}{2}\mathbf{I}. \tag{86}$$

This ansatz satisfies the expectations of the state $|\beta\rangle$ including one point functions (69). It is easy to show that $c^\dagger + c = \sqrt{2}\left(W_{11}^\dagger d^\dagger + W_{11}^T d\right)$, where $W$ is the unitary transformation which diagonalizes the matrix $\mathbf{\Gamma}^\beta$. Then, in terms of $d$ and $d^\dagger$ operators we have:

$$\rho_1^\beta(d, d^\dagger) = \frac{\text{Re}[\beta]}{1 + |\beta|^2}\left( \frac{\sqrt{2}}{2}d_0 + \frac{\sqrt{2}}{2}d_0^\dagger + \frac{1}{2}\mathbf{I} \right) \times \prod_{k=1}^\ell \left( \frac{1 + \nu_k}{2} d_k^\dagger d_k + \frac{1 - \nu_k}{2} d_k d_k^\dagger \right), \tag{87}$$

The ansatz (87) respects the generalized Wick's theorem, which means that this RDM produces all the correlation functions correctly. As an example, one point function is not necessarily zero for the $|\beta\rangle$:

$$\langle d_k^\dagger \rangle_\beta = \frac{1}{\sqrt{2}} \frac{\text{Re}[\beta]}{1 + |\beta|^2} \delta_{k,0}, \tag{88}$$

$$\langle d_k \rangle_\beta = \frac{1}{\sqrt{2}} \frac{\text{Re}[\beta]}{1 + |\beta|^2} \delta_{k,0}. \tag{89}$$

Based on (72), (73) and (57), the first row and column of matrix $\mathbf{\Gamma}^\beta$ is zero which evidently means it has a zero eigenvalue, $\nu_0 = 0$. From the earlier statement it can be inferred that the matrix which diagonalizes the $\mathbf{\Gamma}^\beta$ -called $W$- has the following form:

$$W = \begin{pmatrix} W_{11} & W_{12} \\ W_{11}^* & W_{12}^* \end{pmatrix}; \qquad W_{11} = \begin{pmatrix} 1 & 0 & \cdots & 0 \\ 0 & & & \\ \vdots & & \boldsymbol{\omega} & \\ 0 & & & \end{pmatrix}. \tag{90}$$

The reason we have defined the $W$ as above is that the $d^\dagger$ and $d$ should be related with a conjugate transpose transformation. From (84), The $W$ and $V$ matrices are related by $W \propto \begin{pmatrix} \mathbf{I} & i\mathbf{I} \\ \mathbf{I} & -i\mathbf{I} \end{pmatrix} V$, which means we can write:

$$\begin{pmatrix} d^\dagger \\ d \end{pmatrix} = \frac{1}{\sqrt{2}} \begin{pmatrix} W_{11}\gamma + W_{12}\bar{\gamma} \\ W_{11}^*\gamma + W_{12}^*\bar{\gamma} \end{pmatrix}. \tag{91}$$

We know that $\langle \bar{\gamma}_i \rangle_\beta = 0$ and $\langle \gamma_i \rangle_\beta = \delta_{i,0}$, then we can write $\langle d_k^\dagger \rangle_\beta = (W_{11})_{k,0}$. Making use of the relation (90), we have $\langle d_k^\dagger \rangle_\beta \propto \delta_{k,0}$. For the $\langle d_k \rangle_\beta$ case, we use the fact that $d$ and $d^\dagger$ are related by a conjugation, which means that $\langle d_k \rangle_\beta \propto \delta_{k,0}$ and the proof is complete.

Note that, the above calculation is also valid for states $|G_+\rangle$, $|G_-\rangle$ and any state created from the action of $\eta^\dagger$'s on these states. Consequently, with a proper $\Gamma$ matrix, a similar ansatz to (87) can be used to study the entanglement for all the states belonging to the four sectors created with $|G_\pm\rangle$ as introduced in subsection 3.6.

## 5.2 Vacuum state

As it was mentioned before the limit $\beta \rightarrow 0$ in the $|\beta\rangle$, gives the $|0\rangle_\eta$. Equivalently, $\lim_{\beta \to 0} \rho^\beta = \rho^0 = |0\rangle_\eta {}_\eta\langle 0|$. Simply, we are going to apply this limit on the results of subsection 5.1 to find the required quantities in this subsection. For instance, in the limit $\beta \rightarrow 0$, the only nonzero term in (78) is $|\mathcal{C}^\beta|^2 \frac{1}{|\beta|^2}$. Therefore for the RDM of vacuum state in the coherent basis we have:

$$\rho_1^0(\xi, \xi') = \frac{\sqrt{\det\left[\mathbf{I} + \mathbf{R}_{22}{}^\dagger \mathbf{R}_{22}\right]}}{\sqrt{\det\left[\mathbf{I} + \mathbf{R}^\dagger \mathbf{R}\right]}} e^{\frac{1}{2}\begin{pmatrix} \bar{\xi} & \xi' \end{pmatrix} \Omega \begin{pmatrix} \bar{\xi} \\ \xi' \end{pmatrix}}, \tag{92}$$

where:

$$\mathcal{A} = \begin{pmatrix} \mathbf{R}_{22} & -\mathbf{I} \\ \mathbf{I} & -\mathbf{R}_{22}^* \end{pmatrix}, \tag{93a}$$

$$\Omega = \begin{pmatrix} \mathbf{R}_{11} & 0 \\ 0 & -\mathbf{R}_{11}{}^* \end{pmatrix} + \begin{pmatrix} \mathbf{R}_{12} & 0 \\ 0 & \mathbf{R}_{12}{}^* \end{pmatrix} \mathcal{A}^{-1} \begin{pmatrix} \mathbf{R}_{12}{}^T & 0 \\ 0 & \mathbf{R}_{12}{}^\dagger \end{pmatrix}. \tag{93b}$$

The operator form of RDM reads as

$$\rho_1^0(c, c^\dagger) = \frac{\sqrt{\det\left[\mathbf{I} + \mathbf{R}_{22}{}^\dagger \mathbf{R}_{22}\right]}}{\sqrt{\det\left[\mathbf{I} + \mathbf{R}^\dagger \mathbf{R}\right]}} e^{\frac{1}{2}\begin{pmatrix} \mathbf{c}^\dagger & \mathbf{c} \end{pmatrix} \mathcal{M} \begin{pmatrix} \mathbf{c} \\ \mathbf{c}^\dagger \end{pmatrix}} e^{\frac{1}{2}\operatorname{tr}\ln(\frac{1}{2}\Omega_{12} - \frac{1}{2}\Omega_{21}^T)}, \tag{94}$$

where

$$\mathcal{M} = \ln \mathcal{T}; \quad \mathcal{T} = \begin{pmatrix} \frac{1}{2}\Omega_{12} - \frac{1}{2}\Omega_{21}{}^T + 2\Omega_{11}(\Omega_{12}^T - \Omega_{21})^{-T}\Omega_{22} & 2\Omega_{11}(\Omega_{12}^T - \Omega_{21})^{-T} \\ 2(\Omega_{12}^T - \Omega_{21})^{-T}\Omega_{22} & 2(\Omega_{12}^T - \Omega_{21})^{-T} \end{pmatrix}, \tag{95a}$$

$$\Omega = \begin{pmatrix} \mathbf{R}_{11} & 0 \\ 0 & -\mathbf{R}_{11}^* \end{pmatrix} + \begin{pmatrix} \mathbf{R}_{12} & 0 \\ 0 & \mathbf{R}_{12}^* \end{pmatrix} \mathcal{A}^{-1} \begin{pmatrix} \mathbf{R}_{12}^T & 0 \\ 0 & \mathbf{R}_{12}^\dagger \end{pmatrix}; \quad \mathcal{A} = \begin{pmatrix} \mathbf{R}_{22} & -\mathbf{I} \\ \mathbf{I} & -\mathbf{R}_{22}^* \end{pmatrix}. \tag{95b}$$

Since one can use Wick theorem for the state $|0\rangle_\eta$, the RDM can be written in terms of Majorana fermions and their correlations (introduced in section 4) in subsystem **1**. The reduced density matrix can be written as [53]:

$$\rho_1^0(\gamma, \bar{\gamma}) = [\det \frac{\mathbf{I} - \mathbf{\Gamma}_1^0}{2}]^{\frac{1}{2}} e^{\frac{1}{4}\left(\gamma \quad \bar{\gamma}\right)\ln\frac{\mathbf{I}+\mathbf{\Gamma}_1^0}{\mathbf{I}-\mathbf{\Gamma}_1^0}\begin{pmatrix}\gamma \\ \bar{\gamma}\end{pmatrix}}, \tag{96}$$

where the $\left(\gamma \quad \bar{\gamma}\right)$ contains all the Majorana fermions of subsystem **1**. The $\mathbf{\Gamma}_1^0$ stands for the correlation matrix defined in section 4 calculated for the vacuum state and subsystem **1**. The constant in (96) can be simplified as:

$$\det[\frac{\mathbf{I} - \mathbf{\Gamma}_1^0}{2}] = \det[\mathbf{K}_1^0]\det[\frac{\bar{\mathbf{K}}_1^0 - \mathbf{G}_1^0.\mathbf{K}_1^{0^{-1}}.\mathbf{G}_1^{0^T}}{4}]. \tag{97}$$

### 5.3 ZME state

The ZME state can be obtained by the limit $|\emptyset\rangle = \lim_{\beta \to \infty} |\beta\rangle$. Therefore, the RDM $\rho^\emptyset = |\emptyset\rangle\langle\emptyset|$ in the operator form will be the large $\beta$ limit of the results of the subsection 5.1.

$$\rho_1^\beta(c, c^\dagger) = \mathcal{C}^\emptyset e^{\frac{1}{2}\left(c^\dagger \quad c\right)\mathcal{M}\begin{pmatrix}c \\ c^\dagger\end{pmatrix}} e^{\frac{1}{2}\mathrm{tr}\ln(\frac{1}{2}\Omega_{12} - \frac{1}{2}\Omega_{21}^T)}\Big[\mathrm{Pf}[\mathcal{W}] - \mathcal{L}_3\mathcal{T}_{22}\mathcal{L}_2$$
$$+ \left(\mathcal{L}_1\mathcal{T}_{22}c^\dagger + (\mathcal{L}_1\mathcal{T}_{21} + \mathcal{L}_2)c\right)\left(\mathcal{L}_3\mathcal{T}_{22}c^\dagger + (\mathcal{L}_4 + \mathcal{L}_3\mathcal{T}_{21})c\right)\Big] \tag{98}$$

where

$$\mathcal{W} = \begin{pmatrix}\mathfrak{M}_2 & 0 \\ 0 & -\mathfrak{M}_2^*\end{pmatrix}\mathcal{A}^{-T}\begin{pmatrix}\mathfrak{M}_2^T & 0 \\ 0 & -\mathfrak{M}_2^\dagger\end{pmatrix}; \qquad \mathcal{A} = \begin{pmatrix}\mathbf{R}_{22} & -\mathbf{I} \\ \mathbf{I} & -\mathbf{R}_{22}^*\end{pmatrix}, \tag{99a}$$

$$\mathcal{M} = \ln\mathcal{T}; \quad \mathcal{T} = \begin{pmatrix}\frac{1}{2}\Omega_{12} - \frac{1}{2}\Omega_{21}^T + 2\Omega_{11}(\Omega_{12}^T - \Omega_{21})^{-T}\Omega_{22} & 2\Omega_{11}(\Omega_{12}^T - \Omega_{21})^{-T} \\ 2(\Omega_{12}^T - \Omega_{21})^{-T}\Omega_{22} & 2(\Omega_{12}^T - \Omega_{21})^{-T}\end{pmatrix}, \tag{99b}$$

$$\Omega = \begin{pmatrix}\mathbf{R}_{11} & 0 \\ 0 & -\mathbf{R}_{11}^*\end{pmatrix} + \begin{pmatrix}\mathbf{R}_{12} & 0 \\ 0 & \mathbf{R}_{12}^*\end{pmatrix}\mathcal{A}^{-1}\begin{pmatrix}\mathbf{R}_{12}^T & 0 \\ 0 & \mathbf{R}_{12}^\dagger\end{pmatrix}, \tag{99c}$$

$$\begin{pmatrix}\mathcal{L}_1 & \mathcal{L}_2 \\ \mathcal{L}_3 & \mathcal{L}_4\end{pmatrix} = \begin{pmatrix}\mathfrak{M}_1 & 0 \\ 0 & \mathfrak{M}_1^*\end{pmatrix} + \begin{pmatrix}\mathfrak{M}_2 & 0 \\ 0 & -\mathfrak{M}_2^*\end{pmatrix}\mathcal{A}^{-1}\begin{pmatrix}\mathbf{R}_{12}^T & 0 \\ 0 & \mathbf{R}_{12}^\dagger\end{pmatrix}. \tag{99d}$$

The above equation is lengthy and calculation of entanglement seem difficult with the above RDM. However, since the ZME state has an opposite parity with respect to $|0\rangle_\eta$. One can use the *Tilda transformation* introduced in the subsection 3.4 to write the ZME state as:

$$|\emptyset\rangle = C^\emptyset e^{\frac{1}{2} \sum_{i,j} R^\emptyset_{ij} \tilde{c}^\dagger_i \tilde{c}^\dagger_j} |\tilde{0}\rangle_c \tag{100}$$

with properly defined $\mathbf{R}^\emptyset$ matrix and constant $C^\emptyset$. The above Gaussian form for the ZME state will simplify some of the calculations exceedingly. For instance, using the (100), it is possible to write a shorter notation for RDM as:

$$\rho^\emptyset_1(c,c^\dagger) = C^{\emptyset'} e^{\frac{1}{2} \begin{pmatrix} \mathbf{c}^\dagger & \mathbf{c} \end{pmatrix} \mathcal{M}^\emptyset \begin{pmatrix} \mathbf{c} \\ \mathbf{c}^\dagger \end{pmatrix}}, \tag{101}$$

where $C^{\emptyset'}$ is the normalization factor and $\mathcal{M}^\emptyset = \ln \mathcal{T}^\emptyset$ which we have:

$$\mathcal{T}^\emptyset = \begin{pmatrix} \frac{1}{2}\mathbf{\Omega}^\emptyset_{12} - \frac{1}{2}\mathbf{\Omega}^{\emptyset\,T}_{21} + 2\mathbf{\Omega}^\emptyset_{11}(\mathbf{\Omega}^{\emptyset\,T}_{12} - \mathbf{\Omega}^\emptyset_{21})^{-T}\mathbf{\Omega}^\emptyset_{22} & 2\mathbf{\Omega}^\emptyset_{11}(\mathbf{\Omega}^{\emptyset\,T}_{12} - \mathbf{\Omega}^\emptyset_{21})^{-T} \\ 2(\mathbf{\Omega}^{\emptyset\,T}_{12} - \mathbf{\Omega}^\emptyset_{21})^{-T}\mathbf{\Omega}^\emptyset_{22} & 2(\mathbf{\Omega}^{\emptyset\,T}_{12} - \mathbf{\Omega}^\emptyset_{21})^{-T} \end{pmatrix}, \tag{102a}$$

$$\mathbf{\Omega}^\emptyset = \begin{pmatrix} \mathbf{R}^\emptyset_{11} & 0 \\ 0 & -\mathbf{R}^{\emptyset\,*}_{11} \end{pmatrix} + \begin{pmatrix} \mathbf{R}^\emptyset_{12} & 0 \\ 0 & \mathbf{R}^{\emptyset\,*}_{12} \end{pmatrix} \mathcal{A}^{\emptyset-1} \begin{pmatrix} \mathbf{R}^{\emptyset\,T}_{12} & 0 \\ 0 & \mathbf{R}^{\emptyset\,\dagger}_{12} \end{pmatrix}; \qquad \mathcal{A}^\emptyset = \begin{pmatrix} \mathbf{R}^\emptyset_{22} & -\mathbf{I} \\ \mathbf{I} & -\mathbf{R}^{\emptyset\,*}_{22} \end{pmatrix}. \tag{102b}$$

The Wick theorem can also be applied to ZME state. Likewise, it is possible to write the RDM in terms of correlation matrices and Majorana fermions of subsystem **1**. We can write the RDM as:

$$\rho^\emptyset_1(\gamma,\bar\gamma) = [\det \frac{\mathbf{I} - \mathbf{\Gamma}^\emptyset_1}{2}]^{\frac{1}{2}} e^{\frac{1}{4} \begin{pmatrix} \gamma & \bar\gamma \end{pmatrix} \ln \frac{\mathbf{I}+\mathbf{\Gamma}^\emptyset_1}{\mathbf{I}-\mathbf{\Gamma}^\emptyset_1} \begin{pmatrix} \gamma \\ \bar\gamma \end{pmatrix}}. \tag{103}$$

In the above expression, the matrix $\mathbf{\Gamma}^\emptyset_1$ is defined in subsection 4.2, and the subscript stands for the correlation matrix for the subsystem.

## 5.4 Zero parity eigenstates

The zero parity eigenstates $|G_\pm\rangle$ are the cases where $\beta = \pm 1$. Using the results of (78) and (81), the RDM matrix in coherent basis and operator form for $|G_\pm\rangle$ are given by setting $\beta = \pm 1$.

## 5.5 Spin versus fermion representation

Although the previous considerations are advantageous numerically to study RDM's, we have to point out that the RDM for the spin representation and the fermionic representation (of the Hamiltonian) are not identical, necessarily. Correspondingly, the entanglement entropies could end up to be different. Based on the way of selecting the subsystem, RDM's (of spin and fermion representations) could be different or equal. This difference can be expected due to the non-local

structure of the Jordan-Wigner transformation [28]. In general, we are interested in two scenarios for subsystem bipartition as demonstrated in the figure 1.

For start, if our desired state (like $|0\rangle$ and $|\emptyset\rangle$) is an eigenstate of parity operator, then RDM has equal form in spin and fermion representations for types (a) and (b) of subsystems in figure 1. Since the effect of the Jordan-Wigner strings disappears in these cases. This statement is true for any boundary as long as boundary terms do not break the parity symmetry. If subsystem is not connected, then starting from fermionic representation, to obtain a spin correlation function (like $\langle \sigma_i^x \sigma_j^x \rangle$), one needs information about the string of sites between two blocks of subsystem in the fermionic picture, while, this is not necessary if one asks only for fermionic correlations.

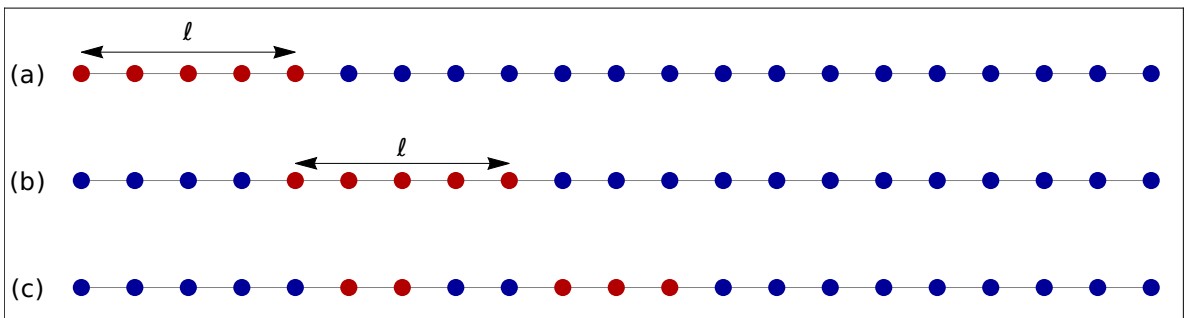

Figure 1: The three subsystem block configurations we consider here. (a) a block of length $\ell$ starting from the boundary. (b) is a block of length $\ell$ at a distance from the boundary. The case (c) shows a disconnected subsystem which does not start from any of boundary points.

In the related case to our study, which we are dealing with an open boundary case and a state which does not respects the parity (like $|\beta\rangle$), then the relation between spin and fermion version of RDM is more peculiar. Essentially, the string of $\sigma^z$'s from Jordan-Wigner transformation would break off the correspondence between spins and fermions for a subsystem separated from the boundary, similar to (b) in figure 1. It is an anomaly of parity broken state that the RDM of any interval starting from the boundary is the same for spins and fermions. While not starting from one of the boundaries of the chain, other techniques should be used to find the RDM of spin representation [27]. The arguments that are presented in this part can be summarized in the table 2.

Table 2: A handy representation of the statements in the section 5.5. In the below, $\rho_{\text{spin}}$ stands for the RDM in the spin representation and $\rho_{\text{fermion}}$ is the RDM in the fermionic representation of the Hamiltonian. The subsystem types are demonstrated in figure 1. In addition, we have assumed the restriction on $\beta \neq 0, \infty$.

| State \ Subsystem | type (a) | type (b) | type (c) |
|---|---|---|---|
| $|0\rangle, |\emptyset\rangle$ | $\rho_{\text{spin}} = \rho_{\text{fermion}}$ | $\rho_{\text{spin}} = \rho_{\text{fermion}}$ | $\rho_{\text{spin}} \neq \rho_{\text{fermion}}$ |
| $|\beta\rangle$ | $\rho_{\text{spin}} = \rho_{\text{fermion}}$ | $\rho_{\text{spin}} \neq \rho_{\text{fermion}}$ | $\rho_{\text{spin}} \neq \rho_{\text{fermion}}$ |

# 6 Entanglement Entropy

Given the reduced density matrix $\rho_A$ describing knowledge of the state of a subsystem $A$, the Rényi entanglement entropy is given by:

$$S_n(A) = \frac{1}{1-n} \log(\text{tr}_A \rho_A^n). \tag{104}$$

The Rényi entanglement entropy can be seen as the generalized version of von Neumann entanglement measure. In the limit $\alpha \to 1$, Rényi EE produces the Von Neumann EE ($S_{vN}$),

$$S_{vN}(A) = -\text{tr}_A[\rho_A \log \rho_A]. \tag{105}$$

The difficulty of calculation of the entanglement grows exponentially with size of the subsystem. One can find a basis which RDM is diagonal, however, it would still be computationally disadvantageous.

It is possible to have special structure for the RDM to simplify the entanglement calculation. For instance having a Gaussian form for RDM simplifies the calculation, or being able to write the RDM in terms of correlation matrices. In the rest of this section, we first present the result of entanglement for the vacuum state and the ZME state which are basically the results in [4–7]. We are going to give the entanglement in terms of the eigenvalues of the correlation matrices and Gaussian form of the RDM. Next, we will discuss the entanglement calculation for a general parity broken state such as $|\beta\rangle$ and the limiting cases of $|G_\pm\rangle$ which are new.

## 6.1 Vacuum and ZME state

Since the Wick theorem can be applied to the vacuum state we were able to write the RDM as (94). Using the equations (94) and (104), we can trace the RDM and write the Rényi entanglement for this state as

$$S_n^0(A) = \frac{1}{1-n} \log \left( C^\alpha \det\left[ \mathbf{I} + e^{n\mathcal{M}} \right]^{\frac{1}{2}} \right), \tag{106}$$

where the constant C is given by

$$C = \frac{\sqrt{\det\left[ \mathbf{I} + \mathbf{R}_{22}^\dagger \mathbf{R}_{22} \right]}}{\sqrt{\det\left[ \mathbf{I} + \mathbf{R}^\dagger \mathbf{R} \right]}} e^{\frac{1}{2}\text{tr}\ln(\frac{1}{2}\mathfrak{n}_{12} - \frac{1}{2}\mathfrak{n}_{21}^T)}. \tag{107}$$

The matrices $\mathcal{M}$ and $\boldsymbol{\Omega}$ are already defined in (95). This relation is computationally favorable, since all we need to calculate is a determinant.

As it was written in section 5.2, we could have also express the RDM in terms of correlation matrices of Majorana fermions (96). Therefore, it is possible to find the entanglement in terms of the correlation matrix $\boldsymbol{\Gamma}^0$ (of the vacuum state) for the subsystem $A$. Therefore, using the RDM (96) and the trace formula for Gaussian function of fermions, the Rényi EE is given by

$$S_n^0(A) = \frac{1}{2(1-n)} \log \det\left[ \left(\frac{1+\boldsymbol{\Gamma}_A^0}{2}\right)^n + \left(\frac{1-\boldsymbol{\Gamma}_A^0}{2}\right)^n \right] = \frac{1}{(1-n)} \sum_j \log\left[ \left(\frac{1+\nu_j^0}{2}\right)^n + \left(\frac{1-\nu_j^0}{2}\right)^n \right] \tag{108}$$

where the set of $\{\pm\nu_k\}$ are the eigenvalues of $\boldsymbol{\Gamma}_A^0$ for the subsystem $A$. The above relation is favorable because it is straightforward to find the correlation matrix for states that obeys Wick theorem.

The entanglement for the ZME state can have a similar form. For instance, in subsection (5.3), it was mentioned that through some canonical transformations it is possible to write the RDM in Gaussian form (101). Similar to previous state, using the Gaussian form of RDM and tracing that, we can get the Rényi entanglement for subsystem $A$ same as (106) but we have to use $\mathbf{R}^{\emptyset}$ instead of $\mathbf{R}^0$. On the other hand, the Wick theorem can be applied to the ZME state. Hence, it is possible to relate the EE to the correlation matrices, similar to (108). We only need to use the $\mathbf{\Gamma}_A^{\emptyset}$ given in section 4.2.

Based on the form of $\mathbf{\Gamma}_A^{\emptyset}$ and $\mathbf{\Gamma}_A^0$, one can deduce that for any subsystem $A$ which does not include the last lattice point $L+1$, they have equal set of eigenvalues ($\{\nu^0\} = \{\nu^{\emptyset}\}$). It also means that the entanglement $S_{\alpha}^0$ and $S_{\alpha}^{\emptyset}$ are equal. It means that the states $|0\rangle$ and $|\emptyset\rangle$ have the same entanglement properties for any given model ($\mathbf{A}$ and $\mathbf{B}$ matrices which depend on the model).

## 6.2 General $\beta$-parity broken state

Unlike the vacuum and ZME states, there are not many computationally easy methods to study the entanglement entropy for a state such as $|\beta\rangle$. Namely, we can not use the correlation matrix blindly for this state. In this subsection, we first focus on a computationally favorable method to study the EE for such state. Then, we offer a relation for special type of bipartition which relies on correlation matrix. For start, when $n \in \mathbb{N}$ we can use Berezin integrations to find the $\rho_A^{\beta^n}$. The steps of calculation are written in appendix C. The Rényi entanglement entropy for $n = 2$ is given by:

$$
\begin{aligned}
\mathrm{tr}(\rho_A^{\beta^2}) =& \mathcal{C}^{\beta^2} \mathrm{Pf}[\mathcal{B}] \Big\{ \Big( \frac{1}{|\beta|^2} + \mathrm{Pf}[\mathcal{W}] \Big)^2 + \mathrm{Pf}[\mathcal{W}] \big( \mathbb{C}\mathcal{B}^{-T}\mathbb{C}^T|_{1,2} + \mathbb{C}\mathcal{B}^{-T}\mathbb{C}^T|_{3,4} \big) \\
& - \frac{\mathbb{C}\mathcal{B}^{-T}\mathbb{C}^T|_{2,4}}{\beta^2} - \frac{\mathbb{C}\mathcal{B}^{-T}\mathbb{C}^T|_{1,3}}{\beta^{*2}} + \frac{1}{|\beta|^2} \big( \mathbb{C}\mathcal{B}^{-T}\mathbb{C}^T|_{1,2} + \mathbb{C}\mathcal{B}^{-T}\mathbb{C}^T|_{3,4} - \mathbb{C}\mathcal{B}^{-T}\mathbb{C}^T|_{2,3} \\
& - \mathbb{C}\mathcal{B}^{-T}\mathbb{C}^T|_{1,4} \big) + \mathrm{Pf}[\mathbb{C}.\mathcal{B}^{-T}.\mathbb{C}^T] \Big\}.
\end{aligned}
\tag{109}
$$

Where,

$$
\mathcal{B} = \begin{pmatrix} \mathbf{\Omega}_{11} & \mathbf{0} & -\mathbf{\Omega}_{12} & \mathbf{I} \\ \mathbf{0} & \mathbf{\Omega}_{11} & \mathbf{I} & \mathbf{\Omega}_{12} \\ -\mathbf{\Omega}_{21} & -\mathbf{I} & \mathbf{\Omega}_{22} & \mathbf{0} \\ -\mathbf{I} & \mathbf{\Omega}_{21} & \mathbf{0} & \mathbf{\Omega}_{22} \end{pmatrix}, \qquad \mathbb{C} = \begin{pmatrix} -\mathcal{L}_1 & \mathbf{0} & \mathcal{L}_2 & \mathbf{0} \\ -\mathcal{L}_3 & \mathbf{0} & \mathcal{L}_4 & \mathbf{0} \\ \mathbf{0} & \mathcal{L}_1 & \mathbf{0} & \mathcal{L}_2 \\ \mathbf{0} & \mathcal{L}_3 & \mathbf{0} & \mathcal{L}_4 \end{pmatrix}.
\tag{110}
$$

The $\mathbf{\Omega}$ and $\mathcal{L}$ are given by (79). Also, the notation $\mathbb{X}|_{r,s}$ stand for the element of the matrix $\mathbb{X}$ at the position $(r, s)$. The above expression can be used for any bipartition of the system. However, we are restricted to the $n = 2$. From section 4.3, we expect to see no $\beta$-dependence in the entanglement entropy when the subsystem does not contain boundary points [28]. Despite that, the entanglement content would not be the same from spin perspective to the fermion one. For the case of general parity broken state, if subsystem is not a connected bipartition starting from boundary, then spin entanglement and fermion entanglement do not agree.

In section 5.1, we proved that it is possible to have an ansatz to write the RDM in the diagonal form of (87). Using this form of RDM one can write the Rényi EE as:

$$S_n^\beta(A) = \frac{1}{1-n} \sum_{j=1}^{l} \log\left((\frac{1+\nu_j}{2})^n + (\frac{1-\nu_j}{2})^n\right) + \frac{1}{1-n} \log[(\tfrac{1}{2} + f_\beta)^n + (\tfrac{1}{2} - f_\beta)^n], \tag{111}$$

where $f_\beta = \frac{\text{Re}[\beta]}{1+|\beta|^2}$, and it is the eigenvalue of the zeroth part of RDM (87). In the above $\nu_j$'s are the eigenvalues of correlation matrix $\mathbf{\Gamma}^\beta$. The formula above simplifies the entanglement studies, however, it is valid for a certain type of the subsystems. The set $A$ should be connected subsystem of the system containing the site 0 (first site). Otherwise, we would not be able to get entanglement from correlations of the system.

It is crucial to mention that form of correlation $\mathbf{\Gamma}^\beta$, in (72) and (73), indicates that the eigenvalues of $\mathbf{\Gamma}_A^\beta$ for any subsystem $0 \in A$ and $L+1 \notin A$ are the same as $\mathbf{\Gamma}_A^0$. This statement means that there is a relation between entanglements $S_n^0(A)$, $S_n^\emptyset(A)$ and $S_n^\beta(A)$ (only when $n \neq 1$);

$$S_n^0(A) = S_n^\emptyset(A) = S_n^\beta(A) + \frac{1}{1-n} \log[\frac{2^{1-n}}{(\tfrac{1}{2} + f_\beta)^n + (\tfrac{1}{2} - f_\beta)^n}]. \tag{112}$$

### 6.3 Zero parity eigenstates

The Rényi entanglement for states $|G_\pm\rangle$ can be computed in different ways. One can use the (109) in the limit $\beta \to 1$. However, similar to section 5.4, it is possible to have an ansatz to write the RDM in the diagonal form of (87). Using this form of RDM one can write the Rényi EE as [49]:

$$S_n^\pm(A) = \frac{1}{1-n} \sum_{j=0}^{l} \log\left((\frac{1+\nu_j}{2})^n + (\frac{1-\nu_j}{2})^n\right) - \log 2. \tag{113}$$

In the above $\nu^\pm$ are the eigenvalues of correlation matrix $\mathbf{\Gamma}^\pm$. The above formula simplifies the entanglement studies, however, it is valid for a certain type of the subsystems. The set $A$ should be connected subsystem of the system containing the site 0 (first site). Otherwise, we would not be able to get entanglement from correlations of the system.

It is crucial to mention that form of correlation $\mathbf{\Gamma}^\pm$, in (72) and (73), indicates that the eigenvalues of $\mathbf{\Gamma}_A^\pm$ for any subsystem $0 \in A$ and $L+1 \notin A$ are the same as $\mathbf{\Gamma}_A^0$. This statement means that there is a relation between entanglements $S_n^0(A)$, $S_n^\emptyset(A)$ and $S_n^\pm(A)$;

$$S_n^0(A) = S_n^\emptyset(A) = S_n^\pm(A) + \log 2. \tag{114}$$

And it is correct for any valid choices of $\mathbf{A}$ and $\mathbf{B}$ matrices. In fact, one can extend this argument to any excited state created from $|0\rangle$, $|\emptyset\rangle$ and $|G_\pm\rangle$ with excitation of same modes, as in

$$|\psi\rangle = \prod_{k_j \in \mathbb{E}} \eta_{k_j}^\dagger |0\rangle, \qquad |\phi\rangle = \prod_{k_j \in \mathbb{E}} \eta_{k_j}^\dagger |\emptyset\rangle, \qquad |\chi_\pm\rangle = \prod_{k_j \in \mathbb{E}} \eta_{k_j}^\dagger |G_\pm\rangle.$$

where set $\mathbb{E}$ does not contain mode zero. The general form of correlations for states above can be found in appendix A. Based on the results of that appendix, we conclude that

$$S_n^\psi(A) = S_n^\phi(A) = S_n^{\chi_\pm}(A) + \log 2. \tag{115}$$

## 7 Physical interpretation of the parity-broken state

In this section, we give a simple interpretation of the $|\beta\rangle$ state which helps to understand some simple cases of the results that we presented so far. This sate can be written as follows:

$$|\beta\rangle = \frac{1}{\sqrt{2(1+|\beta|^2)}}\Big((1+\beta)|G_+\rangle + (1-\beta)|G_-\rangle\Big). \tag{116}$$

In the spin representation the above state can be written in an interesting form. Consider $\delta_+ = 1$ then we can write

$$|\beta\rangle = \frac{1}{\sqrt{2(1+|\beta|^2)}}\Big((1+\beta)|+\rangle_0\,|\phi_{++}\rangle\,|+\rangle_{L+1} + (1-\beta)|-\rangle_0\,|\phi_{--}\rangle\,|-\rangle_{L+1}\Big), \tag{117}$$

where $|\phi_{--}\rangle$ and $|\phi_{++}\rangle$ are normalized states. The above form suggests that the whole system can be considered as two qubit with one qubit at site $0$ $(L+1)$ and the other the rest of the system. Interestingly, the entanglement structure of these three parts is independent of the size of the system and one can easily calculate for example the entanglement of the site $0$ $(L+1)$ with the rest, i.e. $S_n(0)$ $(S_n(L+1))$;

$$S_n(0) = S_n(L+1) = \frac{1}{1-n}\log[(\tfrac{1}{2}+f_\beta)^n + (\tfrac{1}{2}-f_\beta)^n], \tag{118}$$

which is consistent with the equation (111). We note that one can generalize the above argument for all the $|\beta\rangle$ states that can be made out of the eigenstates. The extension to $\delta_+ = -1$ is straightforward. Finally, there is one extra piece that one can add to this story by considering the following mixed state:

$$\rho^\beta = \frac{1}{2(1+|\beta|^2)}\Big(|1+\beta|^2\,|G_+\rangle\,\langle G_+| + |1-\beta|^2\,|G_-\rangle\,\langle G_-|\Big). \tag{119}$$

One can easily show that the reduced density matrix of the above state is exactly equal to the reduced density matrix of the $|\beta\rangle$ state which is guarantied because of the especial form of the state. This makes the preparation of states with the desired reduced density matrix very easy. However, it is clear that in the mixed state scenario the von Neumann entropy does not have entanglement interpretation anymore.

## 8 Examples

In this section, we provide a couple of examples to show how the general results that we derived can be applied in specific cases. The first example which we are able to do the entire calculation analytically is the Hamiltonian (5) with $A = B = 0$. There are a few good reasons to study this Hamiltonian. First of all, for this Hamiltonian one can follow all the calculations analytically and show the validity of all the presented results. Second, it is a Hamiltonian that can be used to diagonalize a Hermitian Hamiltonian with linear creation and annihilation operators which makes it worth studying. Finally interestingly the entanglement structure that emerges from this Hamiltonian is entirely universal. In other words the general Hamiltonians with generic parameters

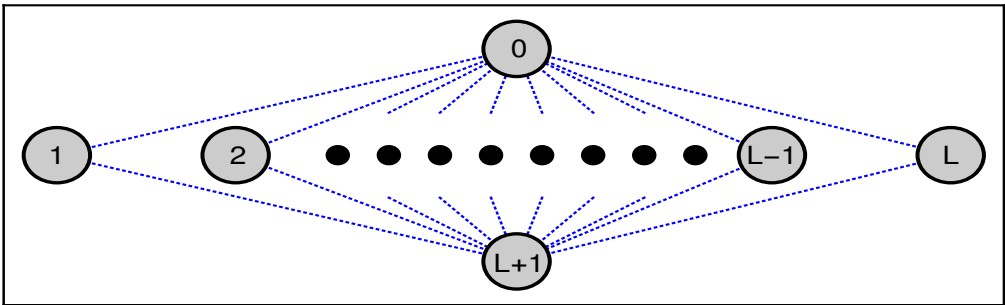

**Figure 2:** In this figure the type of interaction among fermions at different sites, for $\mathbf{A} = \mathbf{B} = 0$ is demonstrated. As one can see, there is no particle hoping among sites 1 to $L$ and the only allowed hoping is from sites 0 or $L + 1$ with the rest of sites.

end up to have similar entanglement structure. This is shown in the example of XY chain with arbitrary boundary magnetic fields which is the second example of this section. The boundary conformal entanglement entropy at the critical point in this case has been studied already in [49], however, here we are more concentrated on general aspects of entanglement with respect to the $\beta$ parameter and boundary magnetic fields.

## 8.1   $A = B = 0$

We are going to focus on the entanglement properties here of the case where the $\mathbf{A}$ and $\mathbf{B}$ matrices are zero. The coupling of fermions in the system has been demonstrated in figure 2. A detailed study is presented in Appendix D for diagonalization and correlations of this special case.

For the vacuum state and ZME state, which Wick theorem is applicable as it was stated in section 6.1, we can relate the Rényi entanglement to the eigenvalues of correlation matrix $\Gamma$ introduced in (57). In a special type of system bipartition, we could also use the $\Gamma^{\pm}$ to find entanglement for the $|G_{\pm}\rangle$ case. For any connected subsystem which contains the site 0 (or site $L + 1$), the (positive valued) eigenvalues of correlation matrices $\Gamma^0$, $\Gamma^{\emptyset}$ and $\Gamma^{\pm}$ are given by the set:

$$\{\nu\} = \begin{cases} \{0, \sqrt{1 - \frac{(\sum_{j=1}^{\ell} |\alpha_j|^2)(\sum_{j=\ell+1}^{L} |\alpha_j|^2)}{(\sum_{j=1}^{L} |\alpha_j|^2)^2}}, 1, \cdots, 1\} & \ell \leq L, \\ \{1, \cdots\cdots, 1\} & \ell = L + 1. \end{cases} \tag{120}$$

The fact that we get the same eigenvalues for each of the states above was explained in section 6. Based on the form of correlation matrices similar (up to a zero eigenvalue) eigenvalues was expected. Using the eigenvalues above the entanglement will be given by ($\ell \leq L$)

$$\begin{aligned} S_n^0(\ell) &= S_n^{\emptyset}(\ell) = \frac{1}{1-n} \log \Big( \big( \frac{1}{2} + \frac{1}{2} \sqrt{1 - \frac{(\sum_{j=1}^{\ell} |\alpha_j|^2)(\sum_{j=\ell+1}^{L} |\alpha_j|^2)}{(\sum_{j=1}^{L} |\alpha_j|^2)^2}} \big)^n \\ &\quad + \big( \frac{1}{2} - \frac{1}{2} \sqrt{1 - \frac{(\sum_{j=1}^{\ell} |\alpha_j|^2)(\sum_{j=\ell+1}^{L} |\alpha_j|^2)}{(\sum_{j=1}^{L} |\alpha_j|^2)^2}} \big)^n \Big) + \log 2, \end{aligned} \tag{121}$$

$$S_n^{\pm}(\ell) = S_n^0(\ell) - \log 2. \tag{122}$$

In case where all the $\alpha_i$'s are constant, we would get

$$S_n^0(\ell) = \frac{1}{1-n} \log \Big( \big( \tfrac{1}{2} + \tfrac{1}{2} \sqrt{\tfrac{L^2 - \ell L + \ell^2}{L^2}} \big)^n + \big( \tfrac{1}{2} - \tfrac{1}{2} \sqrt{\tfrac{L^2 - \ell L + \ell^2}{L^2}} \big)^n \Big) + \log 2, \tag{123}$$

In the thermodynamic limit ($L \to \infty$), we simply get $S_n^0(\ell) = S_n^\emptyset(\ell) = \log 2$ and $S_n^\pm(\ell) = 0$.

For any connected bipartition of system with length $\ell$ which does not contain the $0^{\text{th}}$ and $L+1^{\text{th}}$ sites, eigenvalues of $\mathbf{\Gamma}^0$ and $\mathbf{\Gamma}^\emptyset$ are given by

$$
\{\nu\} = \begin{cases} \{\frac{(\sum_{k=\ell+1}^{L} |\alpha_k|^2)}{(\sum_{k=1}^{L} |\alpha_k|^2)^2}, 1, 1, \cdots, 1\}; & \ell \leq L, \\ \{0, 1, 1, \cdots, 1\}; & \ell = L + 1. \end{cases}
\tag{124}
$$

With this type of bipartition for the system, then the correlation matrix can be used to calculate the entanglement for states $|0\rangle$ and $|\emptyset\rangle$. While for the state $|G_\pm\rangle$ we can not, despite the fact that the $\mathbf{\Gamma}^\pm$ has the same eigenvalues.

In the case of general $\beta$-state, and when $\alpha_i \in \mathbb{R}$, the von Neumann entanglement will be:

$$
\underset{vN}{S^\beta}(A) = -\tfrac{1}{2} \log\Big[(\tfrac{1}{4} - f_\beta)\frac{\Sigma_l \Sigma_{L-l}}{4\Sigma_L^2}\Big] - f_\beta \log\Big(\frac{1 + 2f_\beta}{1 - 2f_\beta}\Big)
$$
$$
- \frac{\sqrt{\Sigma_L^2 - \Sigma_l \Sigma_L + \Sigma_l^2}}{2\Sigma_L} \log\Big[\frac{\Sigma_L + \sqrt{\Sigma_L^2 - \Sigma_l \Sigma_L + \Sigma_l^2}}{\Sigma_L - \sqrt{\Sigma_L^2 - \Sigma_l \Sigma_L + \Sigma_l^2}}\Big].
\tag{125}
$$

Where in the above $\Sigma_l = \sum_{j=1}^{l} \alpha_j^2$ and $f_\beta = \frac{\text{Re}[\beta]}{1 + |\beta|^2}$. It is particularly interesting to see the behavior of entanglement with respect to the $\beta$. As it was mentioned previously, parameter $\beta$ can be thought as a parameter which breaks the parity continuously. In figure 3, we have demonstrated a typical behavior of EE for different values of $\beta$ in complex plain when all $\alpha_i$'s are real constants. As you can see, for real values of $\beta$, the entanglement is maximum for $\beta = 0, \infty$ and minimum for $\beta = \pm 1$, which the first one corresponds to vacuum and ZME state and second one is the $|G_\pm\rangle$. On the other hand, for the purely imaginary $\beta$, entanglement is constant and equal to $S_2^0(l)$ for any value of $\beta$ which is not a trivial observation.

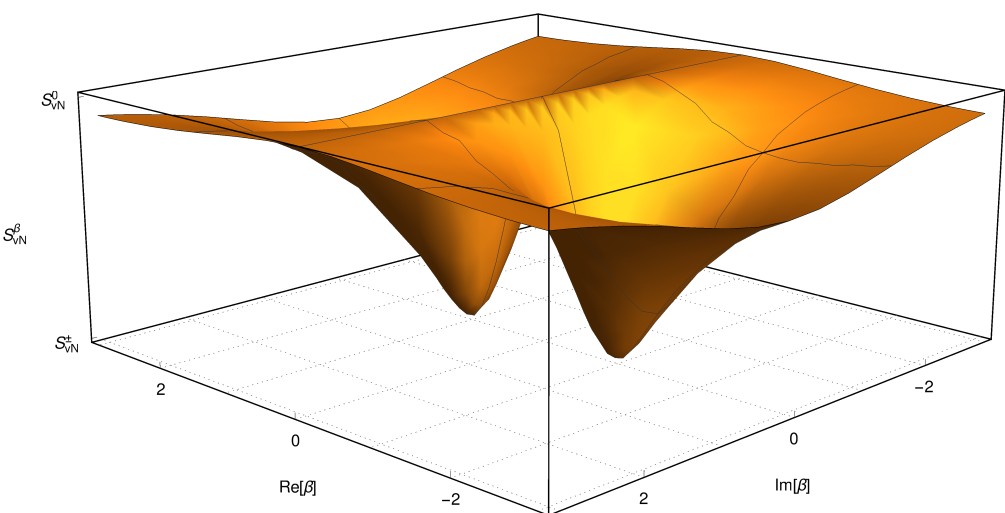

**Figure 3:** Plot of von Neumann entanglement entropy with respect to $\beta$. We already expected that the entanglement to be minimum for the case of $\beta = \pm 1$ which is apparent in the figure. In the case where $\beta$ is purely imaginary, we notice that the EE is again maximum and would not change. This figure is valid for a connected subsystem starting from site 0, for a general set of couplings in the Hamiltonian and size of the full system and subsystem.

## 8.2 Modified XY chain with boundary magnetic field

In this section, we consider the modified transverse field XY chain with arbitrary direction of the boundary magnetic field. We are interested in the following XY Hamiltonian [49] with open boundary conditions

$$
\begin{aligned}
\mathcal{H}^{XY} =& J \sum_{i=1}^{L-1} \Big[ \frac{1+\gamma}{4} \sigma_i^x \sigma_{i+1}^x + \frac{1-\gamma}{4} \sigma_i^y \sigma_{i+1}^y \Big] - \frac{h}{2} \sum_{i=1}^{L} \sigma_i^z \\
&+ \frac{1}{2} \big( b_{1,x} \sigma_0^x \sigma_1^x + b_{1,y} \sigma_0^x \sigma_1^y + b_{1,z} \sigma_1^z + b_{L,x} \sigma_L^x \sigma_{L+1}^x + b_{L,y} \sigma_L^y \sigma_{L+1}^x + b_{L,z} \sigma_L^z \big)
\end{aligned}
\tag{126}
$$

where $\overrightarrow{b}_1 = (b_1 \sin\theta_1 \cos\varphi_1, b_1 \sin\theta_1 \sin\varphi_1, b_1 \cos\theta_1)$ and $\overrightarrow{b}_L = (b_L \sin\theta_1 \cos\varphi_1, b_L \sin\theta_1 \sin\varphi_1, b_L \cos\theta_1)$ are constant vectors. Form of $\mathcal{H}^{XY}$ suggests that we can diagonalize and find the eigenstates analytically. Using the Jordan-Wigner transformation

$$
c_l^\dagger = \prod_{j=0}^{l-1} \sigma_j^z \sigma_l^+,
\tag{127}
$$

where $\sigma_n^\pm = \frac{\sigma_n^x \pm i\sigma_n^y}{2}$, we can map the Hamiltonian (126) to the free fermion Hamiltonian (8), with properly chosen matrices **M** (see appendix E for a demonstration of this matrix). The diagonalization process has been explained in details in section 3.3. The above Hamiltonian is related to an XY chain with boundary magnetic fields as:

$$
H^{XY} = J \sum_{i=1}^{L-1} \Big[ \frac{1+\gamma}{4} \sigma_i^x \sigma_{i+1}^x + \frac{1-\gamma}{4} \sigma_i^y \sigma_{i+1}^y \Big] - \frac{h}{2} \sum_{i=1}^{L} \sigma_i^z + \frac{1}{2} ( \overrightarrow{b}_1 . \overrightarrow{\sigma}_1 + \overrightarrow{b}_L . \overrightarrow{\sigma}_L ).
\tag{128}
$$

In the above expression, we can think of $\overrightarrow{b}_1$ and $\overrightarrow{b}_L$ as magnetic fields at boundaries of our spin chain. If we add two auxiliary spins at positions 0 and $L+1$, if they can only interact with the $x$ component of spins at position 1 and $L$, then we get (126). It is apparent that $\mathcal{H}^{XY}$ commutes with $\sigma_0^x$ and $\sigma_{L+1}^x$. Due to this fact it is possible to divide the Hilbert space of $\mathcal{H}^{XY}$ in four distinct sectors, labeled by the eigenvalues of $\sigma_0^x$ and $\sigma_{L+1}^x$. Therefore, the eigenstates of Hamiltonian (128) would be found in one of these sectors. This procedure has been explained in section 3.6.

### 8.2.1 Determination of BMF Hamiltonian eigenstates

The eigenstates of Hamiltonian (128) can be found with a correct projection in the Hilbert space of (126), which was mentioned in the section (3.6). Here, we focus on the XY-chain case and present (apparent) patterns to find the ground state of (128). Based on results of (52) and (53), one can identify the ground state of BMF spin chain in $|G_+\rangle$ or $\eta_{min}^\dagger |G_-\rangle$ by finding the $\delta_+$. Numerical investigations suggests that the value of $\delta_+$ does not depend on the parameters such as $\gamma, h$ (fixing $J = -1$) and strength of the magnetic field at boundaries. The size of the full system however changes the value of $\delta_+$; For equal directions of boundary fields ( $\overrightarrow{b}_1$ and $\overrightarrow{b}_L$) we get:

$$
\begin{cases}
\delta_+ = -1 & L \text{ even,} \\
\delta_+ = +1 & L \text{ odd.}
\end{cases}
\tag{129}
$$

Nonetheless, the change in the directions of BMF's at ends of the chain can affect the value of $\delta_+$. Figure 4 shows the $\delta_+$ with respect to the change in the azimuthal and polar angle of direction of the magnetic field at boundaries. If one eliminates the BMF at one end of the system (either by putting $\theta_1 = 0, \pi$ or $b_1 = 0$), then changing the direction or strength of BMF at the other end would not affect the $\delta_+$ outcome.

To be able to use the relation (81), one need to have the unitary transformation **U**, which diagonalizes the Hamiltonian, in the form given by (16). Necessary condition is to make sure that eigenstates of **M** corresponding to zero modes are written correctly, they are orthogonal and satisfy the (14). We have already introduced the zero mode of the system in section 3.2, which explains the 2-fold degeneracy of the ground state. However, based on some values of coupling parameters (such as uniform external magnetic field or direction of boundary fields) there could arise more zero modes in the spectrum of **M** matrix. For instance, when $\gamma = 1$ and $\varphi_1, \varphi_L = \frac{\pi}{2}$ (and general bulk and boundary magnetic fields $h$, $\overrightarrow{b_1}$, $\overrightarrow{b_L}$), we would have extra zero mode and more degeneracy. For this values, the extra zero modes eigenvectors of **M** would be (besides (12))

$$
|u_0^3\rangle = \frac{1}{\sqrt{4+2\zeta_1^2+2\zeta_L^2}}
\begin{pmatrix}
-i\zeta_1 \\
1 \\
0 \\
\vdots \\
0 \\
-1 \\
i\zeta_L \\
i\zeta_1 \\
1 \\
0 \\
\vdots \\
0 \\
1 \\
i\zeta_L
\end{pmatrix},
\qquad
|u_0^4\rangle = \frac{1}{\sqrt{4+2\zeta_1^2+2\zeta_L^2}}
\begin{pmatrix}
-i\zeta_1 \\
1 \\
0 \\
\vdots \\
0 \\
1 \\
-i\zeta_L \\
i\zeta_1 \\
1 \\
0 \\
\vdots \\
0 \\
-1 \\
-i\zeta_L
\end{pmatrix};
\qquad (130)
$$

where $\zeta_{1,L} = \frac{h-b_{1,L}\cos\theta_{1,L}}{b_{1,L}\sin\theta_{1,L}}$. This is an exact zero mode which comes from the fact that BMF's does not have any component in the $x$-direction. In general, there could be more zero modes for large sizes. It does not seem easy to identify analytically all the points where we face degeneracy more than the two mentioned. To study entanglement, one should be careful with the zero modes. As an example, numerical investigations suggests that for $L \gg 1$, we would expect one more zero mode for $h > 1$ and independent from BMF values.

### 8.2.2 Entanglement studies

Regarding the entanglement, we have looked into the entanglement properties of parity broken state $|\beta\rangle$. For instance, the behavior of von Neumann entanglement entropy is plotted with respect

to the parameter $\beta$ in the figure 5 for fixed and equal angles of boundary fields. Looking to the discussion of section 6.2, the entanglement entropy obeys the relation

$$S_n^0(A) = S_n^\emptyset(A) = S_n^\beta(A) + \frac{1}{1-n} \log[\frac{2^{1-n}}{(\frac{1}{2}+f_\beta)^n + (\frac{1}{2}-f_\beta)^n}], \qquad f_\beta = \frac{\text{Re}[\beta]}{1+|\beta|^2}. \qquad (131)$$

Since the relation above is general and does not depend on parameters of the model, it is expected to see the same behavior of entanglement entropy with the change in the $\beta$ as in figure 3.

For most parts of this article, we are interested in case where the boundary magnetic fields are the same in both edges, i. e., $\vec{b}_1 = \vec{b}_L$. Another compelling observation would be the effect of the boundary field direction (same in both ends) on the entanglement of a particular state such as $|G_+\rangle$. Results have been demonstrated in figure 6. For specific angles of BMF in the $xy$-plane, we observe a huge change in the value of entanglement entropy.

For the small magnetic field ($h < J$ as in figure 6a), there are two degenerate ground states for BMF Ising model: one can write $\frac{1}{2}(|\rightarrow\rightarrow\cdots\rightarrow\rangle + |\leftarrow\leftarrow\cdots\leftarrow\rangle)$ as the ground state. When $\varphi \approx 0$, the first spin prefers $|\rightarrow\rangle$ over the other possibility, $|\leftarrow\rangle$, which lowers the boundary entanglement [4] . As $\varphi \to \frac{\pi}{2}, \frac{3\pi}{2}$, both possible state of $x$-spin for the first spin would be equally probable, since the boundary magnetic field aligns the first spin in the (positive or negative) $y$ direction. This would result in increase in entanglement. As the angle $\theta$ increases, the intensity of BMF interaction $\vec{b}_1.\vec{S}_1$ opposes the uniform magnetic field $h$ in the system. So, we would expect smaller jumps in the entanglement for bigger $\theta$.

In the large magnetic fields ($h > J$ as in figure 6c or the paramagnetic state) the ground state is not degenerate, being all spins almost aligned in the direction of $h$. When BMF in the xy-plan is small ($\theta \approx 0, \pi$), entanglement would not change much by change in $\varphi$. Although almost constant, entanglement is a bit higher for small $\theta \sim 0$ rather than $\theta \sim \pi$. As the intensity of BMF in xy-plan increases (for example $\theta = \frac{\pi}{4}$), first few spins at the beginning of the chain would get out of all parallel positioning and the entanglement changes with the angle of $\varphi$.

# 9   Conclusions

In this paper we investigated a generic quantum spin chain Hamiltonian with arbitrary boundary magnetic fields. As far as the bulk Hamiltonian can be mapped to free fermions even though the boundary terms are not quadratic with respect to free fermions we were able to diagonalize the Hamiltonian exactly. This was done using two ancillary extra sites and later projection of the eigenstates. The extended Hamiltonian was studied in depth and many properties of the Hamiltonian was studied including eigenstates in configuration bases, the correlation function of eigenstates and the reduced density matrix. The extended Hamiltonian always has at least one zero mode which guaranties the presence of degeneracy. To get the eigenstates of the original spin chain one needs to go to the sector in which the parity number symmetry is broken. We studied comprehensively these eigenstates and found the correlation functions, reduced density matrix and the entanglement entropy. Interestingly the general features are independent of the parameters of the Hamiltonian and one can get universal results for the reduced density matrix

---

[4]Size of the subsystem has considered to be small so effect of boundary entanglement be more apparent

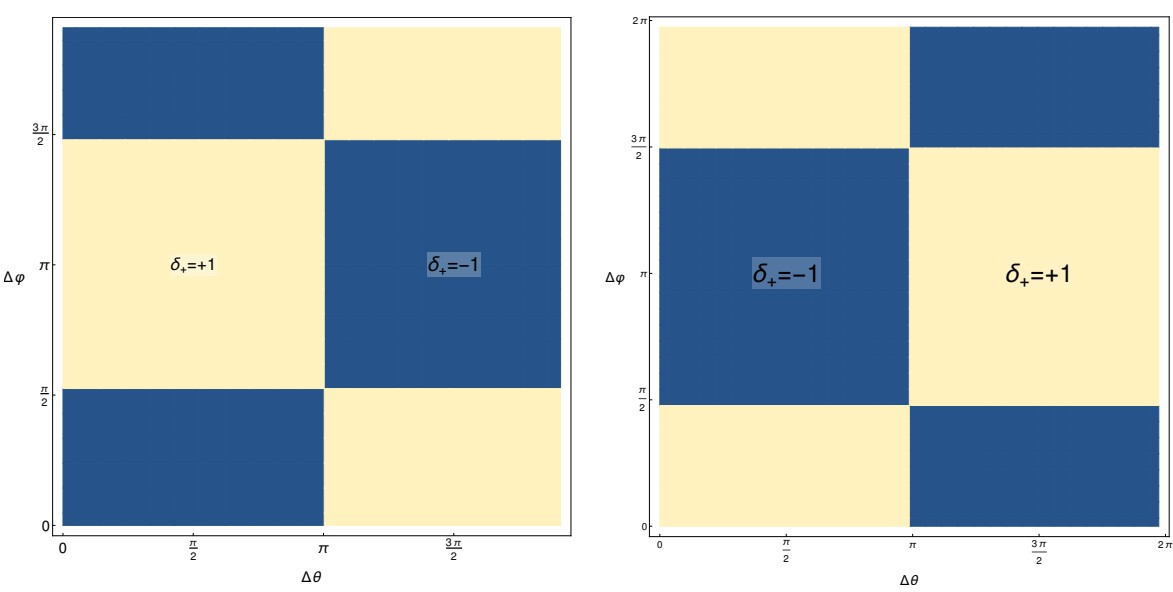

**Figure 4:** Different values of $\delta_+$ for difference in the directions of boundary fields $\vec{b}_1 = (\sin\theta_1\cos\varphi_1, \sin\theta_1\sin\varphi_1, \cos\theta_1)$ and $\vec{b}_L = (\sin\theta_L\cos\varphi_L, \sin\theta_L\sin\varphi_L, \cos\theta_L)$. We have defined $\Delta\varphi = \varphi_L - \varphi_1$ and $\Delta\theta = \theta_L - \theta_1$. The direction of boundary field at the first site is fixed by $\theta_1 = \varphi_1 = \frac{\pi}{200}$ and we change the angles at end point of the chain. On the right, we have $L = 9$ and left hand side corresponds to $L = 10$. Rest of the parameters are: $J = -1$, $\gamma = 1/2$ and $h = 0.518$.

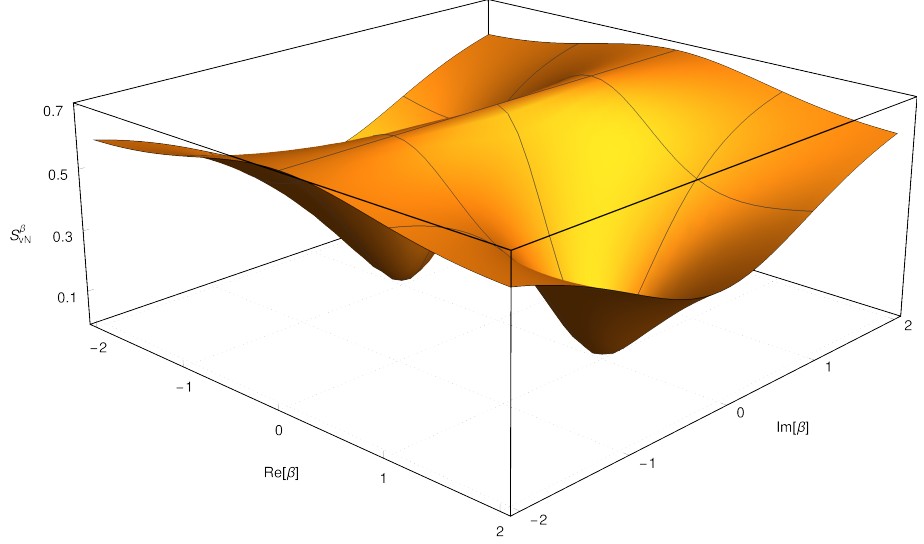

**Figure 5:** In above, the typical behavior of von Neumann entropy with respect to parameter $\beta$ is plotted for a subsystem of length $\ell$ starting from boundary of the chain. Size of the full system is $L = 30$ and subsystem is $l = 14$. Rest of the parameters are: $J = \gamma = 1$, $h = \frac{1}{2}$, $b_{L,x,y} = 1$, $b_{1,x,y} = 1$ and $b_{1,L,z} = 0$.

and entanglement for generic eigenstates. The procedure used here can be extended for the other similar situations such as local breaking of the parity number symmetry in quantum spin chains due to local magnetic field impurity. It can be also useful to study local quantum quenches in quantum spin chains.

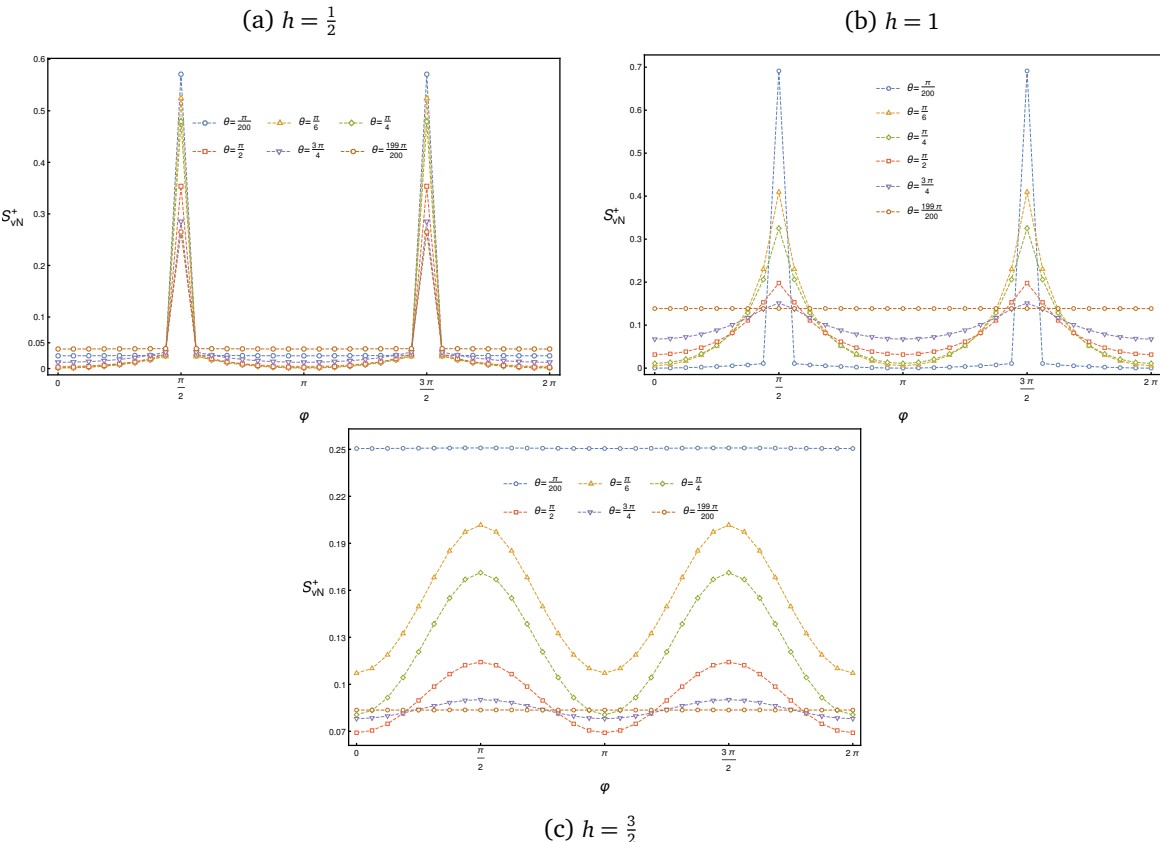

**Figure 6:** In this figure, the von Neumann entropy is plotted with respect to the angles of BMF, for three values of magnetic field, $h = \frac{1}{2}$, $1$, $\frac{3}{2}$ respectively. We have put $b_{1,x} = b_{L,x} = \sin(\theta)\cos(\phi)$, $b_{1,y} = b_{L,y} = \sin(\theta)\sin(\phi)$ and $b_{1,z} = b_{L,z} = \cos(\theta)$. Rest of parameters are: $L = 30$, $l = 2$, $\beta = +1$, $J = -1$ and $\gamma = 1$. To better manifest the change in the entanglement entropy, size of the subsystem is selected to be close to the beginning of the chain. We observe that there is a change in the entanglement at $\varphi = \frac{\pi}{2}, \frac{3\pi}{2}$. As mentioned before, there is (one) more zero mode in the eigenvalues of **M** matrix in (9) for $h > 1$. However, this is not an exact zero mode and for size $L = 30$, it is small but not zero. Therefore, observation above is intact.

# Acknowledgements

MAR thanks CNPq and FAPERJ (grant number 210.354/2018) for partial support. MAR thanks ICTP for its hospitality during the visit which part of this work is done. The work done by AJ has been supported by CNPq.

# Appendix A  Correlations for excited quasiparticle eigenstates

In section 4, we introduced the correlation functions and the method to produce these matrices for the ground state. In this part, we are going to present the method to calculate the correlations for excited quasiparticles (31) using the unitary transformation $\mathbf{U}$ which diagonalize the Hamiltonian.

The quasiparticle excited state such as $|\psi\rangle$ is defined by

$$|\psi\rangle = |k_1, k_2, \cdots, k_N\rangle = \prod_{k_j \in \mathbb{E}} \eta_{k_j}^\dagger |0\rangle_\eta \tag{A.1}$$

where set $\mathbb{E}$ could be any subset of modes. For this state, we start by calculating the $\langle\psi| c_i^\dagger c_j |\psi\rangle$ and $\langle\psi| c_i^\dagger c_j^\dagger |\psi\rangle$

$$\langle c_i^\dagger c_j\rangle_\psi = \langle 0| \prod_{k_j \in \mathbb{E}} \eta_{k_j} \sum_{k,l}(h_{li}^*\eta_l + g_{ki}\eta_k^\dagger)\sum_{n,m}(g_{mj}^*\eta_m + h_{nj}\eta_n^\dagger)\prod_{k_j \in \mathbb{E}}\eta_{k_j}^\dagger |0\rangle, \tag{A.2}$$

$$\langle c_i^\dagger c_j^\dagger\rangle_\psi = \langle 0| \prod_{k_j \in \mathbb{E}} \eta_{k_j} \sum_{k,l}(h_{li}^*\eta_l + g_{ki}\eta_k^\dagger)\sum_{n,m}(h_{mj}^*\eta_m + g_{nj}\eta_n^\dagger)\prod_{k_j \in \mathbb{E}}\eta_{k_j}^\dagger |0\rangle. \tag{A.3}$$

Using the Wick theorem we can simplify these expressions to get:

$$\langle c_i^\dagger c_j\rangle_\psi = (\mathbf{h}^\dagger.\mathbf{h})_{i,j} + \sum_{k_j \in \mathbb{E}}(g_{k_j,i}g_{k_j,j}^* - h_{k_j,i}^* h_{k_j,j}),$$
$$\langle c_i^\dagger c_j^\dagger\rangle_\psi = (\mathbf{h}^\dagger.\mathbf{g})_{i,j} + \sum_{k_j \in \mathbb{E}}(g_{k_j,i}h_{k_j,j}^* - h_{k_j,i}^* g_{k_j,j}). \tag{A.4}$$

Therefore, using the definitions (58), we can write the $\mathbf{K}$, $\bar{\mathbf{K}}$ and $\mathbf{G}$ for the state $|\psi\rangle$ in a short version as

$$\mathbf{K}_{ij}^\psi = (\mathbf{h}^\dagger + \mathbf{g}^\dagger).(\mathbf{h} + \mathbf{g})_{ij} - 2i\Im\Big[\sum_{k_j \in \mathbb{E}}(h_{k_j,i}^* + g_{k_j,i}^*)(g_{k_j,j} + h_{k_j,j})\Big], \tag{A.5}$$

$$\bar{\mathbf{K}}_{ij}^\psi = (\mathbf{h}^\dagger - \mathbf{g}^\dagger).(\mathbf{h} - \mathbf{g})_{ij} - 2i\Im\Big[\sum_{k_j \in \mathbb{E}}(h_{k_j,i}^* - g_{k_j,i}^*)(h_{k_j,j} - g_{k_j,j})\Big], \tag{A.6}$$

$$\mathbf{G}_{ij}^\psi = (\mathbf{h}^\dagger - \mathbf{g}^\dagger).(\mathbf{h} + \mathbf{g})_{ij} + 2\Re\Big[\sum_{k_j \in \mathbb{E}}(g_{k_j,i}^* - h_{k_j,i}^*)(h_{k_j,j} + g_{k_j,j})\Big]. \tag{A.7}$$

The prior expressions are useful for the study of entanglement in excited quasiparticle states. One can use the $\mathbf{\Gamma}^\psi$ to calculate the RDM for a given subsystem $A$ using the relation

$$\rho_A^\psi(\gamma, \bar{\gamma}) = [\det\frac{\mathbf{I} - \mathbf{\Gamma}_A^\psi}{2}]^{\frac{1}{2}} e^{\frac{1}{4}\begin{pmatrix}\gamma & \bar{\gamma}\end{pmatrix}\ln\frac{\mathbf{I}+\mathbf{\Gamma}_A^\psi}{\mathbf{I}-\mathbf{\Gamma}_A^\psi}\begin{pmatrix}\gamma \\ \bar{\gamma}\end{pmatrix}}, \tag{A.8}$$

where $\gamma$ and $\bar{\gamma}$ are the Majorana fermions defined in section 4. Equivalently, it is possible to use the preceding correlations to find the $\mathbf{R}^\psi$ matrix (as in (32)), $\mathbf{R}^\psi = \mathbf{F}^{\psi*}(\mathbf{I} - \mathbf{C}^\psi)^{-1}$, which could lead to finding the RDM.

For excited states created from ZME state, we can do the same calculations. These quasiparticle excited state are created as:

$$\left| \phi_{\emptyset} \right\rangle = |n_1, n_2, \cdots, n_N\rangle = \prod_{n_j \in \mathbb{E}} \eta_{n_j}^{\dagger} |\emptyset\rangle , \tag{A.9}$$

where $n_j \neq 0$. Having the correlation matrices for the above states allows us to study the excited state entanglement for these states likewise. To write such matrices, we can use the result of section 4.2. Correlations for the family of ZME excited states are demonstrated in 40. The matrices $\mathbf{K}^{\phi_{\emptyset}}$ and $\bar{\mathbf{K}}^{\phi_{\emptyset}}$ have the same form as (A.5) and (A.6), therefore, we have not included their forms.

As it was explained in subsection 3.6, one could divide the Hilbert space of Hamiltonian (5) into four different towers which one of the these towers or sectors corresponds to the eigenstates of boundary magnetic field Hamiltonian (1). We devote this part to find the correlation functions for states in a given tower, given the ground state for the tower (sector), can be found with $|G_{\pm}\rangle$. The base of calculation is similar to the above cases, therefore here we only hand out the final result. For start, an excited state in one of the sectors has the form

$$\left| \chi_{\pm} \right\rangle = \prod_{m_j \in \mathbb{E}} \eta_{m_j}^{\dagger} \left| G_{\pm} \right\rangle . \tag{A.10}$$

In the preceding expression, set $\mathbb{E}$ contains excited modes acting on $\left| G_{\pm} \right\rangle$ and $0 \notin \mathbb{E}$. For a general case of $\mathbb{E}$ we can write the correlations as in page 41. Similar to the ZME and the $|G_{\pm}\rangle$ case, the matrices $\mathbf{K}^{\chi_{\pm}}$ and $\bar{\mathbf{K}}^{\chi_{\pm}}$ have the same form as $\mathbf{K}^{\psi}$ and $\bar{\mathbf{K}}^{\psi}$ in (A.5) and (A.6).

$$
\mathbf{C}^{\phi_0} = \begin{pmatrix}
\frac{1}{2} & C^\emptyset_{0,j} + \sum_{n\in\mathbb{E}}\left(g_{n,0}g^*_{n,j} + g^*_{n,0}h_{n,j}\right) & C^\emptyset_{0,L+1} + \sum_{n\in\mathbb{E}}\left(g_{n,0}g^*_{n,L+1} + g^*_{n,0}g_{n,L+1}\right) \\[2mm]
C^\emptyset_{i,0} + \sum_{n\in\mathbb{E}}\left(g_{n,i}g^*_{n,0} + h^*_{n,i}g_{n,0}\right) & C^\emptyset_{i,j} + \sum_{n\in\mathbb{E}}\left(g_{n,i}g^*_{n,j} - h^*_{n,i}h_{n,j}\right) & C^\emptyset_{i,L+1} + \sum_{n\in\mathbb{E}}\left(g_{n,i}g^*_{n,L+1} - h^*_{n,i}g_{n,L+1}\right) \\[2mm]
C^\emptyset_{L+1,0} + \sum_{n\in\mathbb{E}}\left(g_{n,L+1}g^*_{n,0} + g^*_{n,L+1}g_{n,0}\right) & C^\emptyset_{L+1,j} + \sum_{n\in\mathbb{E}}\left(g_{n,L+1}g^*_{n,j} - g^*_{n,L+1}h_{n,j}\right) & \frac{1}{2}
\end{pmatrix},
\tag{A.11}
$$

$$
\mathbf{F}^{\phi_0} = \begin{pmatrix}
0 & F^\emptyset_{0,j} + \sum_{n\in\mathbb{E}}\left(g_{n,0}h^*_{n,j} + g^*_{n,0}g_{n,j}\right) & F^\emptyset_{0,L+1} + \sum_{n\in\mathbb{E}}\left(g_{n,0}g^*_{n,L+1} + g^*_{n,0}g_{n,L+1}\right) \\[2mm]
F^\emptyset_{i,0} - \sum_{n\in\mathbb{E}}\left(g_{n,i}g^*_{n,0} + h^*_{n,i}g_{n,0}\right) & F^\pm_{i,j} + \sum_{n\in\mathbb{E}}\left(g_{n,i}h^*_{n,j} - h^*_{n,i}g_{n,j}\right) & F^\emptyset_{i,L+1} + \sum_{n\in\mathbb{E}}\left(g_{n,i}g^*_{n,L+1} - g^*_{n,i}g_{n,L+1}\right) \\[2mm]
F^\emptyset_{L+1,0} - \sum_{n\in\mathbb{E}}\left(g_{n,L+1}g^*_{n,0} + g^*_{n,L+1}g_{n,0}\right) & F^\emptyset_{L+1,j} + \sum_{n\in\mathbb{E}}\left(g_{n,L+1}h^*_{n,j} - g^*_{n,L+1}g_{n,j}\right) & 0
\end{pmatrix}
\tag{A.12}
$$

$$
\mathbf{G}^{\phi_0} = \begin{pmatrix}
0 & G^\emptyset_{0,j} + 4\Re\left[\sum_{n\in\mathbb{E}} g^*_{n,0}(h_{n,j}+g_{n,j})\right] & G^\emptyset_{0,L+1} + 8\Re\left[\sum_{n\in\mathbb{E}} g^*_{n,0}g_{n,L+1}\right] \\[2mm]
0 & G^\emptyset_{i,j} + 2\Re\left[\sum_{n\in\mathbb{E}}(g^*_{n,0}-h^*_{n,0})(h_{n,j}+g_{n,j})\right] & G^\emptyset_{i,L+1} + 4\Re\left[\sum_{n\in\mathbb{E}}(g^*_{n,i}-h^*_{n,i})g_{n,L+1}\right] \\[2mm]
1 & 0 & 0
\end{pmatrix}.
\tag{A.13}
$$

$$\mathbf{C}^{\chi_\pm} = \begin{pmatrix} \frac{1}{2} & C_{0,j}^{\pm} + \sum_{m\in\mathbb{E}}\left(g_{m,0}g_{m,j}^* + g_{m,0}^* h_{m,j}\right) & C_{0,L+1}^{\pm} + \sum_{m\in\mathbb{E}}\left(g_{m,0}g_{m,L+1}^* + g_{m,0}^* g_{m,L+1}\right) \\[6pt] C_{i,0}^{\pm} + \sum_{m\in\mathbb{E}}\left(g_{m,i}g_{m,0}^* + h_{m,i}^* g_{m,0}\right) & C_{i,j}^{\pm} + \sum_{m\in\mathbb{E}}\left(g_{m,i}g_{m,j}^* - h_{m,i}^* h_{m,j}\right) & C_{i,L+1}^{\pm} + \sum_{m\in\mathbb{E}}\left(g_{m,i}g_{m,L+1}^* - h_{m,i}^* g_{m,L+1}\right) \\[6pt] C_{L+1,0}^{\pm} + \sum_{m\in\mathbb{E}}\left(g_{m,L+1}g_{m,0}^* + g_{m,L+1}^* g_{m,0}\right) & C_{L+1,j}^{\pm} + \sum_{m\in\mathbb{E}}\left(g_{m,L+1}g_{m,j}^* - g_{m,L+1}^* h_{m,j}\right) & \frac{1}{2} \end{pmatrix}, \tag{A.14}$$

$$\mathbf{F}^{\chi_\pm} = \begin{pmatrix} 0 & F_{0,j}^{\pm} + \sum_{m\in\mathbb{E}}\left(g_{m,0}h_{m,j}^* + g_{m,0}^* g_{m,j}\right) & F_{0,L+1}^{\pm} + \sum_{m\in\mathbb{E}}\left(g_{m,0}g_{m,L+1}^* + g_{m,0}^* g_{m,L+1}\right) \\[6pt] F_{i,0}^{\pm} - \sum_{m\in\mathbb{E}}\left(g_{m,i}g_{m,0}^* + h_{m,i}^* g_{m,0}\right) & F_{i,j}^{\pm} + \sum_{m\in\mathbb{E}}\left(g_{m,i}h_{m,j}^* - h_{m,i}^* g_{m,j}\right) & F_{i,L+1}^{\pm} + \sum_{m\in\mathbb{E}}\left(g_{m,i}g_{m,L+1}^* - g_{m,i}^* g_{m,L+1}\right) \\[6pt] F_{L+1,0}^{\pm} - \sum_{m\in\mathbb{E}}\left(g_{m,L+1}g_{m,0}^* + g_{m,L+1}^* g_{m,0}\right) & F_{L+1,j}^{\pm} + \sum_{m\in\mathbb{E}}\left(g_{m,L+1}h_{m,j}^* - g_{m,L+1}^* g_{m,j}\right) & 0 \end{pmatrix}, \tag{A.15}$$

$$\mathbf{G}^{\chi_\pm} = \begin{pmatrix} 0 & G_{0,j}^{\pm} + 4\Re\left[\sum_{m\in\mathbb{E}}g_{m,0}^*\left(h_{m,j}+g_{m,j}\right)\right] & G_{0,L+1}^{\pm} + 8\Re\left[\sum_{m\in\mathbb{E}}g_{m,0}^* g_{m,L+1}\right] \\[6pt] 0 & G_{i,j}^{\pm} + 2\Re\left[\sum_{m\in\mathbb{E}}\left(g_{m,0}^* - h_{m,0}^*\right)\left(h_{m,j}+g_{m,j}\right)\right] & G_{i,L+1}^{\pm} + 4\Re\left[\sum_{m\in\mathbb{E}}\left(g_{m,i}^* - h_{m,i}^*\right)g_{m,L+1}\right] \\[6pt] 0 & 0 & 0 \end{pmatrix}. \tag{A.16}$$

# Appendix B  Reduced density matrix calculations

In this appendix, we are presenting the calculations of (81) in details. Starting from (77), we can write

$$
\rho_1^\beta(\xi, \xi') = \int \prod_{l\in 2} d\bar{\xi}_l d\xi_l \, e^{-\sum_{n\in 2}\bar{\xi}_n \xi_n} \langle \xi_1, \cdots, \xi_k, -\xi_{k+1}, \cdots, -\xi_L| \rho^\beta \left| \xi'_1, \cdots, \xi'_k, \xi_{k+1}, \cdots, \xi_L \right\rangle
$$

$$
= |C^\beta|^2 e^{\frac{1}{2}(\mathbf{R}_{11})_{ij}\bar{\xi}_i\bar{\xi}_j - \frac{1}{2}(\mathbf{R}^*_{11})_{ji}\xi'_j\xi'_i} \int \prod_{l\in 2} d\bar{\xi}_l d\xi_l \, e^{-\bar{\xi}_n \xi_n} \mathcal{F}(\{\xi_n\}, \{\xi'_i\}, \{\bar{\xi}_j\}, \{\bar{\xi}_m\})
$$

$$
\times \left[ e^{-\frac{1}{2}(\mathbf{R}_{12})_{in}\bar{\xi}_i\bar{\xi}_n - \frac{1}{2}(\mathbf{R}_{21})_{ni}\bar{\xi}_n\bar{\xi}_i} e^{\frac{1}{2}(\mathbf{R}_{22})_{mn}\bar{\xi}_m\bar{\xi}_n} e^{-\frac{1}{2}(\mathbf{R}^*_{22})_{nm}\xi_n\xi_m + \frac{1}{2}(\mathbf{R}^*_{12})_{jm}\xi'_j\xi_m + \frac{1}{2}(\mathbf{R}^*_{21})_{mj}\xi_m\xi'_j} \right],
$$
(B.1)

where in the above Einstein summation convention is used. To clarify the notation above, the indices $i, j$ belong to subsystem **1** and indices $n, m$ to subsystem **2**. Eventually, the function $\mathcal{F}$ is given by

$$
\mathcal{F}(\{\xi_n\}, \{\xi'_i\}, \{\bar{\xi}_j\}, \{\bar{\xi}_m\}) = 1 + \beta(\mathfrak{M}_1)_{0j}\bar{\xi}_j - \beta(\mathfrak{M}_2)_{0m}\bar{\xi}_m + \beta^*(\mathfrak{M}^*_1)_{0i}\xi'_i
$$

$$
+ \beta^*(\mathfrak{M}^*_2)_{0n}\xi_n + |\beta|^2(\mathfrak{M}_1)_{0j}(\mathfrak{M}^*_1)_{0i}\bar{\xi}_j\xi'_i + |\beta|^2(\mathfrak{M}_1)_{0j}(\mathfrak{M}^*_2)_{0n}\bar{\xi}_j\xi_n
$$

$$
- |\beta|^2(\mathfrak{M}_2)_{0m}(\mathfrak{M}^*_1)_{0i}\bar{\xi}_m\xi'_i - |\beta|^2(\mathfrak{M}_2)_{0m}(\mathfrak{M}^*_2)_{0n}\bar{\xi}_m\xi_n.
$$
(B.2)

In (B.1) we have divided $\mathbf{R}$ into four submatrices $\mathbf{R}_{11}$, $\mathbf{R}_{12}$, $\mathbf{R}_{21} = -\mathbf{R}_{12}{}^T$ and $\mathbf{R}_{22}$, according to the part we are tracing out (or not). Although these submatrices do not need to have same size, $\mathbf{R}_{11}$ and $\mathbf{R}_{22}$ should be square matrices. We write the (B.1) in the compact form:

$$
\rho_1(\xi, \xi') = |C^\beta|^2 e^{\frac{1}{2}\left(\bar{\xi} \ \ \xi'\right) \begin{pmatrix} \mathbf{R}_{11} & 0 \\ 0 & -\mathbf{R}^*_{11} \end{pmatrix} \begin{pmatrix} \bar{\xi} \\ \xi' \end{pmatrix}} \int \mathbf{D}\eta \, \mathcal{F}(\xi'_i, \bar{\xi}_j, \eta) e^{\frac{1}{2}\eta^T \mathcal{A}\eta + \lambda^T \eta}.
$$
(B.3)

where $\mathbf{D}\eta = d\bar{\xi} d\xi$ and

$$
\eta^T = \left( \bar{\xi} \ \ \xi \right), \qquad \lambda^T = \left( -\bar{\xi}\mathbf{R}_{12} \ \ -\xi'\mathbf{R}^*_{12} \right),
$$
(B.4)

$$
\mathcal{A} = \begin{pmatrix} \mathbf{R}_{22} & -\mathbf{I} \\ \mathbf{I} & -\mathbf{R}^*_{22} \end{pmatrix}.
$$
(B.5)

It is much easier to solve the integration in (B.3) with this new variables. In the expression of $\mathcal{F}$, the first, second, forth and sixth terms do not depend on the variables of integration. Therefore, for those terms we can write:

$$
\rho_1^\beta(1^{st}, 2^{nd}, 4^{th}, 6^{th}, \xi, \xi') = |C^\beta|(\det[\mathcal{A}])^{\frac{1}{2}}(1 + \beta\mathfrak{M}_1\bar{\xi})(1 + \beta^*\mathfrak{M}^*_1\xi') e^{\frac{1}{2}\left(\bar{\xi} \ \ \xi'\right)\Omega\begin{pmatrix} \bar{\xi} \\ \xi' \end{pmatrix}},
$$
(B.6)

where

$$\boldsymbol{\Omega} = \begin{pmatrix} \mathbf{R}_{11} & 0 \\ 0 & -\mathbf{R}_{11}{}^* \end{pmatrix} + \begin{pmatrix} \mathbf{R}_{12} & 0 \\ 0 & \mathbf{R}_{12}{}^* \end{pmatrix} \mathcal{A}^{-1} \begin{pmatrix} \mathbf{R}_{12}{}^T & 0 \\ 0 & \mathbf{R}_{12}{}^\dagger \end{pmatrix}. \tag{B.7}$$

For the terms with linear Grassmann variables in the integration we first substitute $\eta \to \eta + \mathcal{A}^{-1}\lambda$ in (B.3). Using the Berezin integration techniques in presented in [56], we are left with

$$\rho_1^\beta(3^{\text{rd}}, 5^{\text{th}}, 7^{\text{th}}, 8^{\text{th}}, \xi, \xi') = |C^\beta|(\det[\mathcal{A}])^{\frac{1}{2}}(\det[\mathcal{W}])^{\frac{1}{2}}F(\bar{\xi}_1, \xi'_1)e^{\frac{1}{2}\left(\bar{\xi} \ \ \xi'\right)\boldsymbol{\Omega}\begin{pmatrix} \bar{\xi} \\ \xi' \end{pmatrix}}, \tag{B.8}$$

with

$$\begin{aligned} F(\bar{\xi}_1, \xi'_1) &= |\beta|^2 \text{Pf}[\mathcal{W}] + \begin{pmatrix} -\beta\mathfrak{M}_2 & 0 \\ 0 & \beta^*\mathfrak{M}_2^* \end{pmatrix} \mathcal{A}^{-1}\lambda + |\beta|^2\mathfrak{M}_1\bar{\xi} \times \begin{pmatrix} 0 & 0 \\ 0 & \mathfrak{M}_2^* \end{pmatrix} \mathcal{A}^{-1}\lambda \\[2mm] &\quad + |\beta|^2 \begin{pmatrix} -\mathfrak{M}_2 & 0 \\ 0 & 0 \end{pmatrix} \mathcal{A}^{-1}\lambda \times \mathfrak{M}_1^*\xi' + |\beta|^2 \begin{pmatrix} -\mathfrak{M}_2 & 0 \end{pmatrix} \mathcal{A}^{-1}\lambda \times \begin{pmatrix} 0 & \mathfrak{M}_2^* \end{pmatrix} \mathcal{A}^{-1}\lambda, \end{aligned} \tag{B.9}$$

$$\mathcal{W} = \begin{pmatrix} \mathfrak{M}_2 & 0 \\ 0 & -\mathfrak{M}_2^* \end{pmatrix} \mathcal{A}^{-T} \begin{pmatrix} \mathfrak{M}_2^T & 0 \\ 0 & -\mathfrak{M}_2^\dagger \end{pmatrix}. \tag{B.10}$$

Putting (B.6) and (B.8) together, after some algebraic manipulations, we get to (78).

We could have used another trick to solve the Berezin integrations to get RDM. This method ends up in having two exponentials in the final result. The trick is to write the linear Grassmann variable in the integration as exponentials, like:

$$(\mathfrak{M}\bar{\xi})_{k_1}\cdots(\mathfrak{M}\bar{\xi})_{k_N}(\mathfrak{M}^*\xi')_{k_1}\cdots(\mathfrak{M}^*\xi')_{k_N} = \int \prod_i d\bar{\theta}_i d\theta_i e^{\sum_{k_j} \theta_{k_j}(\mathfrak{M}\bar{\xi})_{k_j} + \bar{\theta}_{k_j}(\mathfrak{M}^*\xi)_{k_j}}. \tag{B.11}$$

In the above relation, $\theta$ and $\bar{\theta}$ are Grassmann variables too.

If we start again from density matrix expression in coherent basis (76), and rewrite it as:

$$\begin{aligned} \rho^\beta(\xi, \xi') &= |C^\beta|^2 e^{\frac{1}{2}R_{ij}\bar{\xi}_i\bar{\xi}_j}(1 + \beta\mathfrak{M}_{0k}\bar{\xi}_k + \beta^*\mathfrak{M}_{0l}^*\xi'_l + |\beta|^2\mathfrak{M}_{0k}\mathfrak{M}_{0l}^*\bar{\xi}_k\xi'_l)e^{-\frac{1}{2}R_{nm}^*\xi'_n\xi'_m} \\ &= \rho^\beta(1, \xi, \xi') + \varrho^\beta(2, \xi, \xi') + \varrho^\beta(3, \xi, \xi') + \rho^\beta(4, \xi, \xi'), \end{aligned} \tag{B.12}$$

where the $\rho^\beta(1, \xi, \xi')$ and $\rho^\beta(4, \xi, \xi')$ are the density matrices corresponding to the vacuum-vacuum state and excited-excited state, and the $\varrho^\beta(2, \xi, \xi')$ and $\varrho^\beta(3, \xi, \xi')$ are density cross terms corresponding to the vacuum-excited (and exited-vacuum) state terms. We are going to partial trace each term separately and then put the results together afterwards. For the first term in (B.12), we have:

$$\rho_1^\beta(1, \xi, \xi') = \frac{C^\beta}{|\beta|^2}e^{\frac{1}{2}\left(\bar{\xi} \ \ \xi'\right)\boldsymbol{\Omega}^\beta\begin{pmatrix} \bar{\xi} \\ \xi' \end{pmatrix}}, \tag{B.13}$$

where again

$$\mathcal{C}^{\beta} = \frac{|\beta|^2 \sqrt{\det\left[\mathbf{I} + \mathbf{R}_{22}^{\dagger}\mathbf{R}_{22}\right]}}{(1 + |\beta|^2)\sqrt{\det\left[\mathbf{I} + \mathbf{R}^{\dagger}\mathbf{R}\right]}}, \qquad \mathcal{A} = \begin{pmatrix} \mathbf{R}_{22} & -\mathbf{I} \\ \mathbf{I} & -\mathbf{R}_{22}^* \end{pmatrix}, \tag{B.14a}$$

$$\boldsymbol{\Omega} = \begin{pmatrix} \mathbf{R}_{11} & 0 \\ 0 & -\mathbf{R}_{11}^* \end{pmatrix} + \begin{pmatrix} \mathbf{R}_{12} & 0 \\ 0 & \mathbf{R}_{12}^* \end{pmatrix} \mathcal{A}^{-1} \begin{pmatrix} \mathbf{R}_{12}^T & 0 \\ 0 & \mathbf{R}_{12}^{\dagger} \end{pmatrix}. \tag{B.14b}$$

For the last term in (B.12), we can write:

$$\rho^{\beta}(4, \xi, \xi') = |C^{\beta}\beta|^2 \int d\bar{\theta}d\theta \, e^{\theta(\mathfrak{M}\bar{\xi}) + \bar{\theta}(\mathfrak{M}^*\xi')} e^{\frac{1}{2}R_{ij}\bar{\xi}_i\bar{\xi}_j} e^{-\frac{1}{2}R_{ij}^*\xi_i'\xi_j'}. \tag{B.15}$$

For the reduced density matrix we get

$$\rho_1^{\beta}(4, \xi, \xi') = |C^{\beta}\beta|^2 \int d\bar{\theta}d\theta \, e^{\frac{1}{2}(\mathbf{R}_{11})_{ij}\bar{\xi}_i\bar{\xi}_j - \frac{1}{2}(\mathbf{R}_{11}^*)_{ji}\xi_j'\xi_i'} e^{\theta(\mathfrak{M}_1 \cdot \bar{\xi}) + \bar{\theta}(\mathfrak{M}_1^* \cdot \xi')} \int \prod_{l=k+1}^{L+1} d\bar{\xi}_l d\xi_l \, e^{-\bar{\xi}_l\xi_l}$$
$$\times e^{-\theta(\mathfrak{M}_2\bar{\xi}) + \bar{\theta}(\mathfrak{M}_2^*\xi)} \left[ e^{-(\mathbf{R}_{12})_{in}\bar{\xi}_i\bar{\xi}_n + \frac{1}{2}(\mathbf{R}_{22})_{mn}\bar{\xi}_m\bar{\xi}_n} e^{-\frac{1}{2}(\mathbf{R}_{22}^*)_{nm}\xi_n\xi_m + (\mathbf{R}_{12}^*)_{jm}\xi_j'\xi_m} \right] \tag{B.16}$$

We denote the terms on the left of second integral as $\mathcal{F}(\bar{\xi}, \xi, \theta, \bar{\theta})$. If we introduce new Grassmann variables $\boldsymbol{\eta}^T = \begin{pmatrix} \bar{\xi} & \xi \end{pmatrix}$ and $\boldsymbol{\lambda}^T = \begin{pmatrix} -\bar{\xi}\mathbf{R}_{12} - \theta(\mathfrak{M})_0 & -\xi'\mathbf{R}_{12}^* + \bar{\theta}(\mathfrak{M}^*)_0 \end{pmatrix}$, then we can write the integral as

$$\rho_1^{\beta}(4, \xi, \xi') = \int d\bar{\theta}d\theta \mathcal{F}(\bar{\xi}, \xi', \theta, \bar{\theta}) \int \mathbf{D}\boldsymbol{\eta} \, e^{\frac{1}{2}\boldsymbol{\eta}^T\mathcal{A}\boldsymbol{\eta} + \boldsymbol{\lambda}^T\boldsymbol{\eta}} \tag{B.17}$$

where $\mathcal{A}$ has been defined previously. Solving the first integral we get:

$$\rho_1^{\beta}(4, \xi, \xi') = \mathcal{C}^{\beta} e^{\frac{1}{2}\begin{pmatrix} \bar{\xi} & \xi' \end{pmatrix}\boldsymbol{\Omega}\begin{pmatrix} \bar{\xi} \\ \xi' \end{pmatrix}} \int \mathbf{D}\boldsymbol{\Theta} e^{\frac{1}{2}\boldsymbol{\Theta}^T\boldsymbol{\omega}\boldsymbol{\Theta} + \boldsymbol{\eta}'^T\mathcal{J}\boldsymbol{\Theta}} \tag{B.18}$$

where $\boldsymbol{\Theta}^T = \begin{pmatrix} \bar{\theta} & \theta \end{pmatrix}$, $\mathbf{D}\boldsymbol{\Theta} = d\bar{\theta}d\theta$, $\boldsymbol{\eta}'^T = \begin{pmatrix} \bar{\xi} & \xi' \end{pmatrix}$, and $\boldsymbol{\Omega}$ and $\mathcal{C}^{\beta}$ are given by (79b) and (79a) respectively. Finally, if we take the last integral in the equation (B.18), we get

$$\rho_1^{\beta}(4, \xi, \xi') = \mathcal{C}^{\beta}\text{Pf}[\boldsymbol{\omega}] e^{\frac{1}{2}\begin{pmatrix} \bar{\xi} & \xi' \end{pmatrix}\boldsymbol{\Omega}'\begin{pmatrix} \bar{\xi} \\ \xi' \end{pmatrix}} \tag{B.19}$$

where

$$\boldsymbol{\Omega}' = \boldsymbol{\Omega} + \mathcal{J}\boldsymbol{\omega}^{-1}\mathcal{J}^T \tag{B.20a}$$

$$\boldsymbol{\omega} = \begin{pmatrix} 0 & (\mathfrak{M}_2^*)_0 \\ -(\mathfrak{M}_2)_0 & 0 \end{pmatrix} \mathcal{A}^{-1} \begin{pmatrix} 0 & -(\mathfrak{M}_2)_0^T \\ (\mathfrak{M}_2)_0^{\dagger} & 0 \end{pmatrix}, \tag{B.20b}$$

$$\mathcal{J} = \begin{pmatrix} -\mathbf{R}_{12} & \mathbf{0} \\ \mathbf{0} & -\mathbf{R}_{12}{}^* \end{pmatrix} \mathcal{A}^{-1} \begin{pmatrix} \mathbf{0} & \mathfrak{M}_2^* \\ -\mathfrak{M}_2 & \mathbf{0} \end{pmatrix} - \begin{pmatrix} \mathbf{0} & \mathfrak{M}_1^* \\ \mathfrak{M}_1 & \mathbf{0} \end{pmatrix}. \tag{B.20c}$$

To do the same calculation for cross term $\varrho^\beta(2, \xi, \xi')$ in equation (B.12), we have

$$\varrho^\beta(2, \xi, \xi') = |C^\beta|^2 \beta \, \mathfrak{M}_{0l} \bar{\xi}_l \, e^{\frac{1}{2} R_{ij} \bar{\xi}_i \bar{\xi}_j - \frac{1}{2} R_{ij}^* \xi_i' \xi_j'} \tag{B.21}$$

If we use the equation (B.11) and write the $\xi$ behind the exponential as $(\mathfrak{M}\bar{\xi})_0 = \int d\theta \, e^{\theta(\mathfrak{M}\bar{\xi})_0} = \int d\theta \, e^{\theta \mathfrak{M}_{0l} \bar{\xi}_l}$, where $\theta$ is a new Grassmann variables. We can write the density matrix as

$$\varrho^\beta(2, \xi, \xi') = |C^\beta|^2 \beta \int d\theta \, e^{\theta(\mathfrak{M}\bar{\xi})_0} e^{\frac{1}{2} R_{ij} \bar{\xi}_i \bar{\xi}_j} e^{-\frac{1}{2} R_{ij}^* \xi_i' \xi_j'}. \tag{B.22}$$

Equivalently, If we introduce new Grassmann variables $\eta^T = \begin{pmatrix} \bar{\xi} & \xi \end{pmatrix}$ then we can write the integral as

$$\varrho^\beta(2, \xi, \xi') = \int d\theta \, \mathcal{F}(\bar{\xi}, \xi', \theta) \int \mathbf{D}\eta \, e^{\frac{1}{2}\eta^T \mathcal{A}\eta + \lambda^T \eta}, \tag{B.23}$$

where we have denoted the terms on the left of second integral by $\mathcal{F}(\bar{\xi}, \xi, \theta)$ and

$$\lambda^T = \begin{pmatrix} -\bar{\xi} \mathbf{R}_{12} - \theta(\mathfrak{M}_2)_{0l'} & -\xi' \mathbf{R}_{12}^* \end{pmatrix} \tag{B.24}$$

Taking the integration on $\eta$, we have:

$$\varrho^\beta(2, \xi, \xi') = \frac{C^\beta}{\beta^*} e^{\frac{1}{2} \begin{pmatrix} \bar{\xi} & \xi' \end{pmatrix} \Omega \begin{pmatrix} \bar{\xi} \\ \xi' \end{pmatrix}} \int d\theta \, e^{\theta \zeta} \tag{B.25}$$

where $\Omega$ is the same as (79b) and $C^\beta$ is given by (79a). Then, we can take the integral over the $\theta$ above and write the final result as

$$\varrho^\beta(2, \xi, \xi') = \frac{C^\beta}{\beta^*} (\Upsilon \mathcal{A}^{-1} \mathcal{R}^T + \Upsilon') \eta' e^{\frac{1}{2}\eta'^T \Omega \eta'} \tag{B.26}$$

where $\eta'^T = \begin{pmatrix} \bar{\xi} & \xi' \end{pmatrix}$, and Also we have

$$\Upsilon = \begin{bmatrix} \mathfrak{M}_2 & \mathbf{0} \end{bmatrix}_{1 \times 2L'} \qquad \Upsilon' = \begin{bmatrix} \mathfrak{M}_1 & \mathbf{0} \end{bmatrix}_{1 \times 2l} \qquad \mathcal{R} = \begin{bmatrix} \mathbf{R}_{12} & \mathbf{0} \\ \mathbf{0} & \mathbf{R}_{12}^* \end{bmatrix}_{2l \times 2L'} \tag{B.27}$$

For the other cross density (third term in equation (B.12)) we have:

$$\varrho^\beta(3, \xi, \xi') = \frac{C^\beta}{\beta} (-\Upsilon \mathcal{A}^{-1} \mathcal{R}^T + \Upsilon') \eta' e^{\frac{1}{2}\eta'^T \Omega^\beta \eta'} \tag{B.28}$$

which

$$\Upsilon = \begin{bmatrix} \mathbf{0} & \mathfrak{M}_{2^*} \end{bmatrix}_{1 \times 2L'} \qquad \Upsilon' = \begin{bmatrix} \mathbf{0} & \mathfrak{M}_{1^*} \end{bmatrix} \qquad \mathcal{R} = \begin{bmatrix} \mathbf{R}_{12} & 0 \\ 0 & \mathbf{R}_{12}^* \end{bmatrix}_{2l \times 2L'} \tag{B.29}$$

Putting equations (B.13), (B.19), (B.26) and (B.28) together and moving and reordering some parts we can write the RDM for the state $|\beta\rangle$ as

$$\rho_1^\beta(\xi, \xi') = \mathcal{C}_\beta \Big[ \frac{1}{|\beta|^2} + \mathrm{Pf}[\mathbf{\Omega_2}] e^{\frac{1}{2} \begin{pmatrix} \bar{\xi} & \xi' \end{pmatrix} \mathcal{J} \boldsymbol{\omega}^{-1} \mathcal{J}^T \begin{pmatrix} \bar{\xi} \\ \xi' \end{pmatrix}} + \mathcal{K}^\beta \begin{pmatrix} \bar{\xi} \\ \xi' \end{pmatrix} \Big] e^{\frac{1}{2}(\Omega_{11})_{ij}\bar{\xi}_i\bar{\xi}_j} e^{\mathcal{Y}_{ij}\bar{\xi}_i\xi'_j} e^{\frac{1}{2}(\Omega_{22})_{ij}\xi'_i\xi'_j}, \tag{B.30}$$

where $2\mathcal{Y} = \mathbf{\Omega}_{12} - \mathbf{\Omega}_{21}^T$ and

$$\mathcal{K}^\beta = \left( \frac{1}{\beta^*}\mathfrak{M}_1 \quad \frac{1}{\beta}\mathfrak{M}_{1^*} \right) + \left( \frac{1}{\beta^*}\mathfrak{M}_2 \quad -\frac{1}{\beta}\mathfrak{M}_{2^*} \right) \mathcal{A}^{-1} \begin{pmatrix} \mathbf{R}_{12}^T & 0 \\ 0 & \mathbf{R}_{12}^\dagger \end{pmatrix} \tag{B.31}$$

To write the RDM in operator format, we have to reorder it. Then, the final result is

$$\begin{aligned} \rho_1^\beta(c, c^\dagger) = \mathcal{C}_\beta e^{\frac{1}{2}(\Omega_{11})_{ij}c_i^\dagger c_j^\dagger} \Big[ &\frac{1}{|\beta|^2} e^{(\ln \frac{\Omega_{12}-\Omega_{21}^T}{2})_{ij}c_i^\dagger c_j} + (\mathcal{K}_1^\beta)_k c_k^\dagger e^{(\ln \frac{\Omega_{12}-\Omega_{21}^T}{2})_{ij}c_i^\dagger c_j} \\ &+ e^{(\ln \frac{\Omega_{12}-\Omega_{21}^T}{2})_{ij}c_i^\dagger c_j}(\mathcal{K}_2^\beta)_l c_l \Big] e^{\frac{1}{2}(\Omega_{22})_{ij}c_i c_j} \\ &+ \mathcal{C}^\beta \mathrm{Pf}[\mathbf{\Omega_2}] e^{\frac{1}{2}(\Omega'_{11})_{ij}c_i^\dagger c_j^\dagger} e^{(\ln \frac{\Omega'_{12}-\Omega'_{21}^T}{2})_{ij}c_i^\dagger c_j} e^{\frac{1}{2}(\Omega'_{22})_{ij}c_i c_j}. \end{aligned} \tag{B.32}$$

The $\mathcal{K}_1^\beta$ stands for the first block part of vector $\mathcal{K}^\beta$ (and the same argument for $\mathcal{K}_2^\beta$). To find a shorter notation for the $\rho_1^\beta(c, c^\dagger)$ we use the relation [57]

$$e^{\frac{1}{2}\begin{pmatrix} c^\dagger & c \end{pmatrix}\mathcal{M}_1\begin{pmatrix} c \\ c^\dagger \end{pmatrix}} e^{\frac{1}{2}\begin{pmatrix} c^\dagger & c \end{pmatrix}\mathcal{M}_2\begin{pmatrix} c \\ c^\dagger \end{pmatrix}} e^{\frac{1}{2}\begin{pmatrix} c^\dagger & c \end{pmatrix}\mathcal{M}_3\begin{pmatrix} c \\ c^\dagger \end{pmatrix}} = e^{\frac{1}{2}\begin{pmatrix} c^\dagger & c \end{pmatrix}\mathcal{M}\begin{pmatrix} c \\ c^\dagger \end{pmatrix}} \tag{B.33}$$

where $e^\mathcal{M} = e^{\mathcal{M}_1} e^{\mathcal{M}_2} e^{\mathcal{M}_3}$. Therefore, we can write (B.32) as

$$\begin{aligned} \rho_1^\beta(c, c^\dagger) = \mathcal{C}_\beta \Big[ &\frac{1}{|\beta|^2} e^{\frac{1}{2}\begin{pmatrix} c^\dagger & c \end{pmatrix}\mathcal{M}\begin{pmatrix} c \\ c^\dagger \end{pmatrix}} e^{\frac{1}{2}\mathrm{tr}\ln\mathcal{Y}} + \mathrm{Pf}[\mathbf{\Omega_2}] e^{\frac{1}{2}\begin{pmatrix} c^\dagger & c \end{pmatrix}\mathcal{M}'\begin{pmatrix} c \\ c^\dagger \end{pmatrix}} e^{\frac{1}{2}\mathrm{tr}\ln(\mathcal{Y}+\mathcal{Y})} \\ &+ (\mathcal{K}_1^\beta)_k c_k^\dagger e^{\frac{1}{2}\begin{pmatrix} c^\dagger & c \end{pmatrix}\mathcal{M}\begin{pmatrix} c \\ c^\dagger \end{pmatrix}} e^{\frac{1}{2}\mathrm{tr}\ln\mathcal{Y}} + e^{\frac{1}{2}\begin{pmatrix} c^\dagger & c \end{pmatrix}\mathcal{M}\begin{pmatrix} c \\ c^\dagger \end{pmatrix}} e^{\frac{1}{2}\mathrm{tr}\ln\mathcal{Y}}(\mathcal{K}_2^\beta)_l c_l \Big]. \end{aligned} \tag{B.34}$$

To move the exponentials to one side (left or right) of the (B.34), we use the relation (83). Therefore, we can write the (B.34) as

$$
\rho_1^\beta(c,c^\dagger) = \mathcal{C}_\beta\, e^{\frac{1}{2}\begin{pmatrix} \mathbf{c}^\dagger & \mathbf{c} \end{pmatrix}\mathcal{M}\begin{pmatrix} \mathbf{c} \\ \mathbf{c}^\dagger \end{pmatrix}} e^{\frac{1}{2}\mathrm{tr}\ln(\frac{1}{2}\Omega_{12}-\frac{1}{2}\Omega_{21}^T)}\left[\frac{1}{|\beta|^2}+\begin{pmatrix} \mathcal{L}_1^\beta & \mathcal{L}_2^\beta \end{pmatrix}\begin{pmatrix} \mathbf{c}^\dagger \\ \mathbf{c} \end{pmatrix}\right]
$$

$$
+\,\mathcal{C}_\beta\,\mathrm{Pf}[\Omega_2]\,e^{\frac{1}{2}\begin{pmatrix} \mathbf{c}^\dagger & \mathbf{c} \end{pmatrix}\mathcal{M}'\begin{pmatrix} \mathbf{c} \\ \mathbf{c}^\dagger \end{pmatrix}} e^{\frac{1}{2}\mathrm{tr}\ln(\frac{1}{2}\Omega'_{12}-\frac{1}{2}\Omega'_{21}{}^T)}
\tag{B.35}
$$

where we have:

$$
\mathcal{M} = \ln\begin{pmatrix} \frac{1}{2}\Omega_{12}-\frac{1}{2}\Omega_{21}{}^T+2\Omega_{11}(\Omega_{12}^T-\Omega_{21})^{-1}\Omega_{22} & 2\Omega_{11}(\Omega_{12}^T-\Omega_{21})^{-1} \\ 2(\Omega_{12}^T-\Omega_{21})^{-1}\Omega_{22} & 2(\Omega_{12}^T-\Omega_{21})^{-1} \end{pmatrix},
\tag{B.36a}
$$

$$
\mathcal{M}' = \ln\begin{pmatrix} \frac{1}{2}\Omega'_{12}-\frac{1}{2}\Omega'_{21}{}^T+2\Omega'_{11}(\Omega'_{12}{}^T-\Omega_{21}')^{-1}\Omega'_{22} & 2\Omega'_{11}(\Omega'_{12}{}^T-\Omega'_{21})^{-1} \\ 2(\Omega'_{12}{}^T-\Omega'_{21})^{-1}\Omega'_{22} & 2(\Omega'_{12}{}^T-\Omega'_{21})^{-1} \end{pmatrix}
\tag{B.36b}
$$

$$
\Omega = \begin{pmatrix} \mathbf{R}_{11} & 0 \\ 0 & -\mathbf{R}_{11}{}^* \end{pmatrix} + \begin{pmatrix} \mathbf{R}_{12} & 0 \\ 0 & \mathbf{R}_{12}{}^* \end{pmatrix}\mathcal{A}^{-1}\begin{pmatrix} \mathbf{R}_{12}{}^T & 0 \\ 0 & \mathbf{R}_{12}{}^\dagger \end{pmatrix}; \qquad \mathcal{A} = \begin{pmatrix} \mathbf{R}_{22} & -\mathbf{I} \\ \mathbf{I} & -\mathbf{R}_{22}^* \end{pmatrix}
\tag{B.36c}
$$

$$
\Omega' = \Omega + \mathcal{J}\omega^{-1}\mathcal{J}^T, \qquad \omega = \begin{pmatrix} \mathfrak{M}_2^*(\mathcal{A}^{-1})_{22}\mathfrak{M}_2^\dagger & -\mathfrak{M}_2^*(\mathcal{A}^{-1})_{21}\mathfrak{M}_2^T \\ -\mathfrak{M}_2(\mathcal{A}^{-1})_{12}\mathfrak{M}_2^\dagger & \mathfrak{M}_2^*(\mathcal{A}^{-1})_{11}\mathfrak{M}_2^T \end{pmatrix}
\tag{B.36d}
$$

$$
\mathcal{J} = \begin{pmatrix} -\mathbf{R}_{12} & 0 \\ 0 & -\mathbf{R}_{12}{}^* \end{pmatrix}\mathcal{A}^{-1}\begin{pmatrix} 0 & \mathfrak{M}_2^* \\ -\mathfrak{M}_2 & 0 \end{pmatrix} - \begin{pmatrix} 0 & \mathfrak{M}_1^* \\ \mathfrak{M}_1 & 0 \end{pmatrix},
\tag{B.36e}
$$

$$
\mathcal{K}^\beta = \begin{pmatrix} \frac{1}{\beta^*}\mathfrak{M}_1 & \frac{1}{\beta}\mathfrak{M}_1{}^* \end{pmatrix} + \begin{pmatrix} \frac{1}{\beta^*}\mathfrak{M}_2 & -\frac{1}{\beta}\mathfrak{M}_2{}^* \end{pmatrix}\mathcal{A}^{-1}\begin{pmatrix} \mathbf{R}_{12}{}^T & 0 \\ 0 & \mathbf{R}_{12}{}^\dagger \end{pmatrix}
\tag{B.36f}
$$

$$
\mathcal{L}_1^\beta = 2\mathcal{K}_1^\beta(\Omega_{12}^T-\Omega_{21})^{-1}, \qquad \mathcal{L}_2^\beta = 2\mathcal{K}_1^\beta(\Omega_{12}^T-\Omega_{21})^{-1}\Omega_{22}+\mathcal{K}_2^\beta.
\tag{B.36g}
$$

$\mathcal{K}_1^\beta$ stands for the first block part of vector $\mathcal{K}^\beta$. Since, we can think of $\mathcal{K}^\beta$ as a vector made from two block vectors (and the same argument for $\mathcal{K}_2^\beta$).

## Appendix C   Entanglement calculations of general $\beta$ parity broken state

To calculate the $n = 2$ Rényi EE, we start by:

$$\text{tr}(\rho_A^{\beta\,2}) = \text{tr}(\rho_A^\beta \mathbf{I} \rho_A^\beta). \tag{C.1}$$

We can use the Identity resolution and tracing formula of Grassmann variables to write:

$$\mathbf{I} = \int \prod_l d\bar{\xi}_l d\xi_l e^{-\bar{\xi}\cdot\xi} |\xi\rangle\langle\xi|, \tag{C.2a}$$

$$\text{tr}\mathcal{O} = \int \prod_i d\bar{\xi}_i d\xi_i e^{-\bar{\xi}\cdot\xi} \langle-\xi| \mathcal{O} |\xi\rangle. \tag{C.2b}$$

Therefore, it is possible to calculate the trace of $\rho^2$ in terms of Berezin integrals of Grassmann variables:

$$\text{tr}(\rho_A^{\beta\,2}) = \int \prod_i d\bar{\xi}_i d\xi_i \prod_l d\bar{\eta}_l d\eta_l e^{-\bar{\xi}\cdot\xi} e^{-\bar{\eta}\cdot\eta} \langle-\xi| \rho_A^\beta |\eta\rangle\langle\eta| \rho_A^\beta |\xi\rangle. \tag{C.3}$$

We can use the result of (78) to write the integrand in the expression above as:

$$\text{tr}(\rho_A^{\beta\,2}) = \mathcal{C}^{\beta\,2} \int \prod_i d\bar{\xi}_i d\xi_i \prod_l d\bar{\eta}_l d\eta_l e^{-\bar{\xi}\cdot\xi} e^{-\bar{\eta}\cdot\eta} \big[ (\frac{1}{\beta} - \mathcal{L}_1\cdot\bar{\xi} + \mathcal{L}_2\cdot\eta)(\frac{1}{\beta^*} - \mathcal{L}_3\cdot\bar{\xi} + \mathcal{L}_4\cdot\eta) - \text{Pf}[\mathcal{W}]\big]$$
$$e^{\frac{1}{2}\begin{pmatrix}\bar{\xi} & \eta\end{pmatrix}\Omega'\begin{pmatrix}\bar{\xi}\\\eta\end{pmatrix}} \big[ (\frac{1}{\beta} + \mathcal{L}_1\cdot\bar{\eta} + \mathcal{L}_2\cdot\xi)(\frac{1}{\beta^*} + \mathcal{L}_3\cdot\bar{\eta} + \mathcal{L}_4\cdot\xi) - \text{Pf}[\mathcal{W}]\big] e^{\frac{1}{2}\begin{pmatrix}\bar{\eta} & \xi\end{pmatrix}\Omega\begin{pmatrix}\bar{\eta}\\\xi\end{pmatrix}}, \tag{C.4}$$

By defining a new Grassmann variable such as:

$$\theta = \begin{pmatrix}\bar{\xi}\\\bar{\eta}\\\eta\\\xi\end{pmatrix}, \tag{C.5}$$

the integral (C.3) can be written in a simpler form as:

$$\text{tr}(\rho_A^{\beta\,2}) = \mathcal{C}^{\beta\,2} \int \mathbf{D}\theta\, \text{f}(\theta) e^{\frac{1}{2}\theta^T\cdot\mathcal{B}\cdot\theta}, \tag{C.6}$$

where in the above $\mathbf{D}\theta = \prod_i d\bar{\theta}_i d\theta_i$, and also

$$\text{f}(\theta) = \big[(\frac{1}{\beta} + \mathbb{C}\cdot\theta|_1)(\frac{1}{\beta^*} + \mathbb{C}\cdot\theta|_2) + \text{Pf}[\mathcal{W}]\big]\big[(\frac{1}{\beta} + \mathbb{C}\cdot\theta|_3)(\frac{1}{\beta^*} + \mathbb{C}\cdot\theta|_4) + \text{Pf}[\mathcal{W}]\big].$$

The new $\mathcal{B}$ and $\mathbb{C}$ matrices are defined below

$$
\mathcal{B} = \begin{pmatrix} \boldsymbol{\Omega}_{11} & \mathbf{0} & -\boldsymbol{\Omega}_{12} & \mathbf{I} \\ \mathbf{0} & \boldsymbol{\Omega}_{11} & \mathbf{I} & \boldsymbol{\Omega}_{12} \\ -\boldsymbol{\Omega}_{21} & -\mathbf{I} & \boldsymbol{\Omega}_{22} & \mathbf{0} \\ -\mathbf{I} & \boldsymbol{\Omega}_{21} & \mathbf{0} & \boldsymbol{\Omega}_{21} \end{pmatrix}, \qquad \mathbb{C} = \begin{pmatrix} -\mathcal{L}_1 & \mathbf{0} & \mathcal{L}_2 & \mathbf{0} \\ -\mathcal{L}_3 & \mathbf{0} & \mathcal{L}_4 & \mathbf{0} \\ \mathbf{0} & \mathcal{L}_1 & \mathbf{0} & \mathcal{L}_2 \\ \mathbf{0} & \mathcal{L}_3 & \mathbf{0} & \mathcal{L}_4 \end{pmatrix}. \tag{C.7}
$$

The rest of matrices are given by (79). Before proceeding to solve the integral above, we have to clarify couple of points here. First, the notation $\mathbb{C}.\boldsymbol{\theta}|_r$ stand for the $r^{\text{th}}$ row of the matrix product $\mathbb{C}.\boldsymbol{\theta}$. Second, the f($\boldsymbol{\theta}$) produce terms with product of odd number of Grassmann variables such as $\mathbb{C}.\boldsymbol{\theta}|_r$ or $\mathbb{C}.\boldsymbol{\theta}|_{r_3} \mathbb{C}.\boldsymbol{\theta}|_{r_2} \mathbb{C}.\boldsymbol{\theta}|_{r_1}$, the integration on these terms will be zero automatically. Therefore, The above integral is straightforward to solve, and the final result of tracing is given by (109).

In some cases $\mathcal{B}$ does not have an inverse for odd $n$. However, it is still possible to take the Grassmann integrations.

## Appendix D  Exact diagonalization of A=B=0

In the case where **A** and **B** are zero then the Hamiltonian (5) becomes

$$
H = \sum_{j=1}^{L} \left[ \alpha_j^0 (c_0 c_j - c_0^\dagger c_j) + \alpha_j^{L+1} (c_j c_{L+1}^\dagger + c_j c_{L+1}) + \alpha_j^{0*} (c_j^\dagger c_0^\dagger - c_j^\dagger c_0) + \alpha_j^{L+1*} (c_{L+1} c_j^\dagger + c_{L+1}^\dagger c_j^\dagger) \right]. \tag{D.1}
$$

This model can be related to a linear fermionic model such as the model discussed in [45]

$$
H = \sum_{j=1}^{L} \alpha_j c_j + \alpha_j^* c_j^\dagger. \tag{D.2}
$$

In fact, with a specific projection, one can find the eigenstates of (D.2) in the Hilbert space of (D.1). We can write the new form of the **M** matrix (9)

$$
\mathbf{M} = \begin{pmatrix}
0 & -\alpha_1^0 & -\alpha_2^0 & \cdots & -\alpha_{L-1}^0 & -\alpha_L^0 & 0 & 0 & -\alpha_1^{0*} & -\alpha_2^{0*} & \cdots & -\alpha_{L-1}^{0*} & -\alpha_L^{0*} & 0 \\
-\alpha_1^{0*} & 0 & 0 & \cdots & 0 & 0 & -\alpha_1^{L+1*} & \alpha_1^{0*} & 0 & 0 & \cdots & 0 & 0 & -\alpha_1^{L+1*} \\
-\alpha_2^{0*} & 0 & & & & 0 & -\alpha_2^{L+1*} & \alpha_2^{0*} & 0 & & & & 0 & -\alpha_2^{L+1*} \\
\vdots & \vdots & & & & \vdots & \vdots & \vdots & \vdots & & & & \vdots & \vdots \\
-\alpha_{L-1}^{0*} & 0 & & & & 0 & -\alpha_{L-1}^{L+1*} & \alpha_{L-1}^{0*} & 0 & & & & 0 & -\alpha_{L-1}^{L+1*} \\
-\alpha_L^{0*} & 0 & 0 & \cdots & 0 & 0 & -\alpha_L^{L+1*} & \alpha_L^{0*} & 0 & 0 & \cdots & 0 & 0 & -\alpha_L^{L+1*} \\
0 & -\alpha_1^{L+1} & -\alpha_2^{L+1} & \cdots & -\alpha_{L-1}^{L+1} & -\alpha_L^{L+1} & 0 & 0 & \alpha_1^{L+1*} & \alpha_2^{L+1*} & \cdots & \alpha_{L-1}^{L+1*} & \alpha_L^{L+1*} & 0 \\
\hline
0 & \alpha_1^0 & \alpha_2^0 & \cdots & \alpha_{L-1}^0 & \alpha_L^0 & 0 & 0 & \alpha_1^{0*} & \alpha_2^{0*} & \cdots & \alpha_{L-1}^{0*} & \alpha_L^{0*} & 0 \\
-\alpha_1^0 & 0 & 0 & \cdots & 0 & 0 & \alpha_1^{L+1} & \alpha_1^0 & 0 & 0 & \cdots & 0 & 0 & \alpha_1^{L+1} \\
-\alpha_2^0 & 0 & & & & 0 & \alpha_2^{L+1} & \alpha_2^0 & 0 & & & & 0 & \alpha_2^{L+1} \\
\vdots & \vdots & & & & \vdots & \vdots & \vdots & \vdots & & & & \vdots & \vdots \\
-\alpha_{L-1}^0 & 0 & & & & 0 & \alpha_{L-1}^{L+1} & \alpha_{L-1}^0 & 0 & & & & 0 & \alpha_{L-1}^{L+1} \\
-\alpha_L^0 & 0 & 0 & \cdots & 0 & 0 & \alpha_L^{L+1} & \alpha_L^0 & 0 & 0 & \cdots & 0 & 0 & \alpha_L^{L+1} \\
0 & -\alpha_1^{L+1} & -\alpha_2^{L+1} & \cdots & -\alpha_{L-1}^{L+1} & -\alpha_L^{L+1} & 0 & 0 & \alpha_1^{L+1*} & \alpha_2^{L+1*} & \cdots & \alpha_{L-1}^{L+1*} & \alpha_L^{L+1*} & 0
\end{pmatrix}. \tag{D.3}
$$

It can be shown that $\mathbf{\Lambda}$ has only two nonzero diagonal elements, $\lambda_1$ and $\lambda_2$ and the rest of eigenvalues are zero. For nonzero eigenvalues, we have

$$\lambda_1, \lambda_2 = \sqrt{2}\sqrt{\sum_{j=1}^{L}(|\alpha_j^0|^2 + |\alpha_j^{L+1}|^2) \pm \sqrt{\left|\sum_{j=1}^{L}(\alpha_j^0 + \alpha_j^{L+1})(\alpha_j^{0*} - \alpha_j^{L+1*})\right|^2}}. \tag{D.4}$$

Then, $\mathbf{M}$ would have only four nonzero eigenvalues and the rest are zero independent of size. The next step is to find the eigenvectors to construct the $\mathbf{U}$ matrix. In (12), we have already introduced two eigenvectors of zero mode. Using the orthogonality condition, and the condition that if $|u\rangle$ is an eigenvector with eigenvalue $\lambda$ then $\mathbf{J}|u\rangle$ is also an eigenvector with eigenvalue $-\lambda$, therefore, we can find and present only half of the eigenvectors. For simplicity, we assume $\alpha_j^0 = \alpha_j^{L+1} = \alpha_j$, then eigenvectros of positive modes. we have:

$$|u_1^+\rangle = \begin{pmatrix} u \\ v \end{pmatrix}; \qquad \begin{array}{l} u_{L+1} = u_0 = -v_0 = v_{L+1} = 1 \\[2mm] u_i = -\dfrac{2\alpha_i^*}{\sqrt{\sum_{j=1}^{L}|\alpha_j|^2}} \\[3mm] \text{else} = 0 \end{array} \tag{D.5}$$

$$|u_2^+\rangle = \begin{pmatrix} u \\ v \end{pmatrix}; \qquad \begin{array}{l} u_{L+1} = -u_0 = v_0 = v_{L+1} = \dfrac{\alpha_L}{|\alpha_L|} \\[2mm] v_i = \dfrac{2\alpha_i}{\sqrt{\sum_{j=1}^{L}|\alpha_j|^2}}\dfrac{\alpha_L}{|\alpha_L|} \\[3mm] \text{else} = 0 \end{array} \tag{D.6}$$

For eigenvectros of zero modes, we have:

$$|u_1^0\rangle = \begin{pmatrix} u \\ v \end{pmatrix}; \qquad \begin{array}{l} u_{L+1} = u_0 = v_0 = -v_{L+1} = 1 \\[2mm] \text{else} = 0 \end{array} \tag{D.7}$$

$$|u_k^0\rangle = \begin{pmatrix} u \\ v \end{pmatrix}; \qquad \begin{array}{l} u_i = -\dfrac{\alpha_k \alpha_i^*}{\sum_{j=1}^{k-1}|\alpha_j|^2}; \quad i < k \neq 0 \\[3mm] u_k = 1 \\[2mm] \text{else} = 0, \end{array} \tag{D.8}$$

where $k = 2, 3, \cdots, L$. With this expressions for eigenvectors, we can construct the unitary matrices $\mathbf{U}$. Having the exact $\mathbf{U}$, we can calculate correlation matrices (see section 4). putting these eigenvectors together, one can construct the $\mathbf{U}$.

Having the $\mathbf{U}$ matrix, we can directly calculate the $\delta_+$ introduced in (42). The calculation of $\delta_+$ shows that

$$\delta_+ = \begin{cases} +1 & \text{if } L \text{ even} \\ -1 & \text{if } L \text{ odd} \end{cases} \tag{D.9}$$

### D.1 Vacuum state in configuration basis

Another method to calculate the entanglement of subsystem in a particular state is to use a more direct way. We could use the equation (28) to calculate the entanglement. Using the results of subsection 3.4 for $\mathbf{A}, \mathbf{B} = 0$ we can find the form of $\mathbf{R}$ matrix as:.

$$\mathbf{R} = \begin{pmatrix} 0 & \frac{\alpha_j}{\sqrt{|\alpha_1|^2 + \cdots + |\alpha_L|^2}} & 1 \\ \frac{-\alpha_i}{\sqrt{|\alpha_1|^2 + \cdots + |\alpha_L|^2}} & 0 & \frac{\alpha_i}{\sqrt{|\alpha_1|^2 + \cdots + |\alpha_L|^2}} \\ -1 & \frac{-\alpha_j}{\sqrt{|\alpha_1|^2 + \cdots + |\alpha_L|^2}} & 0 \end{pmatrix} \tag{D.10}$$

where,

$$|0\rangle_\eta = \frac{1}{\sqrt{\det[\mathbf{I} + \mathbf{R}^\dagger \mathbf{R}]}} e^{\frac{1}{2} \sum_{i,j} R_{ij} c_i^\dagger c_j^\dagger} |0\rangle_c .$$

Numerical investigations suggest that for even sizes we get exactly the same expression, and for odd sizes we get the minus of the vacuum ($\alpha_i \in \mathbb{R}$).

### D.2 Correlations

For the special case of $\mathbf{A}, \mathbf{B} = 0$, the correlation matrices look pretty much simple in terms of the parameters $\alpha_i$'s. We are going to demonstrate the matrix form of correlations for general $\alpha_i \in \mathbb{C}$. For start, in the vacuum state, correlations looks like below. (In the following expressions indexes $n$ and $m$ are the number of rows and columns respectively.)

$$\mathbf{C}^0 = \begin{pmatrix} \frac{1}{2} & \frac{\alpha_m}{4\sqrt{\sum_k |\alpha_k|^2}} & -\frac{1}{4} \\ \frac{\alpha_n^*}{4\sqrt{\sum_k |\alpha_k|^2}} & \frac{\alpha_n^* \alpha_m}{2(\sum_k |\alpha_k|^2)} & \frac{\alpha_n^*}{4\sqrt{\sum_k |\alpha_k|^2}} \\ -\frac{1}{4} & \frac{\alpha_m}{4\sqrt{\sum_k |\alpha_k|^2}} & \frac{1}{2} \end{pmatrix}, \quad \mathbf{F}^0 = \begin{pmatrix} 0 & \frac{\alpha_m^*}{4\sqrt{\sum_k |\alpha_k|^2}} & \frac{1}{4} \\ \frac{-\alpha_n^*}{4\sqrt{\sum_k |\alpha_k|^2}} & 0 & \frac{\alpha_n^*}{4\sqrt{\sum_k |\alpha_k|^2}} \\ -\frac{1}{4} & \frac{-\alpha_m^*}{4\sqrt{\sum_k |\alpha_k|^2}} & 0 \end{pmatrix}$$

$$\mathbf{K}^0 = \begin{pmatrix} 1 & 0 & 0 \\ 0 & \delta_{nm} - \frac{m-n}{|m-n|} \frac{i\Im[\alpha_n^* \alpha_m]}{\sum_k |\alpha_k|^2} & \frac{-i\Im[\alpha_n]}{\sqrt{\sum_k |\alpha_k|^2}} \\ 0 & \frac{i\Im[\alpha_m]}{\sqrt{\sum_k |\alpha_k|^2}} & 1 \end{pmatrix}, \quad \bar{\mathbf{K}}^0 = \begin{pmatrix} 1 & \frac{i\Im[\alpha_m]}{\sqrt{\sum_k |\alpha_k|^2}} & 0 \\ \frac{-i\Im[\alpha_n]}{\sqrt{\sum_k |\alpha_k|^2}} & \delta_{nm} - \frac{m-n}{|m-n|} \frac{i\Im[\alpha_n^* \alpha_m]}{\sum_k |\alpha_k|^2} & 0 \\ 0 & 0 & 1 \end{pmatrix}$$

$$\mathbf{G}^0 = \begin{pmatrix} 0 & \frac{\Re[\alpha_m]}{\sqrt{\sum_k |\alpha_k|^2}} & 0 \\ 0 & -\delta_{nm} + \frac{\Re[\alpha_n^* \alpha_m]}{\sum_k |\alpha_k|^2} & \frac{\Re[\alpha_n]}{\sqrt{\sum_k |\alpha_k|^2}} \\ -1 & 0 & 0 \end{pmatrix}$$

For the case where the state is ZME state ($|\emptyset\rangle$) then the correlation matrices look like below.

$$
\mathbf{C}^{\emptyset} = \left(
\begin{array}{c|c|c}
\frac{1}{2} & \frac{\alpha_m}{4\sqrt{\sum_k |\alpha_k|^2}} & \frac{1}{4} \\
\hline
\frac{\alpha_n^*}{4\sqrt{\sum_k |\alpha_k|^2}} & \frac{\alpha_n^*\alpha_m}{2(\sum_k |\alpha_k|^2)} & \frac{\alpha_n^*}{4\sqrt{\sum_k |\alpha_k|^2}} \\
\hline
\frac{1}{4} & \frac{\alpha_m}{4\sqrt{\sum_k |\alpha_k|^2}} & \frac{1}{2}
\end{array}
\right), \quad
\mathbf{F}^{\emptyset} = \left(
\begin{array}{c|c|c}
0 & \frac{\alpha_m^*}{4\sqrt{\sum_k |\alpha_k|^2}} & -\frac{1}{4} \\
\hline
\frac{-\alpha_n^*}{4\sqrt{\sum_k |\alpha_k|^2}} & 0 & \frac{\alpha_n^*}{4\sqrt{\sum_k |\alpha_k|^2}} \\
\hline
\frac{1}{4} & \frac{-\alpha_m^*}{4\sqrt{\sum_k |\alpha_k|^2}} & 0
\end{array}
\right)
$$

$$
\mathbf{G}^{\emptyset} = \left(
\begin{array}{c|c|c}
0 & \frac{\Re[\alpha_m]}{\sqrt{\sum_k |\alpha_k|^2}} & 0 \\
\hline
0 & -\delta_{nm} + \frac{\Re[\alpha_n^*\alpha_m]}{\sum_k |\alpha_k|^2} & \frac{\Re[\alpha_n]}{\sqrt{\sum_k |\alpha_k|^2}} \\
\hline
1 & 0 & 0
\end{array}
\right), \qquad
\mathbf{K}^{\emptyset} = \mathbf{K}^0, \qquad \bar{\mathbf{K}}^{\emptyset} = \bar{\mathbf{K}}^0.
$$

Finally, for this case when the state is $|G_{\pm}\rangle$ then the correlations look like below.

$$
\mathbf{C}^{\pm} = \left(
\begin{array}{c|c|c}
\frac{1}{2} & \frac{\alpha_m}{4\sqrt{\sum_k |\alpha_k|^2}} & 0 \\
\hline
\frac{\alpha_n^*}{4\sqrt{\sum_k |\alpha_k|^2}} & \frac{\alpha_n^*\alpha_m}{2(\sum_k |\alpha_k|^2)} & \frac{\alpha_n^*}{4\sqrt{\sum_k |\alpha_k|^2}} \\
\hline
0 & \frac{\alpha_m}{4\sqrt{\sum_k |\alpha_k|^2}} & \frac{1}{2}
\end{array}
\right), \quad
\mathbf{F}^{\pm} = \left(
\begin{array}{c|c|c}
0 & \frac{\alpha_m^*}{4\sqrt{\sum_k |\alpha_k|^2}} & 0 \\
\hline
\frac{-\alpha_n^*}{4\sqrt{\sum_k |\alpha_k|^2}} & 0 & \frac{\alpha_n^*}{4\sqrt{\sum_k |\alpha_k|^2}} \\
\hline
0 & \frac{-\alpha_m^*}{4\sqrt{\sum_k |\alpha_k|^2}} & 0
\end{array}
\right),
$$

$$
\mathbf{G}^{\pm} = \left(
\begin{array}{c|c|c}
0 & \frac{\Re[\alpha_m]}{\sqrt{\sum_k |\alpha_k|^2}} & 0 \\
\hline
0 & -\delta_{nm} + \frac{\Re[\alpha_n^*\alpha_m]}{\sum_k |\alpha_k|^2} & \frac{\Re[\alpha_n]}{\sqrt{\sum_k |\alpha_k|^2}} \\
\hline
0 & 0 & 0
\end{array}
\right), \qquad
\mathbf{K}^{\pm} = \mathbf{K}^0, \qquad \bar{\mathbf{K}}^{\pm} = \bar{\mathbf{K}}^0.
$$

## Appendix E    M matrix for modified $XY$ chain

In this appendix, the exact forms of the matrices **A** and **B** for modified $XY$ spin chain is demonstrated. For $|\vec{b}_1| = 1$ and $|\vec{b}_L| = 1$, they can be written as:

$$
\mathbf{A} = \begin{pmatrix}
0 & -\frac{\sin\theta_1 e^{i\phi_1}}{2} & 0 & \dots \\
-\frac{\sin\theta_1 e^{-i\phi_1}}{2} & -h + \cos\theta_1 & -\frac{J}{2} & 0 & \dots \\
0 & -\frac{J}{2} & -h & -\frac{J}{2} & 0 & \dots \\
0 & 0 & -\frac{J}{2} & -h & -\frac{J}{2} & 0 & \dots \\
\cdot & \cdot & \cdot \\
\cdot & \cdot & \cdot \\
& & & & \cdot & \cdot & -h + \cos\theta_L & -\frac{\sin\theta_L e^{-i\phi_L}}{2} \\
& & & & \dots & 0 & -\frac{\sin\theta_L e^{i\phi_L}}{2} & 0
\end{pmatrix}, \tag{E.1}
$$

$$
\mathbf{B} = \begin{pmatrix}
0 & -\frac{\sin\theta_1 e^{-i\phi_1}}{2} & 0 & \cdots \\
\frac{\sin\theta_1 e^{-i\phi_1}}{2} & 0 & -\frac{J\gamma}{2} & 0 & \cdots \\
0 & \frac{J\gamma}{2} & 0 & -\frac{\gamma J}{2} & 0 & \dots \\
\vdots & 0 & \frac{J\gamma}{2} & 0 & -\frac{\gamma J}{2} & 0 & \dots \\
\cdot & \cdot \\
\cdot & \cdot \\
\cdot & \cdot & \cdot & \cdot & \cdot & \frac{J\gamma}{2} & 0 & -\frac{\sin\theta_L e^{-i\phi_L}}{2} \\
\cdot & \cdot & \cdot & \cdot & \cdot & 0 & \frac{\sin\theta_L e^{-i\phi_L}}{2} & 0
\end{pmatrix}. \tag{E.2}
$$

With the help of (8) and (9), one can construct the actual **M** matrix.

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
