# Peer review of "Entanglement entropy in quantum spin chains with broken parity number symmetry"

_SciPost Physics, doi:SciPost Phys. 12, 195 (2022)_

## Round 1 · Referee Report · Anonymous · 2022-4-4

Strengths
1 - Clearness in introducing the subject and explaining plainly the logic of the progresses
2 - Balanced and well organised presentation
3 - Giving examples by choosing important models like the XY chain, etc... that sound familiar to the reader and so improve understanding.
Report
The paper deals with the diagonalization of spin chain hamiltonians with non quadratic boundary terms, by use of a trick of adding ancillary spins at the boundary that make the hamiltonian quadratic and hence diagonalisable. Energy levels, correlation functions and reduced density matrices are computed and, out of these, entanglement and Renyi entropies evaluated. Subtle technical problems are afforded and solved with elegance.
The paper is clear in its writing, the logical steps and the results are presented lucidly and illustrated with important examples.
Of course, I could not go through and check all the calculations, but they all seem very sound. Where I could verify, they appear to be absolutely trustable.
Citations are, up to my knowledge, correct and exhaustive, without exceeding.
As the paper is well inserted into the recent stream of developments about spin chains and related integrable models, their entanglement, etc... and the relations all this has with non-equilibrium physics and quantum information, my opinion is that it surely deserves to be published on SciPost.
Anonymous on 2022-05-14 [id 2464]
The authors analyze quantum spin chains coupled with arbitrary boundary magnetic fields,
which give non-quadratic Hamiltonians of fermions after the Jordan-Wigner (JW) transformation.
However, introducing ancillary spins to the original spin systems leads to quadratic
Hamiltonians of fermions after the JW transformation.
General properties of such quadratic fermion systems are investigated, especially energy eigenstates,
correlation functions, reduced density matrices and entanglement entropies. Results for the original spin
systems can be obtained by projecting to an appropriate sector of the above eigenstates of the fermion
systems, which breaks the parity number symmetry.
I think that this paper is elaborated and interesting. I would like to recommend the publication after
the points below are revised.
page 10, eq.(31): |0> is used without its definition. It should be |0>_{\eta}.
The same is found at the end of the line below eq.(39).
Although it is mentioned in the last sentence of section 3.5, I recommend to explicitly
write as |0> \equiv |0>_{\eta}, if the authors want to use |0>.
page 13, 3rd line below eq.(53): Two notations of the boundary magnetic fields are used. One is \vec{b}
here and below, and the other is \vec{B} around eqs. (2) and (3). They should be unified.
page 19, 2nd line of section 5: ... result of 3.4 and ... -> ... result of subsection 3.4 and ...
page 21, eqs. (82b)-(82d): I think that it is better if the length of the paper can be reduced.
The eqs.(82b)-(82d) are not necessary because they are already given
in (79b)-(79e). Similarly, eqs.(93a), (93b), (95a), (95b) and (99a)-(99d) are
not necessary.
page 22, 3rd line from the bottom: From (84), ... -> From (85), ...
page 27, above eq.(105): \alpha -> 1 should be changed to n -> 1.
The abbreviation EE is used without its definition.
page 8, 1st line: subsection (5.3) -> subsection 5.3
page 29, eq.(113): The superscript \pm should be put to \nu_j.
page 30: 2nd line below eq.(119): guarantied -> guaranteed
page 33, below eq.(126): On the RHS of \vec{b}_L= ..., \theta_1 and \varphi_1 should be \theta_L and \varphi_L,
respectively.
2nd line of section 8.2.1: section (3.6) -> section 3.6
page 37, 2nd and 3rd lines of the caption of Fig.6: \phi should be changed to \varphi.
page 50, above eq.(D.5): eigenvectros -> eigenvectors
The same is found above eq.(D.7).
page 53, eqs.(E.1) and (E.2): \phi_1 and \phi_L should be changed to \varphi_1 and \varphi_L, respectively.
Author: Arash Jafarizadeh on 2022-05-18 [id 2489]
(in reply to Anonymous Comment on 2022-05-14 [id 2464])We appreciate your detailed review of our paper. Indeed points brought up by you improve the presentation of the paper. Corrections will be made.
Thank you.

---

## Editorial Decision

published